# Prices, Bids, Values: One ML-Powered Combinatorial Auction to Rule Them All

**Ermis Soumalias** [* 1 2]   **Jakob Heiss** [* 2 3 4]   **Jakob Weissteiner** [1 2 5]   **Sven Seuken** [1 2]

## Abstract

We study the design of *iterative combinatorial auctions (ICAs)*. The main challenge in this domain is that the bundle space grows exponentially in the number of items. To address this, recent work has proposed machine learning (ML)-based preference elicitation algorithms that aim to elicit only the most critical information from bidders to maximize efficiency. However, while the SOTA ML-based algorithms elicit bidders' preferences via *value queries*, ICAs that are used in practice elicit information via *demand queries*. In this paper, we introduce a novel ML algorithm that provably makes use of the full information from both value and demand queries, and we show via experiments that combining both query types results in significantly better learning performance in practice. Building on these insights, we present MLHCA, a new ML-powered auction that uses value and demand queries. MLHCA significantly outperforms the previous SOTA, reducing efficiency loss by up to a factor 10, with up to 58% fewer queries. Thus, MLHCA achieves large efficiency improvements while also reducing bidders' cognitive load, establishing a new benchmark for both practicability and efficiency. Our code is available at https://github.com/marketdesignresearch/MLHCA.

## 1. Introduction

*Combinatorial auctions (CAs)* are used to allocate multiple items among several bidders who may view those items as complements or substitutes. CAs allow bidders to place bids on entire *bundles* of items, enabling more nuanced ex-pression of value. CAs have enjoyed widespread adoption in practice, with their applications ranging from allocating spectrum licenses (Cramton, 2013) to TV ad slots (Goetzendorff et al., 2015) and airport landing/take-off slots (Rassenti et al., 1982).

The key challenge in CAs is that the bundle space grows exponentially in the number of items, making it impossible for bidders to report their full value function in all but the smallest domains. Moreover, Nisan & Segal (2006) showed that, for arbitrary value functions, CAs require an exponential number of bids to guarantee full efficiency. Thus, practical CA mechanisms cannot provide efficiency guarantees in real world settings with more than a modest number of items. Instead, the focus has shifted towards *iterative combinatorial auctions (ICAs)*, where bidders interact with the auctioneer over a series of rounds, providing only a limited (i.e., practically feasible) amount of information, with the aim to maximize the efficiency of the final allocation.

The most established ICA following this interaction paradigm is the *combinatorial clock auction (CCA)* (Ausubel et al., 2006). Extensively used for allocating spectrum licenses, the CCA generated over *USD* 20 *billion* in revenue between 2012 and 2014 alone (Ausubel & Baranov, 2017). However, a key challenge for any ICA, including the CCA, is balancing *speed of convergence* with efficiency. Each bidding round involves significant computational costs and complex business modeling for participants (Kwasnica et al., 2005; Milgrom & Segal, 2017; Bichler et al., 2017), making faster convergence highly desirable.

Large spectrum auctions conducted under the CCA format can require over 100 bidding rounds, prompting practitioners to adopt aggressive price update rules to reduce the number of rounds. For example, prices may be increased by up to 10% per round, but such approaches come at the expense of efficiency (Ausubel & Baranov, 2017). This trade-off highlights the ongoing challenge of designing ICAs that achieve both high efficiency and rapid convergence. Given the value of resources allocated in these auctions, even a one-percentage-point improvement in efficiency translates to welfare gains of hundreds of millions of dollars.

---

[*]Equal contribution  [1]Department of Informatics, University of Zurich, Zurich, Switzerland  [2]ETH AI Center, Zurich, Switzerland  [3]Department of Mathematics, ETH Zurich, Zurich, Switzerland  [4]Department of Statistics, University of California, Berkeley, USA  [5]UBS, Zurich, Switzerland. Correspondence to: Ermis Soumalias <ermis@ifi.uzh.ch>, Jakob Heiss <jakob.heiss@berkeley.edu>.

*Proceedings of the $42^{nd}$ International Conference on Machine Learning*, Vancouver, Canada. PMLR 267, 2025. Copyright 2025 by the author(s).

## 1.1. ML-Powered Iterative Combinatorial Auctions

To tackle this challenge, researchers have explored using *machine learning (ML)* to enhance the efficiency of ICAs. The foundational works of Blum et al. (2004) and Lahaie & Parkes (2004) were the first to frame preference elicitation in CAs as a learning problem. More recently, Brero et al. (2018; 2021) and Weissteiner & Seuken (2020); Weissteiner et al. (2022b;a; 2023) introduced ML-powered ICAs. Central to these approaches is an ML-based preference elicitation algorithm that trains an ML model on each bidder's value function to generate informative *value queries (VQs)* (e.g., "What is your value for the bundle $\{A, B\}$?"), which iteratively refine the ML model of each bidder's values.[1]

Soumalias et al. (2024c) took a different approach. To increase the likelihood of their approach being adopted in practice, they introduced ML-CCA, an ML-powered auction that follows the established interaction paradigm of the CCA using demand queries. Building on earlier works by Brero & Lahaie (2018); Brero et al. (2019), their design iteratively trains individual ML models for each bidder using their previously answered *demand queries (DQs)* and then selects the next DQ with the highest clearing potential.

Although ML-CCA marked a major step towards a practical ML-powered ICA and outperformed the baseline CCA used in real-world applications, it still faced two key shortcomings. First, it fell short of achieving the SOTA efficiency of the VQ-based ML-powered ICAs. Second, like the CCA, it relied on a very large number of *supplementary round bids* to enhance its efficiency, requiring bidders to decide on additional *value bids*—a cognitively demanding task.

We address these shortcomings by introducing the *Machine Learning-powered Hybrid Combinatorial Auction (MLHCA)*. Leveraging sophisticated DQ and VQ generation algorithms, MLHCA maintains the established interaction paradigm of the CCA while achieving unprecedented efficiency gains. MLHCA outperforms the previous SOTA across all tested domains, reducing efficiency loss by up to a factor of ten. Based on the value of goods traded (Ausubel & Baranov, 2017), these efficiency improvements correspond to welfare gains of hundreds of millions of USD. At the same time, MLHCA significantly reduces the cognitive load on bidders: compared to BOCA, the previous SOTA, MLHCA requires at least 42% fewer queries to achieve the same efficiency, and compared to ML-CCA, the SOTA auction following CCA's interaction paradigm, MLHCA requires at least 26% fewer queries. Moreover, unlike the CCA and ML-CCA, in MLHCA bidders do not need to decide which bundles to bid for in its VQ rounds, as the auction automatically suggests these bundles. Thus, MLHCA achieves

unprecedented efficiency gains while significantly reducing bidders' cognitive load, establishing a new benchmark for both practicability and efficiency.

## 1.2. Our Contributions

We introduce the *Machine Learning-powered Hybrid Combinatorial Auction (MLHCA)*, a practical ICA that achieves unprecedented efficiency and convergence speed. First, we establish a theoretical foundation and provide illustrative examples to demonstrate the advantages and limitations of DQs and VQs from an auction design perspective (Section 3). Then we develop a learning algorithm that effectively leverages both query types (Section 4). We provide strong experimental evidence of the learning benefits of combining both query types, as well as the advantages of starting an auction with DQs instead of VQs.

We then integrate these auction and ML insights to design MLHCA, the first ICA to incorporate sophisticated DQ and VQ generation algorithms (Section 5). Simulations in realistic domains (Section 6) show that MLHCA significantly outperforms the previous SOTA, achieving unprecedented efficiency while also using fewer queries, thus setting a new benchmark for both efficiency and practicality.

## 1.3. Further Related work

In the field of *automated mechanism design*, Dütting et al. (2015; 2019), Golowich et al. (2018) and Narasimhan et al. (2016) used ML to learn new mechanisms from data, while Cole & Roughgarden (2014); Morgenstern & Roughgarden (2015) and Balcan et al. (2023) bounded the sample complexity of learning approximately optimal mechanisms. In contrast to this line of prior work, in our design, the ML algorithm is part of the mechanism itself. Lahaie & Lubin (2019) suggest an adaptive price update rule that increases price expressivity as the rounds progress in order to improve efficiency and speed of convergence. Unlike that work, we aim to improve efficiency without increasing price expressivity, as that is not a popular interaction paradigm in practice, and can cause added cognitive load on the bidders. Preference elicitation is also a key challenge in combinatorial allocation without money. Soumalias et al. (2024b) introduce an ML-powered mechanism for course allocation that improves preference elicitation by asking comparison queries. See Appendix A.1 for further related work.

## 1.4. Practical Considerations and Incentives

MLHCA can be seen as a sophisticated modification of the CCA. In practice, many other considerations (beyond preference elicitation complexity and efficiency) are important. For example, Ausubel & Baranov (2017) discussed the vital role of well-designed *activity rules* to induce truthful bid-

---

[1]From an optimization perspective, this can be viewed as a combinatorial Bayesian optimization problem.

ding in the clock phase of the CCA. In Appendix B.3, we provide a detailed discussion of the most common activity rules used in the CCA, and we detail how MLHCA can also leverage these rules for the same goal. Additionally, in Appendix B.4, we prove that MLHCA can immediately detect if a bidder's reports are inconsistent.

The payment rule used in the supplementary round of the CCA is also important for incentives. Cramton (2013) argued that the use of the VCG-nearest payment rule, while not strategyproof, induces good incentives in practice. Similar to the supplementary round of the CCA, the VQ-based phase of MLHCA is not strategyproof. However, in Appendix B.5, we argue that the VQ-based phase of MLHCA offers strong incentives in practice, and we show that, under two additional assumptions, truthful bidding is an ex-post Nash equilibrium (following the arguments from Brero et al. (2021) for the MLCA).

## 2. Preliminaries

### 2.1. Formal Model for ICAs

We consider *multiset* CA domains with a set $N = \{1, \ldots, n\}$ of bidders and a set $M = \{1, \ldots, m\}$ of distinct items with corresponding *capacities*, i.e., number of available copies, $c = (c_1, \ldots, c_m) \in \mathbb{N}^m$. We denote by $x \in \mathcal{X} = \{0, \ldots, c_1\} \times \ldots \times \{0, \ldots, c_m\}$ a bundle of items represented as a positive integer vector, where $x_j = k$ iff item $j \in M$ is contained $k$-times in $x$. The bidders' true preferences over bundles are represented by their (private) *value functions* $v_i : \mathcal{X} \to \mathbb{R}_{\geq 0}$, $i \in N$, i.e., $v_i(x)$ represents bidder $i$'s value for bundle $x \in \mathcal{X}$. We assume that $v_i$ is nondecreasing and satisfies $v_i(0) = 0$. We collect the value functions $v_i$ in the vector $v = (v_i)_{i \in N}$. By $a = (a_1, \ldots, a_n) \in \mathcal{X}^n$ we denote an *allocation* of bundles to bidders, where $a_i$ is the bundle bidder $i$ obtains. We denote the set of *feasible* allocations by $\mathcal{F} = \{a \in \mathcal{X}^n : \sum_{i \in N} a_{ij} \leq c_j, \ \forall j \in M\}$. We assume that bidders have *quasilinear utility functions* of the form $u_i(a_i) = v_i(a_i) - \pi_i$ where $v_i$ can be highly non-linear and $\pi_i \in \mathbb{R}_{\geq 0}$ denotes the bidder's payment. This implies that the (true) *social welfare* $V(a)$ of an allocation $a$ is equal to the sum of all bidders' values $\sum_{i \in N} v_i(a_i)$.[2] We let $a^* \in \arg\max_{a \in \mathcal{F}} V(a)$ denote a social-welfare maximizing, i.e., *efficient*, allocation. The *efficiency* of any allocation $a \in \mathcal{F}$ is determined as $V(a)/V(a^*)$.

An ICA *mechanism* defines how the bidders interact with the auctioneer and how the allocation and payments are determined. We consider ICAs that iteratively ask bidders both *demand queries* (DQs) and *value queries* (VQs).

**Definition 2.1** (Demand Query). In a (linear) demand query,

the auctioneer presents a vector of item prices $p \in \mathbb{R}_{\geq 0}^m$ and each bidder $i$ responds with her utility-maximizing bundle,

$$x_i^*(p) \in \arg\max_{x \in \mathcal{X}} \{v_i(x) - \langle p, x \rangle\} \ i \in N, \qquad (1)$$

where $\langle \cdot, \cdot \rangle$ denotes the Euclidean scalar product in $\mathbb{R}^m$.

**Definition 2.2** (Value Query). In a value query, the auctioneer presents to bidder $i$ a bundle of items $x$ and bidder $i$ responds with her value at those prices, i.e., $v_i(x) \in \mathbb{R}_{\geq 0}$.

For bidder $i \in N$, we denote her $K \in \mathbb{N}$ elicited DQs as $R_i^{\text{DQ}} = \{(x_i^*(p^r), p^r)\}_{r=1}^K$ and her $L \in \mathbb{N}$ elicited VQs as $R_i^{\text{VQ}} = \{(x_i^l, v_i(x_i^l))\}_{l=1}^L$. Bidder $i$'s reports are denoted as $R_i = (R_i^{\text{DQ}}, R_i^{\text{VQ}})$. We collect the elicited reports of all bidders in the tuple $R = (R_1, \ldots, R_n)$.

In auctions using DQs, a key concept is the bidder's *inferred value*. This represents the maximum lower bound on a bidder's value for a bundle that the auctioneer can deduce from the bidder's reports, without assuming monotonicity. The inferred value is always weakly lower than the bidder's true value, with equality achieved if the bidder has answered the corresponding VQ for that bundle. Formally:

**Definition 2.3** (Inferred Value). Bidder $i$'s inferred value for bundle $x \in \mathcal{X}$ given her reports $R_i$ is

$$\widetilde{v}_i(x; R_i) = \begin{cases} v_i(x), & \text{if } (x, v_i(x)) \in R_i^{\text{VQ}}, \\ \max\left\{\{\langle x, p^r \rangle : (x, p^r) \in R_i^{\text{DQ}}\} \cup \{0\}\right\}, & \text{else.} \end{cases} \qquad (2)$$

The ICA's final allocation $a^*(R) \in \mathcal{F}$ and payments $\pi_i := \pi_i(R) \in \mathbb{R}_{\geq 0}^n$ are computed based *only* on the *elicited* reports $R$. Concretely, $a^*(R) \in \mathcal{F}$ is determined by solving the *Winner Determination Problem (WDP)*:

$$a^*(R) \in \arg\max_{a \in \mathcal{F}} \sum_{i \in N} \widetilde{v}_i(a_i; R_i), \qquad (3)$$

where $\sum_{i \in N} \widetilde{v}_i(a_i; R_i)$ is the allocation's *inferred social welfare*, a lower bound on its *social welfare* $\sum_{i \in N} v_i(a_i)$.

### 2.2. Benchmark ICAs

In this section, we briefly introduce the three main benchmark mechanisms considered in this paper.

**CCA** The most established ICA is the *Combinatorial Clock Auction (CCA)* (Ausubel et al., 2006). The CCA consists of two phases. The initial *clock phase* proceeds in rounds. In each round $r$, the auctioneer sets anonymous (i.e., same prices for all bidders) item prices $p^r \in \mathbb{R}_{\geq 0}^m$, prompting each bidder to respond to a DQ, declaring her utility-maximizing bundle at $p^r$. In the next round, the prices of over-demanded items are increased by a fixed percentage, until over-demand is eliminated. The second phase

---

[2]Note that $V(a) = \sum_{i \in N} u_i(a_i) + u_{\text{auctioneer}}(a) = \sum_{i \in N} (v_i(a_i) - \pi_i) + \sum_{i \in N} \pi_i = \sum_{i \in N} v_i(a_i)$.

of the CCA, known as the *supplementary round*, allows bidders to report their *values* for additional bundles of their choice. The *clock bids raised heuristic* suggests that bidders report their values for all bundles they requested during the clock phase. The final allocation is determined by solving the WDP based on all reports as in Equation (3).

**ML-CCA** The most efficient DQ-based ICA is the *Machine Learning-powered Combinatorial Clock Auction (ML-CCA)* (Soumalias et al., 2024c). ML-CCA has the same interaction paradigm as the CCA, but with a substantially more refined DQ-generation algorithm in its clock phase. In each round, an ML model is trained to estimate each bidder's value function based on previously submitted DQ responses. Then, the prices are not increased by a percentage like in the CCA, but instead a convex optimization problem determines the prices with the highest clearing potential.

**BOCA** The SOTA ICA in terms of efficiency is the VQ-based *Bayesian optimization-based combinatorial auction (BOCA)* (Weissteiner et al., 2023). The main idea of BOCA is that in each round, the auctioneer creates an estimate of the upper confidence bound of the value function of each agent based on her past responses. Then, the auctioneer solves an ML-based WDP to find the feasible allocation with the highest upper bound on its estimated social welfare, and queries each agent her value for her bundle in that allocation. This allows the mechanism to balance between exploring and exploiting the bundle space.

### 2.3. ML Framework

The ML models used by ML-CCA, and as basis for the construction of the confidence bound estimates in BOCA are *monotone-value neural networks (MVNNs)* $\mathcal{M}^\theta : \mathcal{X} \to \mathbb{R}$ (Weissteiner et al., 2022a). MVNNs are a class of NNs specifically designed to represent *monotone combinatorial* valuations. MVNNs have also had success in combinatorial allocation domains without money, e.g., for course allocation (Soumalias et al., 2024b). Soumalias et al. (2024c) introduced *multiset MVNNs (mMVNNs)*, an extension of MVNNs that also incorporates at a structural level the information that some items in the auction are identical copies of each other. In this work, we instantiate our ML models using mMVNNs, and denote agent $i$'s model as $\mathcal{M}_i^\theta : \mathcal{X} \to \mathbb{R}$. Within this work, we will refer to all mMVNNs simply as MVNNs. We provide more details in Appendix C.

## 3. A Theoretical Framework for Effectively Combining DQs and VQs

This section develops a theoretical framework for effectively combining DQs and VQs. Proofs are deferred to Appendix D.

**VQ-Based Approaches Rely on Cognitively Complex Random VQs.** At the start of an ICA, no specific information about bidders' preferences is available, making it challenging to identify bundles relevant to them. Thus, most VQ-based auctions, including the SOTA approach (Weissteiner et al., 2023), begin by querying bidders about randomly selected bundles. However, in practice, answering VQs for such random bundles that do not align with the bidders' interests is cognitively demanding. In contrast, the most widely used ICAs in practice (e.g., the CCA) employ DQs, which are easier for bidders to answer effectively (Cramton, 2013). This key advantage highlights why relying exclusively on VQs is often impractical in real-world auctions.

**DQs Offer Superior Efficiency Gains in Initial Rounds.** Even if bidders could easily answer random VQs, in Appendix D.1 we detail the significant advantages of DQs in the initial rounds of an ML-ICA, where DQs are providing more actionable and efficient information. This superior efficiency is formalized in the following proposition:

**Proposition 3.1.** *The expected social welfare of an auction that uses a single random demand query can be arbitrarily larger than that of an auction that uses any constant number ($k \ll 2^m$) of random value queries.*

Additionally, DQs can establish a proof of optimality, allowing the auction to terminate early (Proposition D.3). These theoretical insights are validated by our experimental results in Section 6. An auction initialized with DQs has up to 20% points higher efficiency after its initial queries compared to an auction initialized with VQs.

**VQs Offer Superior Efficiency Gains in Later Rounds.** This raises the question: is it sufficient to rely exclusively on DQs? Theorem 3.2 proves that the answer is negative:

**Theorem 3.2.** *For every $\epsilon > 0$, there exist infinitely many instances of auctions for which no combination of DQs can achieve an efficiency above $50\% + \epsilon$. This remains true even for infinite combinations of DQs and even if the bidders additionally report their true values for all bundles they requested in those DQs.*

Notably, Theorem 3.2 shows that this limitation persists even when supplementing the auction with the *clock-bids raised* heuristic. Without this heuristic, adding more DQs can even *decrease* efficiency. In Proposition 3.3 we prove that a single DQ can reduce the auction's efficiency arbitrarily close to 100%, whereas VQ-based auctions do not face this issue.

**Proposition 3.3.** *In a DQ-based ICA, adding DQs can actually reduce efficiency. A single DQ can cause an efficiency drop arbitrarily close to $100\%$. By comparison, in a*

*VQ-based ICA, adding additional queries can never reduce efficiency (assuming truthful bidding).*

These issues are also very prevalent in the real world. In Section 6, we demonstrate that in realistic domains, the gap in *average* efficiency between the SOTA DQ-based and VQ-based auctions can reach up to 8% points. Furthermore, the *average* efficiency of the CCA, the prominent DQ-based auction, declines by 8% points during the auction. VQ-based auctions avoid these pitfalls entirely. First, they can always achieve 100% efficiency after sufficiently many VQs (Lemma D.14). Second, asking additional VQs never reduces the efficiency of a VQ-based auction. In Appendix D.2, we provide further theoretical and intuitive arguments on why relying solely on DQs is insufficient.

**Optimally Combining DQs and VQs.** Building on the discussion in this section, it is natural to leverage the strengths of both query types by starting with DQs and then transitioning to VQs. Example 1 in Appendix D.3 illustrates why this approach is effective: even after infinitely many DQs could not achieve more than 55% efficiency, a single VQ can achieve 100% efficiency. However, caution is required, as introducing even a single VQ can reduce the auction's efficiency by nearly 100% (Lemma D.7). To address this, we introduce the *bridge bid*, a specialized VQ designed to seamlessly connect the DQ and VQ phases of a hybrid auction. The bridge bid asks each bidder her value for the bundle she would have received according to the WDP (Equation (3)) after the final DQ round. Incorporating the bridge bid guarantees that the auction's final efficiency will be no less than its DQ-only efficiency (Lemma D.9). We demonstrate the significance of this bid in practice in Section 6.3 and Appendix G.8. Appendix D.3 provides further insights into combining DQs with VQs.

# 4. Mixed Query Learning

Combining DQs and VQs not only improves the final efficiency of auctions but also enables the global learning of bidders' value functions. In this section, we introduce a mixed training algorithm that leverages both query types. Specifically, we demonstrate the learning benefits of initializing auctions with DQs over VQs and show how integrating both query types leads to superior learning performance. Further details are presented in Appendix E.

## 4.1. Mixed Training Algorithm

To leverage the advantages of both DQs and VQs, we propose a two-stage training algorithm compatible with modern NN architectures, including mMVNNs. In each epoch, the ML model is first trained on all DQ responses using the loss function of Soumalias et al. (2024c). The key idea is to predict the bidder's utility-maximizing bundle at the given

| OPTIMIZATION | TRAIN POINTS | | $R^2$ | | KT | | MAE SCALED | | $R_c^2$ | |
|---|---|---|---|---|---|---|---|---|---|---|
| METRIC | VQs | DQs | $\mathcal{T}_r$ | $\mathcal{T}_p$ | $\mathcal{T}_r$ | $\mathcal{T}_p$ | $\mathcal{T}_r$ | $\mathcal{T}_p$ | $\mathcal{T}_r$ | $\mathcal{T}_p$ |
| $R^2$ ON $\mathcal{V}_r$ | 20 | 40 | 0.84 | 0.42 | 0.79 | 0.80 | 0.037 | 0.044 | 0.84 | 0.80 |
| | 60 | 0 | 0.73 | −10.07 | 0.68 | 0.64 | 0.052 | 0.236 | 0.74 | 0.20 |
| | 0 | 60 | 0.24 | −3.07 | 0.77 | 0.77 | 0.103 | 0.128 | 0.83 | 0.76 |
| $R^2$ ON $\mathcal{V}_p$ | 20 | 40 | 0.82 | 0.01 | 0.79 | 0.80 | 0.041 | 0.062 | 0.84 | 0.83 |
| | 60 | 0 | 0.76 | −3.40 | 0.72 | 0.62 | 0.049 | 0.141 | 0.77 | 0.05 |
| | 0 | 60 | −0.05 | −6.24 | 0.78 | 0.72 | 0.103 | 0.154 | 0.84 | 0.69 |

Table 1: Learning comparison of training only on DQs, only on VQs, or on both. Shown are averages over ten instances. Winners marked in gray.

prices by treating her ML model as her true value function. If the predicted reply deviates from the bidder's true reply, the loss equals the difference in predicted utility between the two bundles. This loss function provably captures all information provided by the DQ responses. Additionally, the model is trained on the VQ responses using a standard regression loss. For details, please refer to Appendix E.1.

## 4.2. Experimental Analysis

We demonstrate the learning benefits of initializing auctions with DQs rather than VQs and highlight how combining both query types leads to superior learning performance.

We conduct the following experiment: We perform *hyperparameter optimization (HPO)* to train an MVNN for the most critical bidder in the most realistic simulation domain (see Appendix G.1 for details on the simulation and Appendix E.3 for results for other domains). For this bidder, we generate three training sets: (1) 40 DQs simulating 40 CCA clock rounds and 20 random VQs, (2) 60 DQs simulating 60 clock rounds with no VQs, and (3) 60 random VQs with no DQs. The models are evaluated on two validation sets: a *random bundle set* ($\mathcal{V}_r$) with 50,000 uniformly sampled bundles, and a *price-driven set* ($\mathcal{V}_p$) containing bundles requested under 200 random price vectors. $\mathcal{V}_r$ tests generalization across all bundles, while $\mathcal{V}_p$ focuses on utility-maximizing bundles. We select the configuration with the best coefficient of determination ($R^2$), averaged over 10 bidder instances.

The selected configurations are then tested on 10 new bidders, generating hold-out tests sets $\mathcal{T}_r$ and $\mathcal{T}_p$ in the same way as $\mathcal{V}_p$ and $\mathcal{V}_r$. We report $R^2$, *Kendall Tau (KT)*, *scaled Mean Absolute Error (scaled MAE)*, and $R_c^2$. An $R_c^2$ value of 1 indicates perfect learning up to a constant shift, with differences between $R_c^2$ and $R^2$ reflecting shift magnitude. HPO procedures were consistent across all training sets, with identical test instances, seeds, search spaces, and computation time. Additional details are in Appendix G.3.

Table 1 shows that training on a mix of DQs and VQs consistently outperforms training on either query type, particularly for utility-maximizing bundles in $\mathcal{T}_p$, where mixed training

achieves nearly three times lower MAE. The mixed model also closely matches the mean value for both test sets, as indicated by the small gap between $R_c^2$ and $R^2$. In contrast, DQ-only models lack absolute value information, leading to relative but not unique value function learning, as evidenced by the large difference between $R^2$ and $R_c^2$. This limitation goes even beyond constant shifts: Example 1 shows that even with all possible DQs, unique identification up to a constant shift is impossible. Meanwhile, VQ-only models suffer from distributional shifts between test sets, reflected in the significant discrepancy in $R^2$ and MAE across the two test sets. These shifts prevent VQ-trained models from capturing critical, high-value bundles due to the absence of utility-maximizing bundles in their training data.[3]

DQ-trained models generalize better to $\mathcal{T}_p$ than VQ-trained models, as $\mathcal{T}_p$ emphasizes high-value bundles critical for efficient allocations. This motivates initially training with DQs, as they provide global information about the allocation space and focus on high-value regions from the outset. These learning advantages are so pronounced that, as shown in Section 6, MLHCA only needs to follow up its 40 DQs with at most 18 VQs to outperform the previous SOTA, which requires 100 VQs.

## 5. The Mechanism

In this section, we describe our *ML-powered Hybrid Combinatorial Auction (MLHCA)*, which combines the auction and ML insights from Sections 3 and 4.

We present a simplified version of MLHCA in Algorithm 1. In Lines 2 to 5, we generate the first $Q^{\text{CCA}} \in \mathbb{N}$ DQs using the price update rule of the CCA. Similar to ML-CCA, we use larger price increments to arrive to similar prices as the ML-CCA in fewer rounds. In each of the next $Q^{\text{DQ}} \in \mathbb{N}$ ML-powered rounds, we first train, for each bidder, an mMVNN on her demand responses (Line 8), and call NEXTPRICE (Soumalias et al., 2024c) (see Appendix A.3) to generate the next DQ based on the agents' trained mMVNNs (Line 9). If MLHCA has found market-clearing prices, then the corresponding allocation is efficient and is returned, along with payments $\pi(R)$ according to the deployed payment rule (Line 15). MLHCA is plug-and-play compatible with many payment rules, such as VCG and VCG-nearest. If, by the end of the ML-powered DQs, the market has not cleared we switch to VQ rounds. In the first VQ round (Line 17) we ask each bidder for her *bridge bid* (see Definition D.8). This single VQ bid ensures that the MLHCA's efficiency is lower bounded by the efficiency after just the DQ rounds (Lemma D.9). For a detailed experimental evaluation of the bridge bid see Appendix G.8. In the final $Q^{\text{VQ}} - 1$ VQ rounds, for each bidder, we query her value for the bundle

---

**Algorithm 1:** MLHCA($Q^{\text{CCA}}, Q^{\text{DQ}}, Q^{\text{VQ}}, \pi$)

**Parameters:** $Q^{\text{CCA}}, Q^{\text{DQ}}, Q^{\text{VQ}}$ and $\pi$

1   $R^{\text{VQ}}, R^{\text{DQ}} \leftarrow (\{\})_{i=1}^N, (\{\})_{i=1}^N$

2   **for** $r = 1, ..., Q^{CCA}$ **do**    ▷Draw $Q^{\text{CCA}}$ initial prices

3     $p^r \leftarrow CCA(R^{\text{DQ}})$

4     **foreach** $i \in N$ **do**      ▷Initial DQs

5       $R_i^{\text{DQ}} \leftarrow R_i^{\text{DQ}} \cup \{(x_i^*(p^r), p^r)\}$

6   **for** $r = Q^{CCA} + 1, ..., Q^{CCA} + Q^{DQ}$ **do** ▷ML-powered DQs

7     **foreach** $i \in N$ **do**

8       $\mathcal{M}_i^\theta \leftarrow \text{MIXEDTRAINING}(R_i^{\text{DQ}}, R_i^{\text{VQ}})$
       ▷Algorithm 4

9     $p^r \leftarrow \text{NEXTPRICE}((\mathcal{M}_i^\theta)_{i=1}^n)$    ▷ Appendix A.3

10     **foreach** $i \in N$ **do**

11       $R_i^{\text{DQ}} \leftarrow R_i^{\text{DQ}} \cup \{(x_i^*(p^r), p^r)\}$

12     **if** $\sum_{i=1}^n (x_i^*(p^k))_j = c_j \, \forall j \in M$ **then**
      ▷Market-clearing prices found

13       $a^*(R^{\text{DQ}}, R^{\text{VQ}}) \leftarrow (x_i^*(p^r))_{i=1}^n$

14       $\pi(R^{\text{DQ}}, R^{\text{VQ}}) \leftarrow (\pi_i(R^{\text{DQ}}, R^{\text{VQ}}))_{i=1}^n$

15       **return** $a^*(R^{DQ}, R^{VQ})$ and $\pi(R^{DQ}, R^{VQ})$

16   **foreach** $i \in N$ **do**      ▷Bridge bid

17     $R_i^{\text{VQ}} \leftarrow$ $R_i^{\text{VQ}} \cup \{(a_i^*(R^{\text{DQ}}, R^{\text{VQ}}), v_i(a_i^*(R^{\text{DQ}}, R^{\text{VQ}})))\}$

18   **for** $r = Q^{CCA} + Q^{DQ} + 2, ..., Q^{CCA} + Q^{DQ} + Q^{VQ}$ **do**
   ▷ML-powered VQs

19     **foreach** $i \in N$ **do**

20       $\mathcal{M}_i^\theta \leftarrow \text{MIXEDTRAINING}(R_i^{\text{DQ}}, R_i^{\text{VQ}})$
       ▷Algorithm 4

21     $a \leftarrow \text{NEXTALLOCATION}((\mathcal{M}_i^\theta)_{i=1}^n), R^{\text{DQ}}, R^{\text{VQ}})$
     ▷Appendix F

22     **foreach** $i \in N$ **do**

23       $R_i^{\text{VQ}} \leftarrow R_i^{\text{VQ}} \cup \{(a_i, v_i(a_i))\}$   ▷Value query responses

24   Calculate final allocation $a^*(R^{\text{DQ}}, R^{\text{VQ}})$ as in Equation (3)

25   Calculate payments $\pi(R^{\text{DQ}}, R^{\text{VQ}})$    ▷E.g., VCG (Appendix B)

26   **return** $a^*(R^{DQ}, R^{VQ})$ and $\pi(R^{DQ}, R^{VQ})$

---

she is allocated in the predicted optimal allocation (based on all ML models), under the constraint that she has not answered a VQ for that bundle in the past (Lines 21 to 23).[4] The final allocation and payments are then determined based on all reports (Lines 24 to 25). For details, please see Appendix F.

## 6. Experiments

In this section, we experimentally evaluate MLHCA. We compare its efficiency against BOCA (Weissteiner et al., 2023) and ML-CCA (Soumalias et al., 2024b) the SOTA VQ-based and DQ-based ICAs, respectively.

---

[3]At the start of a VQ-based auction, models are not accurate enough to target value-maximizing bundles.

[4]This VQ algorithm was introduced in Brero et al. (2021) and used in most follow-up work following the MLCA framework.

## 6.1. Experiment Setup

To generate synthetic CA instances, we use the *spectrum auction test suite (SATS)* (Weiss et al., 2017), which includes various value models (domains) designed to simulate different auction environments. Following standard practice in this line of research (e.g., Soumalias et al. (2024c); Weissteiner et al. (2023)), we conduct experiments on the GSVM, LSVM, SRVM, and MRVM domains (see Appendix G.1 for details). SATS provides access to the true optimal allocation $a^* \in \mathcal{F}$, allowing us to measure the *efficiency loss*, defined as $1 - V(a^*(R))/V(a^*)$, where $R$ represents elicited reports. We focus on efficiency rather than revenue, as do all mechanisms we compare against. This is consistent with the primary application of ICAs in spectrum allocation, a government-run operation with a welfare-maximization mandate (Cramton, 2013). For results on revenue, see Appendix G.7. To ensure a fair comparison with prior work, we limit all auction mechanisms to 100 total queries. These consist of 100 VQs for BOCA, 100 DQs for ML-CCA, and 40 DQs and 60 VQs for MLHCA. For BOCA and ML-CCA, we use the best mechanism configurations and hyperparameters reported in their respective papers. For MLHCA's VQ rounds, we performed HPO separately for each bidder type in each domain, as detailed in Appendix E.2. For the DQ rounds, we adopted the HPO parameters reported by Soumalias et al. (2024c), since our learning algorithm, when restricted to DQs, is equivalent to theirs. For further experimental details and analysis of MLHCA's low computational costs, please refer to Appendices G.3 and G.4 respectively.

## 6.2. Efficiency Results

In Table 2, we show the average efficiency loss of each mechanism after 100 queries. For ML-CCA, we also report results if it were supplemented with the clock bids raised heuristic (see Section 2.2), which would involve up to an additional 100 VQs per bidder.[5] Finally, we report the number of queries that MLHCA requires to outperform the final efficiency of each other mechanism, i.e., in GSVM, with 42 queries (40 DQs and 2 VQs) MLHCA statistically outperforms ML-CCA, even if ML-CCA were supplemented with 100 VQs from the clock bids raised heuristic.

In Table 2, we observe that MLHCA significantly outperforms all other mechanisms across all domains. Notably, MLHCA is the *only* mechanism capable of achieving a perfect 100% efficiency in SRVM. Remarkably, it accomplishes this with fewer than 60 queries, while the other mechanisms fail even with 100 queries. In the LSVM domain, MLHCA achieves a 10-fold reduction in efficiency loss compared to BOCA, the previous SOTA. The most realistic domain,

MRVM further highlights MLHCA's superiority. Here, MLHCA exceeds the efficiency of *all* other mechanisms by over 2% points, making MLHCA the first mechanism to substantially outperform CCA. MRVM simulates the 2014 Canadian spectrum auction (Weiss et al., 2017) with a revenue of USD 5.27 billion (Ausubel & Baranov, 2017), where 2% points correspond to over USD 100 million.

Speed of convergence is another critical factor in these auctions. In all domains, MLHCA requires at most 74 queries (40 DQs and 34 VQs) to statistically outperform the final efficiency of both BOCA and ML-CCA, which use 100 VQs and 100 DQs, respectively. Furthermore, in three out of four domains, MLHCA surpasses the 100 DQ efficiency of ML-CCA with only 40 DQs and 2 VQs. These results align with our theoretical analysis in Appendix D.3, where we show that, once DQs have sufficiently informed the bidders' value functions, a single VQ can lead to 100% efficiency.

Figure 1 illustrates the efficiency loss path for all domains, highlighting MLHCA's consistent superiority. Up to query 40, MLHCA and ML-CCA perform identically since both mechanisms employ the same DQs and network configurations during these rounds. However, after query 40, MLHCA's integration of VQs leads to a marked reduction in efficiency loss compared to ML-CCA, aligning with our insights on the efficiency of VQs and on the learning advantages of combining DQs and VQs (Sections 3 and 4). Across all domains, MLHCA also consistently outperforms BOCA, leveraging the early-stage advantages of DQs when ML models are still being quite uninformed and the later-stage learning advantages of combining DQs and VQs.

In summary, MLHCA outperforms both DQ-based and VQ-based SOTA mechanisms in terms of both efficiency and speed of convergence, achieving high efficiency with fewer queries. This makes MLHCA a powerful and practical choice for real-world auction scenarios where high efficiency and rapid convergence are crucial. These empirical findings not only highlight the efficiency and convergence speed of MLHCA but also closely align with our theoretical insights. In the next section, we analyze how these results validate the predictions and theoretical guarantees established in this paper.

## 6.3. Alignment with Theoretical Insights

Figure 1 further validates our theoretical findings. The non-monotonicity of DQ-based mechanisms, as suggested in Proposition 3.3, is evident in the efficiency loss path of both the CCA and the ML-CCA. Notably, in the LSVM domain, the CCA achieves higher *average* efficiency after just 5 DQs compared to 100. Additionally, the comparison between BOCA and ML-CCA underscores the inefficiency of random VQs in the early stages (Proposition 3.1), particularly in the MRVM domain, where BOCA's efficiency loss is

---

[5]In the clock bids raised heuristic, the bidders only need to report their value for each *unique* bundle they bid on during the auction, which, for 100 DQs, can be up to 100 bundles.

| | EFFICIENCY LOSS IN % | | | | | QUERIES TO REJECT NULL HYPOTHESIS | | |
|---|---|---|---|---|---|---|---|---|
| DOMAIN | MLHCA | BOCA | ML-CCA$_\text{CLOCK}$ | ML-CCA$_\text{RAISED}$ | CCA | BOCA $\geq$ MLHCA | ML-CCA$_\text{CLOCK}$ $\geq$ MLHCA | ML-CCA$_\text{RAISED}$ $\geq$ MLHCA |
| GSVM | $0.00 \pm 0.00$ | — | $1.77 \pm 0.68$ | $1.07 \pm 0.37$ | $9.60 \pm 1.49$ | — | 42 | 42 |
| LSVM | $0.04 \pm 0.07$ | $0.39 \pm 0.31$ | $8.36 \pm 1.70$ | $3.61 \pm 0.77$ | $17.44 \pm 1.60$ | 58 | 42 | 43 |
| SRVM | $0.00 \pm 0.00$ | $0.06 \pm 0.02$ | $0.41 \pm 0.11$ | $0.07 \pm 0.02$ | $0.37 \pm 0.11$ | 42 | 42 | 42 |
| MRVM | $4.81 \pm 0.57$ | $7.77 \pm 0.35$ | $6.94 \pm 0.24$ | $6.68 \pm 0.22$ | $7.53 \pm 0.48$ | 54 | 74 | 79 |

Table 2: MLHCA (40DQs + 60VQs) vs BOCA (100VQs), ML-CCA (ML-CCA$_\text{clock}$) (100DQs) and ML-CCA with raised clock bids (ML-CCA$_\text{raised}$) (100DQs and up to 100VQs). Shown are averages and a 95% CI. Winners based on a $t$-test with significance level of 5% are marked in grey.

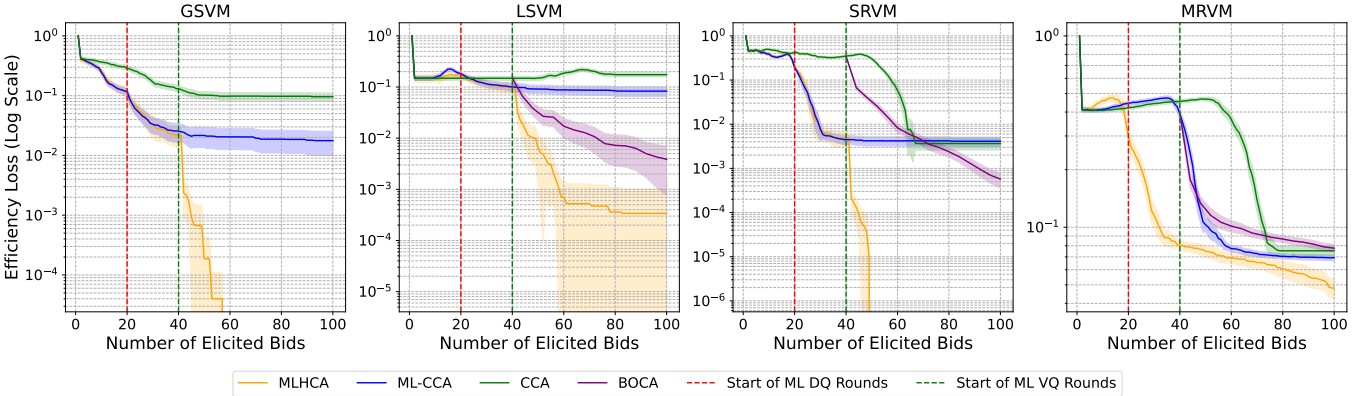

Figure 1: Efficiency loss paths (i.e., regret plots) of MLHCA compared to BOCA, ML-CCA and CCA. Shown are averages over 50 instances with 95% CIs.

orders of magnitude worse than that of mechanisms employing ML-powered DQs. Finally, MLHCA's performance after query 40 demonstrates the potential efficiency gains of supplementing DQs with VQs. The switch to ML-powered VQs results in a dramatic reduction in efficiency loss—by several orders of magnitude in the GSVM and SRVM domains—while the DQ-based ML-CCA, which was identical to MLHCA up to that point, stagnates. This aligns with Theorem 3.2, which proves that once ML models effectively capture bidder preferences, VQs can dramatically enhance efficiency. In contrast, ML-CCA's reliance on DQs prevents further improvements, even with well-trained models.

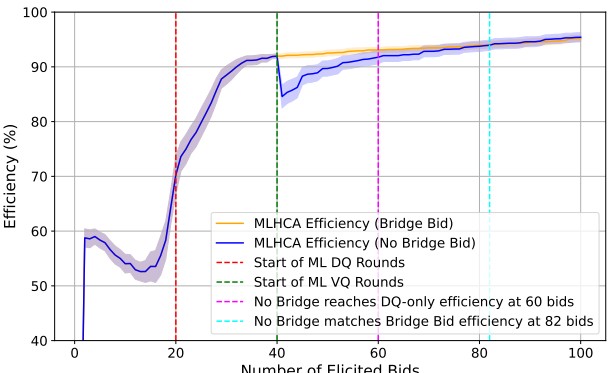

Figure 2: Efficiency of MLHCA with and without the bridge bid (Definition D.8) in the MRVM domain.

To demonstrate the effectiveness of the bridge bid, in Fig-

ure 2, we plot MLHCA's efficiency in MRVM–the most realistic domain–against the number of bids, comparing performance with and without the bridge bid. Without the bridge bid, MLHCA's efficiency drops by 7.3% points when it transitions to its VQ rounds. Notably, MLHCA requires 20 of our powerful ML-powered VQs just to recover the efficiency lost by the introduction of the first VQ. This is consistent with Lemma D.7, where we showed that efficiency can arbitrarily decrease when a VQ is introduced in a DQ-based auction. In contrast, the bridge bid completely mitigates this efficiency drop, as proven in Lemma D.9. In Appendix G.8, we provide a detailed analysis and explaining the bridge bid's efficacy relative to market competition.

Finally, in Appendix G.9, we experimentally evaluate the *Inverse* variant of MLHCA, which uses the inverse query order: it begins with VQs and then transitions to DQs. Across all tested domains, reversing the query order results in substantial efficiency losses, reaching up to 5 percentage points. In the inverse auction, ML-powered DQs fail to improve upon the efficiency achieved by the preceding VQs. Moreover, the early use of VQs alone cannot match the efficiency attained by the later-stage VQs in MLHCA, due to significantly weaker learning performance when the bidders' models have not been trained on both query types. These findings further reinforce our theoretical results on the critical role of query ordering in hybrid auctions.

# 7. Conclusion

We have introduced MLHCA, the first ICA to effectively combine both demand and value queries. By employing tailored query generation algorithms, incorporating the full information from both query types, and leveraging the theoretical insights developed in this work, MLHCA significantly outperforms current SOTA mechanisms across all tested domains and with significantly fewer queries. Notably, prior to MLHCA, the best-performing mechanism varied by domain, but MLHCA unifies the SOTA, delivering the best performance across all domains.

At first glance, it might seem obvious that combining DQs and VQs improves performance. However, one of the key insights of our work is that the *ordering of queries matters.* DQs provide broad but imprecise information across the entire space, while VQs offer targeted, precise information. As a result, DQs are more effective at the beginning of an auction, while VQs become advantageous once the auction's ML model has already been trained for a while. A second insight is that combining both query types requires careful handling. The efficiency of an auction using both DQs and VQs is non-monotone with respect to answered queries, as DQ responses establish lower bounds on bidders' valuations for queried bundles. Naively combining the two can lead to sharp efficiency drops, particularly in low-competition scenarios. However, by introducing a single, carefully-designed VQ, we can mitigate this effect and guarantee that the auction's efficiency does not fall below its DQ-only value.

A promising direction for future work is incorporating epistemic uncertainty into MLHCA to enhance efficiency. Another is developing an algorithm to dynamically determine the optimal switch to VQs, reducing cognitive load.

## Acknowledgments

We are grateful to Greg d'Eon, Bin Yu, Josef Teichmann, and Denise Künzli for helpful discussions and their support. This work was supported by the Swiss National Science Foundation (SNSF) Postdoc.Mobility fellowship [grant number P500PT_225356] and ETH Zürich.

## Impact Statement

This paper advances the field of iterative combinatorial auctions (ICAs) by introducing MLHCA, a novel machine learning-powered auction mechanism that achieves unprecedented efficiency while reducing bidders' cognitive load. The primary goal of this work is to enhance the practicality and efficiency of real-world auctions, such as those used in spectrum allocation, with large potential benefits for social welfare. By enabling more accessible and efficient auctions, MLHCA has the potential to positively impact market design, increasing participation and improving resource allocation across various domains.

The methods proposed rely on standard ML and optimization techniques, and we do not foresee immediate ethical concerns arising from their application. However, as in any setting involving self-interested agents, potential conflicts of interest between participants may arise. While such concerns are important in practice (as they are for any auction mechanism), addressing them lies outside the scope of this paper.

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

# A. Extended Preliminaries and Literature Review

## A.1. Extended Literature Review

In addition to the related work mentioned in Section 1, we also want to mention some further recent work an ML-based ICAs.

Estermann et al. (2023) use more diverse VQs for the initial VQs. They show that this diversity leads to higher efficiency than just asking initial VQs for i.i.d. uniformly random bundles. However, this does not solve the problem of it being cognitively very hard for bidders to answer these VQs that are not aligned with their preferences. Moreover, their efficiency results are outperformed by our MLHCA.

Maruo & Kashima (2024) uses multi-task learning to transfer to improve the generalization of the MVNNs by leveraging similarities among the value functions across bidders. This technique should also be compatible with our MLHCA. Thus, it would be an interesting direction for future work to incorporate multi-task learning into MLHCA and to evaluate how much this would improve efficiency. From a game theoretical perspective, one should think very carefully if multi-task learning could change the incentives of bidders. From a game-theoretical perspective, one would achieve incentive-alignment with social welfare, if each bidder $i$ cannot change the marginal efficiency of the economy $N \setminus \{i\}$ (see Appendix B.5). For MLCA, 3 out of 4 VQs actually query these marginal economies, such that $\mathcal{M}_i^\theta$ has no direct influence on these queries, which provides quite a strong game theoretical argument. Via multi-task learning, bidders have a more direct way to influence other bidders' models. While multi-task learning is a very promising direction to explore, one should be aware of potential game-theoretical risks imposed by multi-task learning.

Lubin et al. (2021) allow bidders to answer VQs with an interval over prices instead of an exact price. It would be interesting to combine this approach with MLHCA in future work.

Weissteiner (2023) and Heiss (2024, Section 4.4) provide a broader picture on ML-based ICAs.

Huang et al. (2025) explore how LLMs can be leveraged to create a new interaction paradigm for auctions, where the bidders interact with the mechanism by providing only natural language input.

Another related line of research is *mechanism design for LLMs*, where participants bid to effect the output of an ML model, specifically an LLM, e.g. Dütting et al. (2024); Soumalias et al. (2024a).

d'Eon et al. (2024); Almahdi et al. (2025) apply reinforcement learning algorithms to combinatorial auctions to better understand bidder strategies. Extending this line of work to mechanisms such as MLHCA would be an interesting direction for future research.

## A.2. A Machine Learning-Powered ICA

In this section, we present in detail the *machine learning-powered combinatorial auction (MLCA)* by Brero et al. (2021).

At the core of MLCA is a *query module* (Algorithm 2), which, for each bidder $i \in I \subseteq N$, determines a new value query $q_i$. First, in the *estimation step* (Line 1), an ML algorithm $\mathcal{A}_i$ is used to learn bidder $i$'s valuation from reports $R_i$. Next, in the *optimization step* (Line 2), an *ML-based WDP* is solved to find a candidate $q$ of value queries. In principle, any ML algorithm $\mathcal{A}_i$ that allows for solving the corresponding ML-based WDP in a fast way could be used. Finally, if $q_i$ has already been queried before (Line 4), another, more restricted ML-based WDP (Line 6) is solved and $q_i$ is updated correspondingly. This ensures that all final queries $q$ are new.

In Algorithm 3, we present MLCA. In the following, let $R_{-i} = (R_1, \ldots, R_{i-1}, R_{i+1}, \ldots, R_n)$. MLCA proceeds in rounds until a maximum number of queries per bidder $Q^{\max}$ is reached. In each round, it calls Algorithm 2 $(Q^{\text{round}} - 1)n + 1$ times: for each bidder $i \in N$, $Q^{\text{round}} - 1$ times excluding a different bidder $j \neq i$ (Lines 5–10, sampled *marginal economies*) and once including all bidders (Line 11, *main economy*). In total each bidder is queried $Q^{\text{round}}$ bundles per round in MLCA. At the end of each round, the mechanism receives reports $R^{\text{new}}$ from all bidders for the newly generated queries $q^{\text{new}}$ and updates the overall elicited reports $R$ (Lines 12–14). In Lines 16–17, MLCA computes an allocation $a_R^*$ that maximizes the *reported* social welfare (see Equation (3)) and determines VCG payments $p(R)$ based on the reported values $R$ (see Appendix Definition B.1).

---

**Algorithm 2:** NEXTQUERIES$(I, R)$ (Brero et al. 2021)

---

**Inputs :** Index set of bidders $I$ and reported values $R$
1 **foreach** $i \in I$ **do** Fit $\mathcal{A}_i$ on $R_i$: $\mathcal{A}_i[R_i]$ ▷ Estimation step
2 Solve $q \in \arg\max\limits_{a \in \mathcal{F}} \sum\limits_{i \in I} \mathcal{A}_i[R_i](a_i)$ ▷ Optimization step
3 **foreach** $i \in I$ **do**
4    **if** $(q_i, v_i(q_i)) \in R_i$ **then** ▷ Bundle already queried
5       Define $\mathcal{F}' = \{a \in \mathcal{F} : a_i \neq x, \forall(x, v_i(x)) \in R_i\}$
6       Re-solve $q' \in \arg\max_{a \in \mathcal{F}'} \sum_{l \in I} \mathcal{A}_l[R_l](a_l)$
7       Update $q_i = q'_i$
8 **return** *Profile of new queries* $q = (q_1, \ldots, q_n)$

---

**Algorithm 3:** MLCA$(Q^{\text{init}}, Q^{\text{max}}, Q^{\text{round}})$ (Brero et al. 2021)

---

**Params :** $Q^{\text{init}}, Q^{\text{max}}, Q^{\text{round}}$ initial, max and #queries/round
1 **foreach** $i \in N$ **do**
2    Receive reports $R_i$ for $Q^{\text{init}}$ randomly drawn bundles
3 **for** $k = 1, ..., \lfloor (Q^{max} - Q^{init})/Q^{round} \rfloor$ **do** ▷ Round iterator
4    **foreach** $i \in N$ **do** ▷ Marginal economy queries
5       Draw uniformly without replacement $(Q^{\text{round}}-1)$ bidders from $N \setminus \{i\}$ and store them in $\tilde{N}$
6       **foreach** $j \in \tilde{N}$ **do**
7          $q^{\text{new}} = q^{\text{new}} \cup$ NEXTQUERIES$(N \setminus \{j\}, R_{-j})$
8    $q^{\text{new}} = q^{\text{new}} \cup$ NEXTQUERIES$(N, R)$ ▷ Main economy queries
9    **foreach** $i \in N$ **do**
10       Receive reports $R_i^{\text{new}}$ for $q_i^{\text{new}}$, set $R_i = R_i \cup R_i^{\text{new}}$
11 Given elicited reports $R$ compute $a_R^*$ as in Equation (3)
12 Given elicited reports $R$ compute VCG-payments $p(R)$
13 **return** *Final allocation* $a_R^*$ *and payments* $p(R)$

---

## A.3. ML-Powered Demand Query Generation

In this section, we reprint the ML-powered demand query generation algorithm from Soumalias et al. (2024c). The critical notions behind the idea are those of indirect utility and revenue and clearing prices.

**Definition A.1** (Indirect Utility and Revenue)**.** For linear prices $p \in \mathbb{R}^m_{\geq 0}$, a bidder's indirect utility $U$ and the seller's indirect revenue $R$ are defined as

$$U(p, v_i) := \max_{x \in \mathcal{X}} \{v_i(x) - \langle p, x \rangle\} \text{ and} \tag{4}$$

$$R(p) := \max_{a \in \mathcal{F}} \left\{ \sum_{i \in N} \langle p, a_i \rangle \right\}^{6} = \sum_{j \in M} c_j p_j, \tag{5}$$

i.e., at prices $p$, Equations (4) and (5) are the maximum utility a bidder can achieve for all $x \in \mathcal{X}$ and the maximum revenue the seller can achieve among all feasible allocations.

**Definition A.2** (Clearing Prices)**.** Prices $p \in \mathbb{R}^m_{\geq 0}$ are *clearing prices* if there exists an allocation $a(p) \in \mathcal{F}$ such that

1. for each bidder $i$, the bundle $a_i(p)$ maximizes her utility, i.e., $v_i(a_i(p)) - \langle p, a_i(p) \rangle = U(p, v_i), \forall i \in N$, and

2. the allocation $a(p) \in \mathcal{F}$ maximizes the sellers revenue, i.e., $\sum_{i \in N} \langle p, a_i(p) \rangle = R(p)$.[6]

Theorem A.3 extends Bikhchandani & Ostroy (2002, Theorem 3.1), establishing a connection between the aforementioned definitions:

**Theorem A.3** (Soumalias et al. (2024c))**.** *Consider the notation from Definitions A.1 and A.2 and the objective function*

---

[6]For linear prices, this maximum is achieved by selling every item, i.e., $\forall j \in M : \sum_{i \in N}(a_i)_j = c_j$.

$W(p, v) := R(p) + \sum_{i \in N} U(p, v_i)$. *Then it holds that, if a linear clearing price vector exists, every price vector*

$$p' \in \underset{\tilde{p} \in \mathbb{R}^m_{\geq 0}}{\arg \min} \qquad\qquad W(\tilde{p}, v) \qquad\qquad (6a)$$

$$\text{such that} \qquad\qquad (x_i^*(\tilde{p}))_{i \in N} \in \mathcal{F} \qquad\qquad (6b)$$

*is a clearing price vector and the corresponding allocation* $a(p') \in \mathcal{F}$ *is* efficient.[7]

Theorem A.3 does not claim the existence of *linear clearing prices (LCPs)* $p \in \mathbb{R}^m_{\geq 0}$. For general value functions $v$, LCPs may not exist (Bikhchandani & Ostroy, 2002). However, in the case that LCPs do exist, Theorem A.3 shows that *all* minimizers of equation 6 are LCPs and their corresponding allocation is efficient. This is at the core of their ML-powered demand query generation algorithm.

Their key idea to generate ML-powered demand queries is the following: As an approximation for the true value function $v_i$, they use for each bidder a distinct mMVNN $\mathcal{M}_i^\theta : \mathcal{X} \to \mathbb{R}_{\geq 0}$ that has been trained on the bidder's elicited DQ data $R_i$. Motivated by Theorem A.3, they then try to find the DQ $p \in \mathbb{R}^m_{\geq 0}$ minimizing $W(p, (\mathcal{M}_i^\theta)_{i=1}^n)$ subject to the feasibility constraint equation 6b. This way, we find demand queries $p \in \mathbb{R}^m_{\geq 0}$ which, given the already observed demand responses $R$, have high clearing potential.

Note that equation 6 is a hard, bi-level optimization problem. Instead, Theorem A.4 allows them to minimize the problem via gradient descent:

**Theorem A.4** ((Soumalias et al., 2024c)). *Let* $(\mathcal{M}_i^\theta)_{i=1}^n$ *be a tuple of trained mMVNNs and let* $\hat{x}_i^*(p) \in \arg\max_{x \in \mathcal{X}} \{\mathcal{M}_i^\theta(x) - \langle p, x \rangle\}$ *denote each bidder's predicted utility maximizing bundle w.r.t.* $\mathcal{M}_i^\theta$. *Then it holds that* $p \mapsto W(p, (\mathcal{M}_i^\theta)_{i=1}^n)$ *is convex, Lipschitz-continuous and* a.e. *differentiable. Moreover,*

$$c - \sum_{i \in N} \hat{x}_i^*(p) \in \nabla_p^{\text{sub}} W(p, (\mathcal{M}_i^\theta)_{i=1}^n) \qquad\qquad (7)$$

*is always a sub-gradient and a.e. a classical gradient.*

With Theorem A.4, we obtain the following update rule of classical GD $p_j^{\text{new}} \overset{a.e.}{=} p_j - \gamma(c_j - \sum_{i \in N}(\hat{x}_i^*(p))_j), \forall j \in M$. Interestingly, this equation has an intuitive economic interpretation. If the $j^{\text{th}}$ item is over/under-demanded based on the predicted utility-maximizing bundles $\hat{x}_i^*(p)$, then its new price $p_j^{\text{new}}$ is increased/decreased by the learning rate times its over/under-demand. To enforce constraint equation 6b in GD, they asymmetrically increase the prices $1 + \mu \in \mathbb{R}_{\geq 0}$ times more in case of over-demand than they decrease them in case of under-demand. This leads to the final update rule:

$$p_j^{\text{new}} \overset{a.e.}{=} p_j - \tilde{\gamma}_j(c_j - \sum_{i \in N}(\hat{x}_i^*(p))_j), \forall j \in M, \qquad\qquad (8a)$$

$$\tilde{\gamma}_j := \begin{cases} \gamma \cdot (1 + \mu) & , c_j < \sum_{i \in N}(\hat{x}_i^*(p))_j \\ \gamma & , \text{else} \end{cases} \qquad\qquad (8b)$$

## B. Payment and Activity Rules

In this section, we reprint the VCG and VCG-nearest payment rules, as well as give an overview of activity rules for the CCA, and argue why the most prominent choices are also applicable to our MLHCA. Finally, we show how MLHCA can immediately detect if a bidder's reports are inconsistent with any valuation function.

### B.1. VCG Payments

**Definition B.1.** (VCG PAYMENTS FROM DEMAND AND VALUE QUERY DATA) Let $R = (R_1, \ldots, R_n)$ denote an elicited set of both demand and value query data from each bidder and let $R_{-i} := (R_1, \ldots, R_{i-1}, R_{i+1}, \ldots, R_n)$. We then calculate

---

[7]More precisely, constraint equation 6b should be reformulated as

$$\exists (x_i^*(\tilde{p}))_{i \in N} \in \underset{i \in N}{\bigtimes} \mathcal{X}_i^*(\tilde{p}) : (x_i^*(\tilde{p}))_{i \in N} \in \mathcal{F},$$

where $\mathcal{X}_i^*(\tilde{p}) := \arg\max_{x \in \mathcal{X}} \{v_i(x) - \langle \tilde{p}, x \rangle\}$, since in theory, $x_i^*(\tilde{p})$ does not always have to be unique.

the VCG payments $\pi^{\text{VCG}}(R) = (\pi_1^{\text{VCG}}(R)\ldots, \pi_n^{\text{VCG}}(R)) \in \mathbb{R}_{\geq 0}^n$ as follows:

$$\pi_i^{\text{VCG}}(R) := \sum_{j \in N \setminus \{i\}} \widetilde{v}_j\left(a^*(R_{-i})_j; R_j\right) - \sum_{j \in N \setminus \{i\}} \widetilde{v}_j\left(a^*(R)_j; R_j\right). \tag{9}$$

where $a^*(R_{-i})$ is the allocation that maximizes the inferred social welfare when excluding bidder $i$, i.e.,

$$a^*(R_{-i}) \in \arg\max_{a \in \mathcal{F}} \sum_{j \in N \setminus \{i\}} \widetilde{v}_j(a_j; R_j), \tag{10}$$

and $a^*(R)$ is the inferred social welfare-maximizing allocation (see Equation (3)).

Thus, when using VCG payments, bidder $i$'s utility is:

$$\begin{aligned}
u_i &= v_i(a^*(R)_i) - \pi_i^{\text{VCG}}(R) \\
&= v_i(a^*(R)_i) + \sum_{j \in N \setminus \{i\}} \widetilde{v}_j\left(a^*(R)_j; R_j\right) \\
&\quad - \sum_{j \in N \setminus \{i\}} \widetilde{v}_j\left(a^*(R_{-i})_j; R_j\right).
\end{aligned}$$

## B.2. VCG-Nearest Payments

To define the VCG-nearest payments, we must first introduce the core:

**Definition B.2.** (THE CORE) An outcome $(a, \pi) \in \mathcal{F} \times \mathbb{R}_{\geq 0}^n$ (i.e., a tuple of a feasible allocation $a$ and payments $\pi$) is in the core if it satisfies the following two properties:

1. The outcome is *individual rational*, i.e, $u_i = v_i(a_i) - \pi_i \geq 0$ for all $i \in N$

2. The core constraints

$$\forall\, L \subseteq N \quad \sum_{i \in N \setminus L} \pi_i(R) \geq \max_{a' \in \mathcal{F}} \sum_{i \in L} v_i(a_i') - \sum_{i \in L} v_i(a_i) \tag{11}$$

where $v_i(a_i)$ is bidder $i$'s value for bundle $a_i$ and $\mathcal{F}$ is the set of feasible allocations.

In words, a payment vector $\pi$ (together with a feasible allocation $a$) is in the core if no coalition of bidders $L \subset N$ is willing to pay more for the items than the mechanism is charging the winners. Note that by replacing the true values $v_i(a_i)$ with the bidders' (possibly untruthful) *inferred values* based on their reports $\widetilde{v}_i(a_i; R_i)$ in Definition B.2 one can equivalently define the *revealed core*.

Now, we can define

**Definition B.3.** (MINIMUM REVENUE CORE) Among all payment vectors in the (revealed) core, the (revealed) minimum revenue core is the set of payment vectors with smallest $L_1$-norm, i.e., which minimize the sum of the payments of all bidders.

We can now define VCG-nearest payments:

**Definition B.4.** (VCG-NEAREST PAYMENTS) Given an allocation $a_R$ for bidder reports $R$, the VCG-nearest payments $\pi^{\text{VCG-nearest}}(R)$ are defined as the vector of payments in the (revealed) minimum revenue core that minimizes the $L_2$-norm to the VCG payment vector $\pi^{\text{VCG}}(R)$.

## B.3. On the Importance of Activity Rules to Align Incentives

In the CCA, activity rules serve multiple purposes. First, they help accelerate the auction process. Second, they reduce "bid-sniping" opportunities—bidders concealing their true intentions until the very last rounds of the auction.[8] Third, they limit surprise bids in the supplementary round of the CCA, significantly reducing a bidder's ability to drive up opponents' payments by overbidding on bundles they cannot win (Ausubel & Baranov, 2017). There are two types of activity rules that are implemented in a CCA:

---

[8]The notion of "bid-sniping" originated in eBay auctions with predetermined ending times, where high-value bidders could reduce their payments by submitting bids at the very last moment.

1. *Clock phase activity rules*, which limit the bundles that an agent can bid on during the clock phase, based on their bids in previous clock rounds.

2. *Supplementary round activity rules*, which restrict the amounts that an agent can bid on specific sets of items during the supplementary round.

Traditionally, most clock phase activity rules in the CCA have relied on either revealed-preference principles or a *points-based system*, where points are assigned to each item before the auction, and bidders are only allowed to submit monotonically non-increasing bids in terms of points. In other words, as prices rise across rounds, bidders cannot submit bids for larger sets of items. Both of these approaches, as well as hybrid combinations thereof, were shown to actually further interfere with truthful bidding in some cases (Ausubel & Baranov, 2014; 2020).

However, Ausubel & Baranov (2019) showed that basing clock phase activity rules entirely on the *generalized axiom of revealed preference (GARP)* can dynamically approximate VCG payoffs, thus improving the bidding incentives of the CCA. GARP imposes revealed-preference constraints (see Definition B.5) on the bidder's demand responses. The GARP activity rule requires that the bidder demonstrates rational behavior in her demand choices, without necessitating a monotonic price trajectory. As a result, it can also be applied during the ML-powered DQ phase of MLHCA, allowing our mechanism to enjoy similar improvements in bidding incentives.

For the supplementary round, the CCA's most prominent activity rules are again based on a combination of points-based systems and revealed-preference ideas, which we outline below:

**Definition B.5.** (REVEALED-PREFERENCE CONSTRAINT) The revealed-preference constraint for bundle $x \in X$ with respect to clock round $r$ is

$$b_i(x) \leq b_i(x^r) + \langle p^r, x - x^r \rangle , \tag{12}$$

where $b_i(x) \in \mathbb{R}_{\geq 0}$ is bidder $i$'s bid for bundle $x \in \mathcal{X}$ in the supplementary round, $x^r \in \mathcal{X}$ is the bundle demanded by the agent at clock round $r$, $b_i(x^r) \in \mathbb{R}_{\geq 0}$ is the final bid for bundle $x^r \in \mathcal{X}$ and $p^r \in \mathbb{R}_{\geq 0}^m$ is the linear price vector of clock round $r$.

Intuitively, the revealed-preference constraint ensures that a bidder cannot claim a higher value for bundle $x$ relative to bundle $x^r$, given that they expressed a preference for bundle $x^r$ at the given prices $p^r$ (see Equation (1)). The difference between the three most prominent supplementary round activity rules is with respect to *which clock rounds* the revealed-preference constraint should be satisfied. Specifically:

1. *Final Cap:* A bid for bundle $x \in \mathcal{X}$ should satisfy the *revealed-preference constraint (Definition B.5)* with respect to the *final* clock round's price $p^{Q^{\mathrm{CCA}}} \in \mathbb{R}_{\geq 0}$ and bundle $x^{Q^{\mathrm{CCA}}} \in \mathcal{X}$.

2. *Relative Cap:* A bid for bundle $x \in \mathcal{X}$ should satisfy the *revealed-preference constraint (Definition B.5)* with respect to the last clock round for which the bidder was eligible for that bundle $x \in \mathcal{X}$, based on the points-based system.

3. *Intermediate Cap:* A bid for bundle $x \in \mathcal{X}$ should satisfy the *revealed-preference constraint (Definition B.5)* with respect to all eligibility-reducing rounds, starting from the last clock round for which the bidder was eligible for $x \in \mathcal{X}$ based on the point system.

Ausubel & Baranov (2017) showed that combining the *Final Cap* and *Relative Cap* activity rules leads to the largest amount of reduction in bid-sniping opportunities for the UK 4G auction, as measured by the theoretical bid amount that each bidder would need to increase her bid by in the supplementary round in order to protect her final clock round bundle. Finally, note that the *Final-* and *Intermediate Cap* activity rules can also be applied to the ML-powered DQ phase of our MLHCA.[9]

To conclude, both the DQ and VQ phases of MLHCA are compatible with the most prominent activity rules of the CCA, and MLHCA also remains compatible with the commonly used VCG-nearest pricing rule (Definition B.4). Combined with MLHCA's similar interaction paradigm to the CCA, these aspects provide strong evidence that our mechanism can leverage activity rules to effectively mitigate bidder misreporting opportunities, much like the classical CCA.

---

[9]Soumalias et al. (2024c) argued that with the modification for the *Relative Cap* rule that the revealed-preference constraint should hold for the $Q^{\mathrm{CCA}}$ rounds that follow the same price update rule as the CCA, and then the ML-powered clock rounds should be treated as corresponding to the same amount of points, since the prices in these rounds on aggregate stay very close to the prices of the last $Q^{\mathrm{init}}$ round.

### B.4. MLHCA Can Detect Inconsistent Misreports

In the following lemma, we formally prove that if a bidder's reports are inconsistent with any valuation function, then the training loss for that bidder's network will be strictly positive, thus MLHCA can detect such misreports.

**Lemma B.6** (Strictly Positive Loss from an Inconsistent Datapoint). *Let $R = (R^{DQ}, R^{VQ})$ be a set of elicited reports by a bidder that is rationalizable by some monotone valuation function $v_0 : \mathcal{X} \to \mathbb{R}_{\geq 0}$. Suppose, that during the MLHCA auction (Algorithm 1), the bidder responds to the next query, either a DQ $(\widetilde{x}^*(p^{\widetilde{r}}), p^{\widetilde{r}})$ or a VQ $(\widetilde{x}, \widetilde{v}(\widetilde{x}))$ and assume that no monotone valuation $v$ can simultaneously rationalize all of her responses $\mathcal{R}'$. Then, when using Algorithm 4 (with any regression loss $F$ for the VQ responses that satisfies $F \geq 0$ and $y = \widetilde{y} \iff F(y, \widetilde{y}) = 0$) to fit an MVNN $\mathcal{M}^\theta$ to $\mathcal{R}'$, we have $\min_\theta L(\theta) > 0$.*

*Proof.* We prove the claim in cases. Case 1: Suppose that the bidder misreports in a way that is non-rationalizable by any valuation function during the DQ phase of the auction. In that phase, the bidder's set of reports consists only of demand queries.

For each datapoint $(x^*(p^r), p^r)$ in $R^{DQ}$, Algorithm 4 attempts to make

$$\hat{x}^*(p^r) \in \arg\max_{x \in \mathcal{X}}\Big[ \mathcal{M}^\theta(x) - \langle p^r, \ x \rangle \Big]$$

match the reported $x^*(p^r)$. If it does *not* match, the loss is incremented by a nonnegative amount:

$$\Delta L_r(\theta) = \big[ \mathcal{M}^\theta(\hat{x}^*(p^r)) - \langle p^r, \hat{x}^*(p^r) \rangle \big] - \big[ \mathcal{M}^\theta(x^*(p^r)) - \langle p^r, x^*(p^r) \rangle \big] \geq 0.$$

Hence the total loss $L(\theta)$ is always weakly positive.

Suppose, for contradiction, that there exists $\theta$ with $L(\theta) = 0$. If $L(\theta) = 0$, it means the predicted best response matches the reported one, i.e., $\hat{x}^*(p^r) = x^*(p^r)$ for all $r$, including $r = \tilde{r}$.

However, for any $\theta \in \Theta$, the (m)MVNN $\mathcal{M}^\theta$ is by construction a valid valuation function satisfying free disposal (Weissteiner et al., 2022a; Soumalias et al., 2024c). The condition $\hat{x}^*(p^r) = x^*(p^r)$ for all $r$ means precisely that $\mathcal{M}^\theta$ *rationalizes all* data in $\mathcal{D}'$. Thus, there exists a valuation function rationalizing all data points, including $\widetilde{x}^*(p^r)$, a contradiction.

Case 2: Suppose that the bidder misreports in a way that is non-rationalizable by any valuation function during the VQ phase of the auction. Similarly, given that the loss function in each datapoint (both DQs and VQs) is weakly positive, the only way the loss can be zero is if it is zero on every point. But then, the MVNN once again has rationalized the agent's reports. Thus, a value function exists that rationalizes all of the agent's reports, a contradiction.

$\square$

Note that Lemma B.6 can also be applied to the case where we observe 0 VQs. Thus, Lemma B.6 can also be applied to detect inconsistent misreporting for DQ-only auctions such as ML-CCA.

Further note that Lemma B.6 can always detect inconsistent misreporting, while other forms of misreporting cannot be detected this way.

### B.5. On the Importance of Marginal Economies to Align Incentives

In this section, we review the key arguments from Brero et al. (2021) on why MLCA provides strong incentives for truthful reporting in practice. These arguments extend to any ML-powered ICA that employs the same VQ-generation algorithm, including MLHCA.

Bidder $i$'s utility in MLCA (and MLHCA) under VCG payments (see Definition B.1) can be expressed as:

$$u_i = v_i(a^*(R)_i) - \pi_i^{\text{VCG}}(R)$$
$$= v_i(a^*(R)_i) + \underbrace{\sum_{j \in N \setminus \{i\}} \widetilde{v}_j\left(a^*(R)_j; R_j\right)}_{(a)} - \underbrace{\sum_{j \in N \setminus \{i\}} \widetilde{v}_j\left(a^*(R_{-i})_j; R_j\right)}_{(b) \text{ Inferred SW of marginal economy}} \ .$$

Any beneficial misreport by bidder $i$ must increase the difference (a) $-$ (b).

MLCA has two features that mitigate manipulations. First, MLCA explicitly queries each bidder's marginal economy (Algorithm 3, Line 5), which implies that (b) is practically independent of bidder $i$'s reports. Experimental evidence supporting this claim is provided in Section 7.3 of Brero et al. (2021). Second, MLCA (and also MLHCA) enables bidders to "push" information to the auction which they deem useful. This mitigates certain manipulations that target (a), as it allows bidders to increase (a) with truthful information. Brero et al. (2021) argue that any remaining manipulation would be implausible as it would require almost complete information.

Under further assumptions, we can also derive two theoretical incentive guarantees:

- Assumption 1 requires that, for all bidders $i \in N$, if all other bidders report truthfully, then the reported social welfare of bidder $i$'s marginal economy (i.e., term (b)) is *independent* of her value reports.

- Assumption 2 requires that, if all bidders $i \in N$ bid truthfully, then MLCA *finds an efficient allocation*.

**Result 1: Social Welfare Alignment**    Under Assumption 1, and given that all other bidders are truthful, MLCA is *social welfare aligned*. This means that the only way for a bidder to increase her true utility is by increasing the reported social welfare of $a^*(R)$ in the main economy (i.e., term (a)), which, in this case, equals the true social welfare of $a^*(R)$ (Brero et al., 2021, Proposition 3). The same is true for the VQ phase of MLHCA, as it employs the same allocation and payment rules.

**Result 2: Ex-Post Nash Equilibrium**    If both Assumption 1 and Assumption 2 hold, then bidding truthfully constitutes an ex-post Nash equilibrium in MLCA (Brero et al., 2021, Proposition 4). The same is true for the VQ phase of MLHCA, as it employs the same allocation and payment rules.

*Remark* B.7 (Experimental Evaluation of Assumption 2).  The results shown in Tables 2 and 9 suggest that Assumption 2 is more realistic for MLHCA than for any other mechanism. For GSVM, Assumption 2 is absolutely realistic for MLHCA and was already realistic for other VQ-based mechanisms such as the ones proposed by (Weissteiner & Seuken, 2020; Weissteiner et al., 2022a; 2023). Also for SRVM, Assumption 2 is very realistic for MLHCA. In fact, MLHCA is the first method from Table 2 that always found an efficient allocation (only methods from Table 9 that use significantly more than 200 can keep up with this). Theoretically achieving 100% efficiency in all 50 random instances of an auction does not suffice as mathematical proof that the auction will always achieve 100% efficiency. However, for GSVM and SRVM, the fact that MLHCA found an efficient allocation within the first 60 out of 100 queries for all 50 instances, strongly suggests that 100 queries allow MLHCA to find an efficient allocation with almost 100% probability. For LSVM, MLHCA found an efficient allocation in 49 out of 50 auction instances, which from a practical point of view also almost satisfies Assumption 2, and with a few queries more fully satisfying Assumption 2 might be in reach. At least for every domain, MLHCA is closer to satisfying Assumption 2 than its competitors.

To conclude, MLHCA's compatibility with both *activity rules* during its DQ rounds and *marginal economies* during its VQ rounds, as well as its compatibility with VCG and VCG-nearest payments, provides strong evidence that MLHCA can effectively mitigate opportunities for bidder misreporting.

## C. MVNN

The original definition (Weissteiner et al., 2022a) is a special case of the more general definition (Soumalias et al., 2024c) that we state here.

**Definition C.1** (MVNN). An MVNN $\mathcal{M}_i^\theta : \mathcal{X} \to \mathbb{R}_{\geq 0}$ for bidder $i \in N$ is defined as

$$\mathcal{M}_i^\theta(x) \coloneqq W^{i,K_i} \varphi_{0,t^{i,K_i-1}} \left( \ldots \varphi_{0,t^{i,1}} (W^{i,1} (Dx) + b^{i,1}) \ldots \right) \tag{13}$$

- $K_i + 2 \in \mathbb{N}$ is the number of layers ($K_i$ hidden layers),
- $\{\varphi_{0,t^{i,k}}\}_{k=1}^{K_i-1}$ are the MVNN-specific activation functions with cutoff $t^{i,k} > 0$, called *bounded ReLU (bReLU)*:

$$\varphi_{0,t^{i,k}}(\cdot) \coloneqq \min(t^{i,k}, \max(0, \cdot)) \tag{14}$$

- $W^i \coloneqq (W^{i,k})_{k=1}^{K_i}$ with $W^{i,k} \geq 0$ and $b^i \coloneqq (b^{i,k})_{k=1}^{K_i-1}$ with $b^{i,k} \leq 0$ are the *non-negative* weights and *non-positive* biases of dimensions $d^{i,k} \times d^{i,k-1}$ and $d^{i,k}$, whose parameters are stored in $\theta = (W^i, b^i)$.
- $D \coloneqq \mathrm{diag}\left(1/c_1, \ldots, 1/c_m\right)$ is the linear normalization layer that ensures $Dx \in [0,1]$ and is not trainable.

*Remark C.2.* The index $i$ of the MVNN $\mathcal{M}_i^\theta(x)$ emphasizes that we train an individual MVNN for every bidder $i$ to approximate $v_i$. In the following, we sometimes omit the index $i$ if we just want to make general arguments about the MVNN architecture without.

*Remark C.3 (Linear Skip Connection).* Sometimes we also use linear skip connections as introduced in Weissteiner et al. (2023, Definition F.1)

*Remark C.4 (Initiaization).* We always use the initialization scheme from Weissteiner et al. (2023, Section 3.2 and Appendix E), which offers crucial advantages over standard initialization schemes as discussed in Weissteiner et al. (2023, Section 3.2 and Appendix E).

### C.1. On the Inductive Bias of MVNNs

Weissteiner et al. (2022a); Soumalias et al. (2024c) have shown that MVNNs can represent any monotonic normalized function on $\mathcal{X}$. However, for finitely many data points, multiple different monotonic functions can fit the data equally well, but the training algorithm will choose only one of these functions. We want to understand according to which preferences the algorithm makes this choice, i.e., we want to understand its inductive bias.

For certain ReLU-NNs it has been shown that L2-regularization (also known as "weight decay") of the parameters $\theta$ corresponds to regularizing a Lp-norm of the second derivative of the function (Heiss et al., 2019; 2023; 2021; Heiss, 2024; Savarese et al., 2019; Ongie et al., 2019; Williams et al., 2019; Parhi & Nowak, 2022). Since the second derivative of linear functions is zero, these NNs prefer linear functions.

However, MVNNs use a different activation function (Weissteiner et al., 2022a). For MVNNs, no theoretical result about their second derivative has been proven so far. It is quite clear that the L2-regularization of the parameters of a MVNN does not exactly correspond to any Lp-norm of the second derivative. Weissteiner et al. (2023) modified the MVNN architecture by adding so-called linear skip connections (Weissteiner et al., 2023, Definition F.1) to obtain an inductive bias towards linear functions. If one uses unregularized linear skip connections but regularizes all other parameters, it is quite obvious that the optimal parameters will only have non-zero weights in the linear skip connections if a monotonic linear function can perfectly explain the data.[10]

In the setting of Example 1 (which is based on the example in the proof of Theorem 3.2) one can also prove that MVNNs with arbitrarily small L2-regularization, would always choose a function that is linear on $\mathcal{X}$ given any possible truthful DQ responses from bidder 2, even without linear skip connections.

**Proposition C.5.** *As in Example 1, let $n = 2$, $m = 1$, $c_1 = 10$ and $v_2$ such that whenever bidder 2 is queried a DQ she answers in the following way:*

- *If the price $p$ is below $\frac{94}{10}$, bidder 2 will answer with $x_2^*(p) = (10)$;*

---

[10]If the data can be perfectly explained by a linear function, then only using the linear skip connections can achieve zero training loss and zero regularization costs, while setting any parameter outside the linear skip connections to any non-zero value would lead to non-zero L2 regularization costs.

- *if the price $p = \frac{94}{10}$, bidder 2 will answer with either $x_2^*(p) = (10)$ or $x_2^*(p) = (0)$;*

- *if the price $p$ is higher than $\frac{94}{10}$, bidder 2 will answer with $x_2^*(p) = (0)$.*

*Let $\{p^1, \ldots, p^{n_{DQ}^{train}}\} \subset [0, \infty)$ be the subset of prices bidder 2 is queried. Let $\theta^*$ be any (local) minimizer of the L2-regularized loss from ([Soumalias et al., 2024c])*

$$L^\lambda(\theta) := \sum_{r=1}^{n_{DQ}^{train}} \left( \mathcal{M}_\theta(\hat{x}_2^*(p^r)) - \langle p^r, \hat{x}_2^*(p^r) \rangle - \left( \mathcal{M}_\theta(x_2^*(p^r)) - \langle p^r, x_2^*(p^r) \rangle \right) \right)^+ + \lambda \|\theta\|_2^2,$$

*where $\hat{x}_2^*(p^r) := \arg\min_{x \in \mathcal{X}} \left( \mathcal{M}_\theta(x) - \langle p^r, x \rangle \right)$. Then the MVNN $\mathcal{M}_{\theta^*} : \mathcal{X} \to \mathbb{R}$ is linear.*

*Proof.* We define $\tilde{p} := \max \left\{ p^r : x^*(r) = (10), 1 \leq r \leq n_{DQ}^{train} \right\}$.[11]

1. First we show that $\mathcal{M}_{\theta^*}(10) \leq 10\tilde{p}$ via a contraposition argument. Let's assume $\mathcal{M}_{\theta^*}(10) \geq 10q > 10\tilde{p}$, then multiplying the last layer's weights by $1 - \delta > \frac{q}{p}$ would both reduce the data-loss-term $L^0$ (since the activation the hidden layers of MVNNs are always non-negative) and the regularization costs $\lambda \|\cdot\|_2^2$. Therefore, no local minima $\theta^*$ can satisfy $\mathcal{M}_{\theta^*}(10) > 10\tilde{p}$. Thus, we have shown that $\mathcal{M}_{\theta^*}(10) \leq 10\tilde{p}$ holds for any local minima $\theta^*$.

2. Next, we show that all pre-activations of our $\mathcal{M}_{\theta^*}$ are smaller or equal to the cut-off of the corresponding bReLU activation function for any input $x \in \mathcal{X}$. Let's assume again the contraposition that at least one pre-activation is larger than the cut-off. In this case, we can scale down all the incoming weights of such a neuron without changing $\mathcal{M}_{\theta^*}(10)$. Scaling down these weights cannot increase the value of $\mathcal{M}_{\theta^*}(x)$ for any $x \in \mathcal{X}$, so it cannot increase the data-loss term $L^0$, but scaling down weights obviously decreased the regularization costs. Thus, via this counterposition argument, we have proven that all the pre-activations are smaller or equal to the cut-off for any local minima $\theta^*$.

3. Next, we show that all biases of $\theta^*$ are zero. First, note that by Item 2, we know that $\mathcal{M}_{\theta^*}$ is convex (since the bReLU is convex below the cut-off). By combining this fact with Item 1, we obtain that $\mathcal{M}_{\theta^*}(x) \leq x\tilde{p}$, since MVNNs always satisfy $\mathcal{M}_\theta(0) = 0$. Let's assume the counterposition of at least one bias being strictly negative (as by definition, biases can never be positive for MVNNs). Then we could increase the bias a little bit without increasing the data-loss-term $L^0$,[12] but increasing the bias reduces its regularization cost. Thus any local minima $\theta^*$ satisfies that the biases are zero.

By combining Items 2 and 3 we obtain that $\mathcal{M}_{\theta^*}$ is linear. $\qquad\square$

*Remark* C.6 (Interpretation of the Inductive Bias of MVNNs). It is important to keep in mind that MVNNs can learn complicated non-linear monotonic functions, *if* the training data requires it. For example, if we receive the 2 VQs, $v(20) = 20\$$ and $v(10) = 19\$$, there is no linear function that can explain both VQs simultaneously, but an MVNN can easily learn a non-linear monotonic function which perfectly fits both VQs simultaneously, i.e., $\mathcal{M}_{\theta^*}(20) = 20\$$ and $\mathcal{M}_{\theta^*}(10) = 19\$$. Therefore, the ability of MVNNs to learn non-linear monotonic functions is important, since we don't want the MVNN to predict $\mathcal{M}_{\theta^*}(10) = 10\$$, if we know already $v(10) = 19\$$. However in the case that we don't know $v(10) = 19\$$, but only observe 1 VQ, $v(20) = 20\$$, then this section provides intuition to understand that the MVNN would typically predict $\mathcal{M}_{\theta^*}(10) = 10\$$. So the goal of this section is to better understand how MVNNs deal with insufficient information.

---

[11]If $\left\{ p^r : x^*(r) = (10), 1 \leq r \leq n_{DQ}^{train} \right\}$ is empty, we define $\tilde{p} := 0$. In Example 1, $\tilde{p} = \frac{94 - \epsilon}{10}$.

[12]This argument relies on the fact that we only queried finitely many DQs. If we asked infinitely many DQs that are dense around $p = \frac{94}{10}$, one would need to modify the argument by not only increasing the biases but simultaneously also decreasing certain weights.

# D. Details on Section 3

In this section, we examine the limitations of using only VQs or only DQs in auctions and highlight the benefits of combining them.

## D.1. Disadvantages of Only Using VQs

Almost all ML-powered VQ-based auctions including the current SOTA, BOCA (Weissteiner et al., 2023) first ask each bidder multiple random VQs (i.e., VQs for randomly selected bundles). These VQs are necessary to initialize the ML estimates of the bidder's value functions. In practice, it is very hard for bidders to answer random VQs since they are not aligned with their preferences.[13] The most popular ICAs in practice (e.g., the CCA) ask the bidders DQs, which have been argued can be answered by the bidders sufficiently well (Cramton, 2013).[14]

Even if bidders manage to respond perfectly to random VQs, the information obtained is limited. This is because, in large combinatorial domains, bidders typically have high values for only a small subset of possible bundles, making the probability of querying one of these high-value bundles at random exceedingly low. On the other hand, querying bidders with DQs at a random price vector is more likely to prompt responses that reveal their high-value bundles. This is formalized in Proposition 3.1, which we reprint for convenience:

**Proposition D.1** (Restatement of Proposition 3.1). *The expected social welfare of an auction that uses a single random demand query can be arbitrarily larger than that of an auction that uses any constant number ($k \ll 2^m$) of random value queries.*

*Proposition 3.1 Proof.* Let $n = 2$ and $c_1 = c_2 = \cdots = c_m = 1$, i.e., the auction has $m$ unique items. Bidder 1 has a value of zero for the empty set and a value of $\epsilon > 0$ for any non-empty set of items, while bidder 2 has a value of $V \to \infty$ for the full bundle, and a value of zero for any other bundle. Note that these are proper value functions, as they are both monotone and assign a value of zero to the empty set. The bundle space $\mathcal{X}$ has a size of $2^m$. For the auction that asks random value queries, the probability that bidder 2 is queried her value for the full bundle conditioned on not having been asked that question in the previous $k$ queries is $\frac{1}{2^m - k}$. For auction instances with large numbers of items, taking $m \to \infty$, the probability of the auction not querying bidder 2 her value for the full bundle in $k$ random value queries is:

$$\lim_{m \to \infty} \prod_{j=1}^{k} \left( 1 - \frac{1}{2^m - (j-1))} \right) = \lim_{m \to \infty} \prod_{j=1}^{k} \left( \frac{2^m - j}{2^m - (j-1))} \right) = 1 \tag{15}$$

If that query for the full bundle is asked to bidder 2, then bidder 2 will be allocated the full bundle and bidder 1 will be allocated the empty bundle, and the social welfare of the final allocation will be equal to $V$. In any other case, bidder 1 will be allocated a non-empty bundle, and the social welfare of the allocation will be equal to $\epsilon$.

Now let's focus on the auction that asks each bidder a single random demand query, and assume that the price of each item is an i.i.d. random variable with mean value $p$. The expected total price for the full bundle is $m \cdot p$. Applying Chebyshev's inequality, the probability that the price of the full bundle is greater than $V$ is zero. Thus, bidder 2 will always request the full bundle.

The only possible scenario in which bidder 2 is not allocated the full bundle is if bidder 1 requests a non-empty bundle with equal (or higher) value than the bundle of bidder 2, and ties are broken in bidder 1's favor. Given that bidder 1's value is at most $\epsilon$ for any non-empty bundle, with probability 1, the value of the bundle requested by bidder 1 is at most $\epsilon$. Given that the expected value of the price of each item is $p > 0$, applying Chebyshev's inequality yields that as $m \to \infty$, the probability of the full bundle having a price less than $\epsilon$ is zero. Thus, with probability 1 bidder 1 will have an inferred value less than bidder 2 for the bundle she requested, and so with probability 1 the full bundle will be allocated to bidder 2, yielding a social welfare of $V \to \infty$ for this auction. This completes the proof. $\square$

*Remark* D.2. This limitation of random VQs is evidenced in practice. Empirical comparisons between VQ-based ML-powered mechanisms, such as Weissteiner et al. (2023), and DQ-based mechanisms, such as (Soumalias et al., 2024c), reveal

---

[13] To provide some intuition, imagine you go to the supermarket because you want to bake a birthday cake for your friend and then you are asked your value for 30 frying pans plus 500 coconuts. It might be hard to estimate your value for such a random combination of items.

[14] In our practical supermarket example, now imagine that you view the price tags for the same items. It is quite doable to decide which items you want to buy and in which quantities.

that efficiency after initial queries is significantly lower for VQ-based approaches across all tested domains (see Figure 1 in Section 6).

Beyond auction efficiency, the limited information provided by random VQs poses challenges for learning algorithms in ML-powered ICAS. In contrast, DQs provide global information about bidder preferences across the entire bundle space. When bidder $i$ responds to a DQ at prices $p$, she solves the optimization problem: $x_i^*(p) \in \arg\max_{x \in \mathcal{X}} \{v_i(x) - \langle p, x \rangle\}$, which reveals valuable information about her preferences across all possible bundles. Strong evidence for this is presented in Appendix E.2, where we show that the network trained only on DQs exhibits better generalization performance than one trained on random VQs.

Additionally, if DQ prices are sufficiently low, bidders respond with their value-maximizing bundles, which may be hard to recover through VQs alone. By incorporating this information, the learning algorithm can more effectively identify critical regions in the allocation space and subsequently focus on refining those areas. This advantage is further supported by our experiments (Figure 1 in Section 6). We show that in our ML-powered hybrid auction, the first ML-powered VQ after a series of DQs achieves significantly higher efficiency compared to the first ML-powered VQ after an equivalent number of random VQs in the current SOTA VQ-based auction.

Moreover, even if the auction finds an efficient allocation by using VQs, it cannot terminate early as there is no way for the auctioneer to certify that the auction has reached 100% efficiency. In contrast, for DQ-based auctions there is an easy condition that allows the auction to terminate early:

**Proposition D.3.** *If clearing prices exist, an auction using DQs can provide a guarantee of optimal efficiency and terminate early.*

*Proof.* If clearing prices have been found, the corresponding allocation constitutes a *Walrasian equilibrium*, and thus has an efficiency equal to 100%. See Soumalias et al. (2024c, Appendix C.1) for a detailed proof. $\square$

*Remark* D.4. This is indeed an issue in practice. In Section 6, we experimentally show that, in realistic domains, our MLHCA can often reach 100% efficiency before the common maximum number of 100 rounds used by most ML-powered ICAs (e.g. Weissteiner & Seuken (2020); Weissteiner et al. (2022b;a; 2023); Soumalias et al. (2024c)) is reached.

### D.2. Disadvantages of Only Using DQs

In this section, we show the disadvantages of using DQs to elicit the bidders' preferences.

The first major disadvantage of an auction employing only DQs is that the auction's efficiency can actually drop by adding more DQs.

**Proposition D.5** (Restatement of Proposition 3.3). *In a DQ-based ICA, adding DQs can actually reduce efficiency. A single DQ can cause an efficiency drop arbitrarily close to 100%. By comparison, in a VQ-based ICA, adding additional queries can never reduce efficiency (assuming truthful bidding).*

*Proof.* Let $m = 2$, $n = 2$, $c_1 = 1$, $c_2 = 1$,

$$v_1 = \max\{400 \cdot \mathbb{1}_{x_1 \geq 1}, 2 \cdot \mathbb{1}_{x_2 \geq 1}\} \text{ and}$$
$$v_2 = 1.1 \cdot \mathbb{1}_{x_1 \geq 1}.$$

Suppose the auction has asked two DQs. The first DQ $p = (1, 1)$ is responded by both bidders with $(1, 0) \in \arg\max_{x \in \mathcal{X}} \{v_i(x) - \langle p, x \rangle\}$. The second DQ $p = (1.2, 1)$ is responded by bidder 1 with $(1, 0) \in \arg\max_{x \in \mathcal{X}} \{v_1(x) - \langle p, x \rangle\}$ and by bidder 2 with $(0, 0) \in \arg\max_{x \in \mathcal{X}} \{v_2(x) - \langle p, x \rangle\}$.

After these 2 DQs the WDP based on the inferred values (see Equation (3)), would assign item 1 to bidder 1 (resulting in an inferred social welfare of $1.2 + 0 = 1.2$). This is the efficient allocation with a true social welfare (SCW) of 400, i.e., an efficiency equal to 100%.

Now suppose that a third DQ $p = (401, 1)$ is added to the auction. Bidder 1's demand response is $(0, 1) \in \arg\max_{x \in \mathcal{X}} \{v_1(x) - \langle p, x \rangle\}$ and bidder 2's response is $(0, 0) \in \arg\max_{x \in \mathcal{X}} \{v_2(x) - \langle p, x \rangle\}$. The WDP would now assign item 2 to bidder 1 and item 1 to bidder 2, resulting in an inferred SCW of $1 + 1 = 2$). This would result only in an efficiency of $\frac{2+1.1}{400} < 1\%$.

While the inferred SCW obviously cannot decrease in any round (since the set we maximize over cannot decrease in any round and inferred values cannot decrease), we have shown here that the true SCW can decrease substantially. In this example, the SCW dropped by more than 99%. One could easily modify this example to even obtain an efficiency drop arbitrarily close to 100% if one decreases the values 1,1.2 and 2 (the prices and the values inside the value functions) by any small factor or increases the numbers 400 and 401, by any large factor. Then the proof would still work, which shows that the efficiency can even fall from 100% to values arbitrarily close to 0%.

On the other hand, if we only ask VQs, there is no difference between inferred SCW and true SCW (assuming truthful bidding), which results in non-decreasing SCW. □

*Remark* D.6. This is a significant issue in practice. In Section 6 we experimentally show that in the most realistic spectrum auction domain, the CCA's efficiency drops by over 7% with the introduction of more DQs. In a second realistic domain, the CCA actually has higher efficiency after just 5 DQs compared to after 100. This efficiency degradation is not only a concern for the CCA but also affects ML-powered DQ-based ICAs in similar ways.

In the next lemma, we show that the same issue arises in an auction that uses both DQs and VQs:

**Lemma D.7.** *In an auction that first uses DQs and then VQs, adding VQs can actually reduce efficiency. The efficiency drop can even be arbitrarily close to 100%.*

*Proof.* Consider the setting from the proof of Proposition 3.3 including the first 2 DQs. Recall that in this setting after these 2 DQs, the WDP would achieve 100% efficiency. Now instead of the third DQ, we ask the following VQ: We ask bidder 1 for her value of the bundle $(0, 1)$ and we ask bidder 2 for her value of the bundle $(1, 0)$. Then the WDP based on these 3 rounds would assign item 2 to bidder 1, and item 1 to bidder 2, as we explain in the following. The inferred SCW $v_1(0, 1) + v_2(1, 0) = 2 + 1.1$ (which is equal to the true SCW of this allocation) is higher than the inferred SCW of all other allocations consisting of elicited bundles: For bidder 1 the DQ responses were always $(1, 0)$ with inferred value 1.2, and the VQ elicited $v_1(0, 1) = 2$. For bidder 2, the DQ responses were $(1, 0)$ with inferred value 1 and $(0, 0)$ with inferred value 0, and the VQ response was $v_2(1, 0) = 1.1$. So we see that the highest inferred SCW among all feasible allocations is achieved by assigning item 2 to bidder 1 and item 1 to bidder 2 (e.g., assigning it the other way around would only achieve an inferred SCW of $1.2 + 0$, while the true SCW $v_1(1, 0) + v_2(0, 1) = 400 + 0$ would be much larger).

So the efficiency dropped from 100% to $\frac{2+1.1}{400} < 1\%$ after the VQ (i.e., the efficiency drops by more than 99%). □

Even though Lemma D.7 shows that an auction using DQs followed by VQs can still experience an arbitrarily large efficiency drop, we can completely address this issue using a *single* carefully designed VQ, which we call the "bridge bid."

**Definition D.8** (Bridge bid)**.** The bridge bid asks each bidder her value for the bundle she would have been allocated according to the WDP after the last DQ.

**Lemma D.9.** *In an ICA that first asks DQs and then VQs, by first using a single specific VQ, the bridge bid from Definition D.8, the auction can ensure its efficiency is at least as high as the efficiency achieved by its DQs alone.*

*Proof.* The bridge bid itself can obviously not decrease efficiency, because it simply replaces the inferred SCW of the winning allocation of the previous WDP with the true SCW of exactly the same allocation. In other words, the inferred values of the bundles of the previously WDP-winning allocation can be increased or stay the same, while all the other inferred values stay the same. Thus the winning allocation stays the winning allocation when the bridge bid is added. For the remainder of the proof, we will show that all the VQs after the bridge bid can also not decrease the efficiency. In every further WDP another allocation can only outperform the bridge bid allocation if it has a higher inferred[15] social welfare. However, if it has a higher inferred SCW, it's true SCW cannot be lower than the one of the bridge bid. And as we have shown in the beginning of the proof, the SCW of the allocation of the bridge bid is equal to the SCW of the last WDP winner after the last DQ. Thus, for any VQ after the bridge bid the SCW cannot be worse than the winning allocation of the WDP right after the last DQ. □

*Remark* D.10. Again, this in practice is highly impactful. In our experimental section (Section 6), we show that in the most realistic domain, our MLHCA without this bridge bid loses 7 percentage points of efficiency. The auction needs another 20

---

[15]Note that after every VQ it is still possible that the WDP combines bundles queried during any VQ with bundles that were DQ responses for any old DQ. Thus even after some VQs the inferred SCW of the WDP-winning allocation can be strictly smaller than its true SCW.

VQs to recover its DQ-only efficiency. By using just a single specialized VQ, the bridge bid, we can completely alleviate this problem. For a more detailed discussion, see Appendix G.8.

The next theorem shows an even more fundamental limitation of only asking DQs. Specifically, asking only DQs can result in low efficiency, *even* in the limit where we ask *all* possible DQs.

**Theorem D.11** (Restatement of Theorem 3.2). *For every $\epsilon > 0$, there exist infinitely many instances of auctions for which no combination[16] of DQs can achieve an efficiency above $50\% + \epsilon$. This remains true even if the bidders additionally report their true values for all bundles they requested in those DQs.*

*Proof.* First, we give an example with concrete numbers to convey the main intuition of the proof. Let $n = 2$, $m = 1$, $c_1 = 10$, $v_1(x) = 100\mathbb{1}_{x_1 \geq 1}$, $v_2(x) = 9x_1 + \frac{1}{25}x_1^2$. Here, the unique efficient allocation would assign 1 item to bidder 1 and the remaining 9 items to bidder 2, resulting in an SCW of $v_1(1) + v_2(9) = 100 + 9 \cdot 9 + \frac{9^2}{25} = 184.24$. However, there is no DQ $p \in \mathbb{R}_{\geq 0}^m$ that bidder 2 would answer with $x_2^*(p) = (9)$:

- If the price $p$ is below $\frac{94}{10}$, bidder 2 will answer with $x_2^*(p) = (10)$;

- if the price $p = \frac{94}{10}$, bidder 2 will answer with either $x_2^*(p) = (10)$ or $x_2^*(p) = (0)$;

- if the price $p$ is higher than $\frac{94}{10}$, bidder 2 will answer with $x_2^*(p) = (0)$.

Therefore, the WDP cannot assign 9 items to bidder 2 if only DQs were asked, no matter how many DQs were asked. Raising the bids for those bundles would also not help, because this would still not give us any value for 9 items for bidder 2. The best SCW that such WDPs based on DQ responses (and raised DQ responses) can achieve is thus 100, which results in an efficiency of $\frac{100}{184.24} \approx 54.28\%$.

After this concrete intuitive example, we give the general proof. Let $n = 2$, $m = 1$, $\max(2, \frac{1}{\epsilon}) < c_1 \in \mathbb{N}$, $v_1(x) = c_1^2\mathbb{1}_{x_1 \geq 1}$, $\delta \in (0, \min(0.5, \epsilon))$, $v_2(x) = (c_1 - \delta)x_1 + \frac{\delta}{c_1^2}x_1^2$. Here, the unique efficient allocation would assign 1 item to bidder 1 and the remaining $c_1 - 1$ items to bidder 2, resulting in an SCW of $v_1(1) + v_2(c_1 - 1) = c_1^2 + (c_1 - \delta)(c_1 - 1) + \frac{\delta}{c_1^2}(c_1 - 1)^2$. However, there is no DQ $p \in \mathbb{R}_{\geq 0}^m$ that bidder 2 would answer with $x_2^*(p) = (c_1 - 1)$, due to the strict convexity of $v_2$:

- If the price $p$ is below $\frac{v_2(c_1)}{c_1} = (c_1 - \delta) + \frac{\delta}{c_1}$, bidder 2 will answer with $x_2^*(p) = (c_1)$;

- if the price $p$ is exactly $\frac{v_2(c_1)}{c_1}$, bidder 2 will answer with either $x_2^*(p) = (c_1)$ or $x_2^*(p) = (0)$;

- if the price $p$ is higher than $\frac{v_2(c_1)}{c_1}$, bidder 2 will answer with $x_2^*(p) = (0)$.

Therefore, the WDP cannot assign $c_1 - 1$ items to bidder 2 if only DQs were asked, no matter how many DQs were asked. Raising the bids for those bundles would also not help, because this would still not give us any value for $c_1 - 1$ items for bidder 2. The best SCW that such WDPs based on DQ responses (and raised DQ responses) can achieve is thus $c_1^2$, which results in an efficiency of $\frac{c_1^2}{c_1^2 + v_2(c_1 - 1)} < 50\% + \epsilon$. Since there are infinitely many possible choices of $\delta \in (0, \min(0.5, \epsilon))$, the proof is concluded. (Note that there are infinitely many other scenarios that were also suitable for this proof.) □

Thus, every method that only asks DQs (e.g., CCA or (Soumalias et al., 2024c)) will result in inefficient allocations even in the limit of infinitely many iterations in the case of certain value functions (even if raised clock bids are added in the supplementary round).

*Remark* D.12. The issue highlighted in Theorem 3.2 also arises in practical settings. In Section 6, we experimentally show that in the most realistic domain, MRVM, the final 50 DQs of ML-CCA (Soumalias et al., 2024c), the current SOTA DQ-based ICA, only increase efficiency by 0.3% points. If the bidders also report their true values for all bundles they requested, this only causes an efficiency increase of less than 0.2% points. In contrast, for MLHCA, the last 30 VQs cause an efficiency increase of over 1.8% points. For the other domains, we see a qualitatively similar picture in Figure 1.

---

[16]We want to emphasize that Theorem 3.2 holds for any combination of DQs, even combinations consisting of all (unaccountably many) possible DQs (i.e., a DQ for every price vector in $[0, \infty)^m$). For these auction instances, even an oracle with complete knowledge about everything and infinite computational abilities could not ask any combination of DQs resulting in a more efficient allocation (see Remark D.13). Theorem 3.2 implies that no DQ-only mechanism can guarantee an efficiency above 55%.

*Remark* D.13. The proof of Theorem 3.2 is *not* related to Proposition 3.3. Theorem 3.2 also holds if you allow an oracle with complete information to choose any set of DQs (i.e., without using any of the harmful DQs from Proposition 3.3). Theorem 3.2 and Proposition 3.3 are two distinct orthogonal problems of DQs. For example, Theorem 3.2 also holds when the *clock-bids raised* heuristic is used, while Proposition 3.3 does not hold if the *clock-bids raised* heuristic is used.[17]

In Theorem 3.2, we showed that a DQ-based auction cannot guarantee full efficiency. Intuitively, the driving force behind this limitation is that despite the broad information that DQs provide, they cannot fully reveal a bidder's value function. In Example 1, we show a practical example where both linear and non-linear value functions would result in exactly the same response to any DQ by the bidder. However, the same is not true for a VQ-based auction, leading to the following result:

**Lemma D.14.** *Asking each bidder $\prod_{j=1}^{m} c_j$ different VQs guarantees that the allocation will be efficient. This result holds true if DQs are added to the auction.*

*Proof.* If the bidders give us their values for all possible bundles, then we have access to their complete value functions. Then the WDP is equivalent to optimizing the SCW. □

Thus, (Weissteiner & Seuken, 2020; Weissteiner et al., 2022b;a; 2023) and the hybrid method that we introduce in this paper have a guarantee to converge to an efficient allocation in the limit of infinitely many iterations.[18]

The number $\prod_{j=1}^{m} c_j$ in Lemma D.14 is obviously just a worst-case bound for the worst possible querying strategy. In theory, it would be sufficient to ask each bidder only one VQ $v_i(a_i^*)$ corresponding to the efficient allocation $a^*$ to achieve 100% efficiency. However, in practice, the auctioneer does not know $a^*$. Our experimental results displayed in Table 2 show that our method MLHCA usually finds the efficient allocation after *much* fewer queries than $\prod_{j=1}^{m} c_j$. And once each bidder answers the VQ $v_i(a_i^*)$, further queries cannot reduce the efficiency anymore.

To better understand the importance of asking VQs, we directly compare Theorem 3.2 and Lemma D.14. While Theorem 3.2 proves that there exist auction instances where even an oracle with full information cannot find any combination of DQs that results in a decent efficiency above 55%, we know that an oracle can always find a single VQ that directly results in a perfect efficiency of 100%. Theorem 3.2 proves that even without any prior information, asking at most $\prod_{j=1}^{m} c_j$ arbitrarily stupidly chosen different VQs always results in 100% efficiency, while Theorem 3.2 proves that even asking infinitely many arbitrarily smartly chosen DQs can result in catastrophically bad efficiencies below 55%. In practice, we neither have full oracle information nor can we afford to ask $\prod_{j=1}^{m} c_j$ stupidly chosen different VQs; however, our experiments show that a few VQs chosen by our MLHCA usually result in very high efficiencies.

### D.3. The Advantages of Combining DQs and VQs

**DQs are Cognitively Simpler Than VQs Early in the Auction.** All ML-powered, VQ-based ICAs in the literature begin by asking bidders their values for uniformly at random selected bundles to initialize the ML models. In contrast, the SOTA ML-powered DQ-based approach (Brero & Lahaie, 2018; Brero et al., 2019; Soumalias et al., 2024c) starts by asking bidders for their preferred bundles at low initial prices that gradually increase over rounds. From a practical standpoint, it is nearly impossible for bidders to accurately assess VQs for randomly chosen bundles, whereas responding to DQs with low prices is far easier.[19] As the auction progresses and the bidders' ML models become more accurate, a VQ-based ML-powered ICA can ask targeted VQs that align better with bidder interests, making them easier to answer.[20] See Appendix D.5 for an extended discussion.

---

[17]When the *clock-bids raised* heuristic is used, there are no harmful DQs (i.e., every further DQ can only increase the SCW, but never decrease the SCW). Even then, Theorem 3.2 shows that any arbitrary (possibly infinite) set of DQs cannot reach more than 55% efficiency for the auction instance in the proof of Theorem 3.2 for example.

[18]Note that all these methods always enforce to ask a new VQ in any round, i.e., if the WDP suggests to ask a bidder a VQ for a bundle she was already asked for in a previous round, then we solve a constrained WDP instead with the constraint that this bidder is not allowed to be asked for any previously asked bundle again.

[19]In the example from Footnotes 13 and 14, imagine being asked your value for a bundle of 30 frying pans and 500 coconuts. It's hard to assess such a random combination. Now, imagine shopping at a supermarket with a 50% discount across all items; it's easier to determine what items you want under these conditions.

[20]Continuing with our example, imagine being asked for the value of ingredients specifically for a strawberry cake in one iteration and for a blueberry cake in the next. If your goal is to bake a cake, these targeted VQs are much easier to respond to.

**DQs are More Effective in the Early Stages of the Auction.** Initially, the auctioneer lacks knowledge of which bundles align with bidders' interests. Beginning with DQs allows the auctioneer to gather early insights about the bidders' preferences over the whole bundle space, facilitating the use of more targeted queries later on. This practice is well-established in the combinatorial auction community. For instance, the initial DQ phase in the CCA is often referred to as a "price discovery phase" (Ausubel et al., 2006). We argue that the same concept holds even in ML-powered auctions. Our experiments in Section 6 confirm that DQ-based approaches (e.g., ML-CCA (Soumalias et al., 2024c)) outperform VQ-based approaches (Weissteiner & Seuken, 2020; Weissteiner et al., 2022b;a; 2023) during the early rounds of the auction. However, as suggested by Theorem 3.2 and Lemma D.14 , VQ-based approaches eventually surpass DQ-based mechanisms in later iterations.

A key contributing factor as to why VQ-based ML-powered approaches perform better than DQ-based approaches is that they can take into account the WDP, i.e., the downstream optimization problem that will determine the final allocation.[21] In contrast, responses to a single DQ often lead to over-demand for certain items or leave some items unassigned (under-demand). In Example 1, bidder 2 lacks information to know that she should bid for 9 items. Only the auctioneer, having information from all bidders, knows that assigning 9 items to bidder 2 would complement bidder 1's preferences. The auctioneer can leverage this aggregated knowledge by asking bidder 2 a VQ for 9 items, whereas DQs alone would not provide this opportunity.

*Example* 1. In the example from the proof of Theorem 3.2, after sufficiently many DQs have been asked, a single VQ would suffice to increase the social welfare from $\approx 56.45\%$ to $100\%$. MLHCA would ask this VQ in its first VQ round, provided that enough DQs had been asked beforehand, as we explain in the remainder of this example. $v_1$ can be very precisely reconstructed from DQs $p = \epsilon$ (which is responded by $x = (1)$), $p = 100 - \epsilon$ (which is responded by $x = (1)$), and $p = 100 + \epsilon$ (which is responded by $x = (0)$). The last two DQs reveal that $100 - \epsilon \leq v_1(1) \leq 100 + \epsilon$. And the first DQ reveals that $v_1(x) - v_1(1) \leq (x - 1)\epsilon$ for any $x > 1$. Combining these information reveals that $v_1 \leq 100 \mathbb{1}_{\cdot > 1} + 10\epsilon$ and with the help of monotonicity these 3 DQs reveal that $v_1 \geq 100 \mathbb{1}_{\cdot \geq 1} - \epsilon$. So, we can reconstruct the true $v_1$ up to $10\epsilon$. For bidder 2, from DQs $p = \frac{94 - \epsilon}{10}$ (which is responded by $x = 10$) and $p = \frac{94 + \epsilon}{10}$ (which is responded by $x = 0$), we can only reconstruct that $v_2(\cdot) \leq \frac{94 + \epsilon}{10}(\cdot)$ and that $v_2(10) \geq 94 - \epsilon$. E.g., the linear function $\frac{94}{10}(\cdot)$ would not contradict any possible DQ response from bidder 2. Our ML algorithm should not have any problem with estimating $v_1$ sufficiently well. If additionally, our ML algorithm estimates $v_2$ (approximately) as this linear function $\frac{94}{10}(\cdot)$, then the WDP would directly assign 1 item to bidder 1 and 9 items to bidder 2, which is the efficient allocation. In theory, MVNNs could also express functions that achieve 0 training loss on all DQs for bidder 2 but do not result in an efficient allocation. However, these functions would be highly non-linear and for many NN architectures it is shown that they prefer functions which are in a certain sense close to linear (Heiss et al., 2019; 2023; 2021; Heiss, 2024). In Appendix C.1 we explain, why our MVNNs would learn a linear approximation of $v_2$. Therefore the WDP would result in the efficient allocation in this example.

*Remark* D.15. Note that this example is not pathological. In Section 6, we will show that in realistic domains using 40 DQs and only 2 VQs (1 bridge bid + 1 ML-VQ), our MLHCA can achieve higher efficiency than the SOTA DQ-based mechanism using 100 queries.

Our MLHCA is the first auction to integrate both a sophisticated DQ and VQ generation algorithm. By leveraging insights from auction theory and starting with DQs before transitioning to VQs, MLHCA achieves state-of-the-art efficiency in all rounds and demonstrates significantly improved final efficiency across all domains compared to the current state-of-the-art.

Moreover, we argue that the combination of DQs and VQs is particularly powerful for learning bidders' value functions, as the information from these two query types complements each other nicely (see Appendix E).

### D.4. Why One Should Ask DQs Before VQs and Not the Other Way Around

We neither want to claim that DQs are more informative or better than VQs, nor the other way around. We strongly believe that DQs are more informative and practical at the start of the auction, while we also believe that VQs are more informative and more effective at the end of the auction. We have multiple different reasons to believe so, adding up to very large effects in our experiments in Appendix G.9. In the following paragraphs, we'll quickly summarize these reasons.

---

[21]By definition, all the bundles in a VQ form a feasible allocation. Furthermore, VQs typically allocate (almost) all items to bidders, as they maximize the estimated social welfare. The MVNN architecture ensures monotonicity in the estimated value functions. If the estimated value functions were strictly monotonic, the solution to the MILPs determining the next VQ would always allocate all items.

**Reason 1: Congnitively Simpler.** While our experiments do not measure the cognitive load on bidders, we already argued in Appendix D.3 that at the early stage of the auction, answering VQs is considered to be extremely difficult by practitioners, while starting ICAs with DQs is and ending them with value-bids is very common in practice. See Appendix D.5 for an extended discussion.

**Reason 2: Global Exploration vs Local Exploitation.** At the start of an auction, it is not known which regions of the bundle space could be particularly relevant for a bidder. DQs provide very quickly a coarse but global overview of a bidder's value function. This overview remains very valuable for all later iterations in deciding which regions are worth exploring and which are not. In contrast, the values of some random bundles (which are with high probability far away from the relevant region of the bundle space for large, combinatorial domains) will become almost irrelevant during the later iterations of the auction. This is why DQs are more valuable at the start of the auction. However, towards the end of the auction, entirely the opposite is true; the ML models already have lots of information on the bidders' value functions and therefore can pinpoint which regions of the bundle space are relevant for which bidder. Therefore very precise local information in exactly those regions becomes extremely valuable, and VQs can ask for exactly this kind of information, whereas the coarse, global information of DQs brings almost no additional value at this point.[22] Theorem 3.2 and Example 1 illustrate this point mathematically. They show scenarios in which it is impossible for DQs to provide *any* additional information on the bidders' value functions, since infinitely many very different value functions with the same concave envelope would result in exactly the same DQ reply, making these value functions indistinguishable from any DQ. At the same time, this information would be crucial for achieving an acceptable efficiency, and those value functions are clearly distinguishable using VQs. In this example, the first ML-VQ after sufficiently many DQs would directly deliver the needed information to almost double the efficiency. And, in contrast to DQs, Lemma D.14 shows that, asking sufficiently many VQs can always exactly recover the true value function.

**Reason 3: ML-VQs Result in Compatible Bids, While DQs Do Not.** Each bidder answers a DQ with a requested bundle. However, combining all these bundles usually does not result in a feasible allocation that allocates all items. In fact, this is only the case if linear *clearing prices* have been found, which is extremely challenging in realistic domains. In fact, in the most realistic domain, the SOTA approach of Soumalias et al. (2024c) clears 0% of the instances tested, and it is not even known whether linear clearing prices exist. For the auction in Example 1, linear clearing prices provably do not exist, making it mathematically impossible to find clearing prices via DQs. However, for our ML-VQs, our algorithm first selects a promising *feasible* allocation (that allocates all items) and then asks each bidder her value for the bundles she receives in that allocation. This implies that for an ML-VQ round, the bids of all the bidders together result in a feasible allocation. In the early rounds of an auction, the efficiency of the intermediate WDPs are completely irrelevant, as they are not final. The goal of the first rounds is mainly to gather relevant information for later rounds. Therefore, it does not hurt that the bids from the first DQ-rounds are not compatible. However, exactly the opposite is true in the last rounds of the auction. For example, in the very last round, improving the models of bidders' value functions does provide *any* value anymore, since these models are not used anymore after the last round. However, obtaining a highly efficient *feasible* allocation is the number one goal at the end of the auction. Theorem 3.2 and Example 1 show that in many scenarios, even if one had perfect information on all the value functions, no combination of DQs can obtain bids that result in a feasible allocation with an acceptable efficiency. In contrast, in any possible scenario, a single ML-VQ would always result in a feasible allocation with 100% efficiency if one had perfect information on all the value functions. And our experiments show that after 40 DQs (and 1 bridge bid), we often have already sufficient information to directly obtain 100% efficiency after a single ML-VQ, and even if we do not directly achieve 100% efficiency after the first ML-VQ, we usually achieve already a very good efficiency a few ML-VQs later.

We think that Reason 1, is the most relevant for implementing practical auctions while being completely unrelated to our experimental results. We hypothesize that Reasons 2 and 3 are the main explanations of our experimental results and explain why adding DQs *after* 80 VQs does not bring any efficiency gains in our experiments in Appendix G.9. Note that both Reason 2 and 3 consist of 4 subreasons each. Each of them consist of an advantage of DQs in early rounds, a disadvantage of VQs in early rounds, a disadvantage of DQs in late rounds and an advantage of VQs in late rounds. They all point in the same direction to first ask DQs and then ask VQs afterwards. While we did not find a single reason to switch the order, we found even further (maybe more subtle) reasons to start the auction with DQs.

---

[22]For strong empirical evidence of this diminished value of DQs, see Appendix G.9

**Reason 4: At the Start DQs Provide Better Bids and Better Local Information Than VQs**  Reason 2 says that at the start of the auction, *mainly* global information is relevant, and Reason 3 says that at the start, one cares *mainly* about learning the value functions rather than identifying bundles relevant for the WDP. However, even at the start it is beneficial to *additionally* obtain useful local information and relevant bids. Proposition 3.1 is telling us that at the beginning of the auction DQs even provide more relevant local information than random VQs, especially for high dimensional bundle spaces. While random VQs result in bids for random bundles, DQs always result in bids for bundles that are at least to some extent aligned with the bidders' interests, providing useful local information as well. Figure 6 in Appendix G.9 suggests that the dimension of GSVM and LSVM is high enough for this argument to hold, but the 3 dimensions of SRVM are not enough.

### D.5. Intuitive Arguments Why DQs are Cognitively Simpler Than VQs Early in the Auction

This subsection offers an intuitive and informal discussion, grounded in common sense and informed by conversations with practitioners, in contrast to the more formal and scientifically rigorous analysis found throughout the rest of the appendix and the main paper. While answering a DQ may appear computationally hard in theory[23], real-world bidders often do not need to compute the precise value of any bundle to answer a DQ. Instead, they can make comparisons based on relative value. For instance, a bidder may know that certain licenses are not worth their prices and can confidently exclude them without quantifying by how much. This is sufficient to give an exact answer to a DQ.

For example, the experimental setting in Scheffel et al. (2012a) does not optimally reflect this reality. In their study, participants were given explicit formulas for their value functions. In that setting, it is relatively easy to evaluate the value of any bundle (i.e., to answer a VQ), but computationally hard to solve the optimization problem required to answer a DQ. However, in actual spectrum auctions, bidders do not have such formulas. Especially for bundles outside their business model, it can be very difficult to estimate values at all. In contrast, bidders often have strong intuition about how different bundles compare to each other, which is exactly what DQs require. In this sense, real-world bidders are often better equipped to answer DQs than VQs—opposite to the assumptions in Scheffel et al. (2012a).

To further illustrate, imagine a shopper who only visits a supermarket once a year and must buy enough food to survive the year—just as telecoms acquire spectrum licenses only in rare, infrequent auctions. It may be difficult for the shopper to assign an exact monetary value to the minimum amount of food they need. Yet, when comparing bananas and apples with price tags, it's relatively easy to decide which one to buy more of, based on their relative value. Making this choice corresponds to answering a DQ and does not require the bidder to give absolute values.

In established real-world spectrum auctions, such as the CCA, the auction begins with DQs and only later allows bidders to report values for additional bundles in a supplementary round (Ausubel et al., 2006). This structure reflects the widely held belief among practitioners—and one we strongly share—that VQs are more difficult to answer at the beginning of an auction than toward the end, when both bidders and the auctioneer have developed a better understanding of relevant bundles and prevailing price levels. While this hypothesis is well aligned with practical auction design, there is still limited scientific work formally analyzing the cognitive demands of DQs versus VQs. We believe future research could provide a more refined and nuanced understanding of when and why each type of query is cognitively easier or harder to answer—insights that could further inform auction design in practice.

---

[23]While the worst-case computational cost of answering a DQ is exponential, our algorithm computes near-optimal answers millions of times per auction in under 0.2 seconds per query (see Line 4 in Algorithm 4). Although real-world value functions may be more complex than our MVNN approximations, the main bottleneck in practice is not computation but uncertainty in the bidder's own value estimates.

# E. Details on Section 4

In this section, we introduce our mixed training algorithm and provide experimental evidence supporting our theoretical analysis from Section 3. Specifically, we demonstrate the learning benefits of initializing auctions with DQs rather than VQs and highlight how combining DQs with VQs leads to superior learning performance.

## E.1. Training Algorithm Detailed Description

In this section, we provide the details on our training algorithm to combine DQs and VQs. To leverage the advantages of both DQs and VQs, we propose a straightforward two-stage training algorithm. In each epoch, the ML model is first trained on all DQ responses using the loss function from (Soumalias et al., 2024c) (Lines 4 to 6). The main idea behind this loss, is that for each DQ, an optimization problem is solved to predict the bidder's utility-maximizing bundle at the given prices, treating her ML model as her true value function. In case the predicted reply disagrees with the bidder's true reply, the loss is the difference in predicted utilities between these 2 bundles, given the current prices. Next, in each epoch, the model is trained on the VQ responses using a standard regression loss (Lines 8 to 10). This mixed approach ensures that the model benefits from both the broad information of DQs and the precise value information from VQs.

---

**Algorithm 4:** MIXEDTRAINING

**Input** : Demand query data $R_i^{\text{DQ}} = \{(x_i^*(p^r), p^r)\}_{r=1}^R$, Value query data $R_i^{\text{VQ}} = \{(x_i^l, v_i(x_i^l))\}_{l=1}^L$ Epochs $T \in \mathbb{N}$, Learning Rate $\gamma > 0$, Cardinal loss function $F$ (e.g., least-square loss)

1   $\theta_0 \leftarrow$ init mMVNN              ▷ Weissteiner et al. (2023, S.3.2)
2   **for** $t = 0$ *to* $T - 1$ **do**
3      **for** $r = 1$ *to* $R$ **do**                       ▷ Demand responses for prices
4         Solve $\hat{x}_i^*(p^r) \in \arg\max_{x \in \mathcal{X}} \mathcal{M}_i^{\theta_t}(x) - \langle p^r, x \rangle$
5         **if** $\hat{x}_i^*(p^r) \neq x_i^*(p^r)$ **then**              ▷ mMVNN is wrong
6            $L(\theta_t) \leftarrow \left( (\mathcal{M}_i^{\theta_t}(\hat{x}_i^*(p^r)) - \langle p^r, \hat{x}_i^*(p^r) \rangle) - (\mathcal{M}_i^{\theta_t}(x_i^*(p^r)) - \langle p^r, x_i^*(p^r) \rangle) \right)^+$    ▷ Add predicted utility difference to loss
7            $\theta_{t+1} \leftarrow \theta_t - \gamma (\nabla_\theta L(\theta))_{\theta = \theta_t}$                  ▷ SGD step
8      **for** $l = 1$ *to* $L$ **do**                            ▷ Value Queries
9         $L(\theta_t) \leftarrow F(\mathcal{M}_i^{\theta_t}(x_i^l), v_i(x_i^l))$            ▷ Cardinal Loss on VQs
10        $\theta_{t+1} \leftarrow \theta_t - \gamma (\nabla_\theta L(\theta))_{\theta = \theta_t}$                  ▷ SGD step
11 **return** *Trained mMVNN* $\mathcal{M}_i^{\theta_T}$

---

*Remark* E.1 (Computationally efficient implementation of Algorithm 4). In practice, running Algorithm 4 exactly the way it is printed here would be quite slow. Our actual implementation does not rerun Line 4 *every* epoch, but reuses $\hat{x}_i^*(p^r)$ for a couple of epochs. This does not violate the bidder's preference due to the positive part in Line 6. In this way, we can significantly speed up the training algorithm.

This training algorithm can be interpreted as a stochastic gradient descent on the loss function

$$L_i(\theta) := \sum_{r=1}^R \left( \mathcal{M}_i^\theta(\hat{x}_i^*(p^r)) - \langle p^r, \hat{x}_i^*(p^r) \rangle - \left( \mathcal{M}_i^\theta(x_i^*(p^r)) - \langle p^r, x_i^*(p^r) \rangle \right) \right)^+ + \sum_{l=1}^L F(\mathcal{M}_i^\theta(x_i^l), v_i(x_i^l)),$$

where and $F$ is a loss function (e.g., $F(y, \tilde{y}) = (y - \tilde{y})^2$) and $\hat{x}_i^*(p^r) \in \arg\max_{x \in \mathcal{X}} \mathcal{M}_i^{\theta_t}(x) - \langle p^r, x \rangle$.

In practice, one can obviously use modifications of this algorithm such as adding regularization or momentum or other typical deep learning techniques.

## E.2. Experimental Analysis

In this section, we demonstrate the learning benefits of initializing auctions with DQs rather than VQs and highlight how combining both query types leads to superior learning performance.

We conduct the following experiment: We perform *hyperparameter optimization (HPO)* to train an mMVNN for the most critical bidder type in the most realistic domain—the national bidder in the MRVM domain. In Appendix E.3 we present the same experiment for all other domains. Our HPO procedure is the following. For a single bidder of that type, we generate three distinct training sets:

| **Optimization Metric** | **Train Points** | | **R²** | | **KT** | | **MAE scaled** | | **R²$_c$** | |
|---|---|---|---|---|---|---|---|---|---|---|
| | VQs | DQs | $\mathcal{T}_r$ | $\mathcal{T}_p$ | $\mathcal{T}_r$ | $\mathcal{T}_p$ | $\mathcal{T}_r$ | $\mathcal{T}_p$ | $\mathcal{T}_r$ | $\mathcal{T}_p$ |
| $R^2$ on $\mathcal{V}_r$ | 20 | 40 | 0.84 | 0.42 | 0.79 | 0.80 | 0.037 | 0.044 | 0.84 | 0.80 |
| | 60 | 0 | 0.73 | $-10.07$ | 0.68 | 0.64 | 0.052 | 0.236 | 0.74 | 0.20 |
| | 0 | 60 | 0.24 | $-3.07$ | 0.77 | 0.77 | 0.103 | 0.128 | 0.83 | 0.76 |
| $R^2$ on $\mathcal{V}_p$ | 20 | 40 | 0.82 | 0.01 | 0.79 | 0.80 | 0.041 | 0.062 | 0.84 | 0.83 |
| | 60 | 0 | 0.76 | $-3.40$ | 0.72 | 0.62 | 0.049 | 0.141 | 0.77 | 0.05 |
| | 0 | 60 | $-0.05$ | $-6.24$ | 0.78 | 0.72 | 0.103 | 0.154 | 0.84 | 0.69 |

Table 3: Learning comparison of training only on DQs, only on VQs, or on both. Shown are averages over ten instances for the winning configuration of each HPO procedure. Winners are marked in gray.

1. The first training set contains 40 DQs simulating 40 CCA clock rounds, along with 20 VQs for bundles chosen uniformly at random.

2. The second training set consists of 60 DQs, simulating 60 CCA clock rounds, with no VQs.

3. The third training set contains 60 VQs and no DQs.

We evaluate the generalization performance of the trained models on two distinct sets: A *random bundle set* ($\mathcal{V}_r$), which consists of 50,000 bundles sampled uniformly at random from the bundle space. A *random price-driven set* ($\mathcal{V}_p$), which consists of the bundles requested by the bidder in 200 randomly generated price vectors $\{p^r\}_{r=1}^{200}$, where each item's price is drawn uniformly between 0 and three times its average value for that bidder type. $\mathcal{V}_r$ evaluates generalization performance over the entire bundle space, while $\mathcal{V}_p$ focuses on the bidder's utility-maximizing bundles for various prices.

For each HPO configuration, we average the performance across 10 bidders of the same type. The best-performing configuration for each validation set is selected based on the coefficient of determination.

For the selected configurations, we evaluate performance on 10 separate test seeds representing new bidders, generating the test sets $\mathcal{T}_r$ and $\mathcal{T}_p$ in the same way as for the validation sets. For each test set, we report the *coefficient of determination ($R^2$)*, *Kendall Tau (KT)*, scaled *Mean Absolute Error (scaled MAE)* normalized with respect to the average value of a bundle in that domain and $R^2$ centered ($R^2_c$), a shift invariant version of $R^2$. An $R^2_c$ value of 1 indicates that the ML model has learned the bidder's value function perfectly, up to a constant shift. By comparing $R^2_c$ with the standard $R^2$, we can assess, for the bundles tested, the shift magnitude in the learned value function.[24]

Each HPO procedure was conducted under identical conditions, including the same test instances, random seeds, hyperparameter search space, and total computation time. For more details on the HPO process, see Appendix G.3.

Table 1 shows that training on a mixture of DQs and VQs consistently outperforms training on either query type alone. This is evident across all metrics, and especially for the utility-maximizing bundles of test set $\mathcal{T}_p$, where mixed training yields almost three times lower MAE compared to other approaches.

Furthermore, the mixed-query model was the only one able to approximately learn the correct mean value for both validation sets, as reflected by the small difference between its $R^2_c$ and standard $R^2$. In contrast, models trained solely on DQs or VQs showed a much larger discrepancy between these two metrics for at least one of the validation sets. As explained in Section 3, when training only on DQs, the model only has relative information about bundle values and thus the value function is not uniquely identifiable, preventing the network from learning it accurately. On the other hand, models trained solely on VQs experience a distributional shift between the two test sets—one set focuses on utility-maximizing bundles, while the other contains bundles selected uniformly at random. Since the VQ training set is drawn uniformly at random and lacks utility-maximizing bundles, the model fails to capture the bidder's value function for these critical bundles.[25]

In Table 1 we observe that the models trained only on DQs exhibit much better generalization performance in the bundles of $\mathcal{T}_p$ than the models trained on random VQs, despite of their lack of absolute value information. The reason for the better generalization performance is the strong distributional shift between the bundles of the two sets. But from an allocative value perspective, the bundles in the set $\mathcal{T}_p$ are those for which the bidders have high utility, and thus value. Thus, this is the critical

---

[24]Note that this shift is not perfectly constant as (m)MVNNs map the zero bundle to zero.

[25]Note that at the start of an ML-powered, VQ-based auction, the ML models are not yet sufficiently accurate, preventing the auctioneer from asking VQs for utility- or value-maximizing bundles.

area of the allocation space where the auctioneer wants the models to perform well. This gives strong empirical motivation as to why starting the learning process with DQs is more effective than starting it with VQs. In Section 6 we showed that the efficiency after the first ML-powered VQ is, across all domains, much higher for the model trained on DQs compared to the one trained on random VQs. The reason behind this improvement is precisely the fact that the DQ-trained models have learned a better approximation of the bidders' value functions in the most critical part of the allocation space. In fact, the learning performance is so much better that, in two out of the four realistic domains tested, a *single* ML-powered VQ in the DQ-trained networks suffices to achieve better auction efficiency than the VQ-trained networks using 60 ML-powered VQs.

Comparing the models trained only on DQs with random VQs in Table 1 provides strong empirical evidence of the two main, orthogonal learning advantages of starting an ML-powered auction with DQs compared to random VQs. The first advantage is that CCA DQs provide global information about the bundle space, which promotes exploration of the allocation space. This global information that DQs provide is evident from the higher KT that the DQ-trained network can achieve across both test sets compared to the VQ-trained one. The reason for this increased performance is that, as explained in Section 3, DQs provide global relative information about the entire allocation space.

The second learning advantage of starting an auction with CCA DQs is that they provide particularly much information about the critical, high valued areas of the allocation space right from the start. This is evident from the fact that the models trained only on DQs exhibit much better generalization performance in the bundles of $\mathcal{T}_p$ than the models trained on random VQs, despite their lack of absolute value information. The reason for the better generalization performance is the strong distributional shift between the bundles of the two sets. But from an allocative value perspective, the bundles in the set $\mathcal{T}_p$ are those for which the bidders have high utility, and thus value. Thus, this is the critical area of the allocation space where the auctioneer wants the models to perform well.

These two learning advantages are so critical that, as we will demonstrate in Section 6, the efficiency gains after the first ML-powered VQs is, across all domains, much higher for the model trained on DQs compared to the one trained on random VQs. In fact, the learning performance is so much better that, in two out of the four domains, our hybrid auction (Section 5) using just two ML-powered VQs, following training on 40 DQs, achieves higher efficiency than the SOTA VQ-based mechanism using 40 random VQs and 60 ML-powered VQs.

In Figure 3, we present prediction vs. true value plots for the top-performing configurations with respect to $R^2$ on $\mathcal{V}_r$ from Table 1. We compare the model trained on 40 DQs and 20 VQs against the one trained on 60 VQs, corresponding to the first and second rows of Table 1. Bundles from $\mathcal{T}_c$ are represented by red circles, while those from $\mathcal{T}_p$ are shown in blue. For bundles in $\mathcal{T}_p$, we also plot their *inferred values*, reflecting their price when the bidder requested them.

In Figure 3a we observe that the model trained solely on VQs consistently under predicts values for bundles in $\mathcal{T}_p$. Furthermore, there is a very large spread in the predicted values of these bundles. These bundles are out of distribution for the network, and thus it cannot generalize to them. If we examine the inferred values for the same bundles, we observe a substantial deviation from the true diagonal. The vertical distance between each bundle's inferred value and the true diagonal line corresponds to the bidder's utility when requesting that bundle - the quantity she is maximizing. In contrast, as shown in Figure 3b, the model trained on the mixed dataset is able to place the bundles of $\mathcal{T}_p$ in an almost perfect parallel line to the true diagonal, and with a much smaller shift. These bundles are not out of distribution for that network, which means it can perform better. For the bundles of $\mathcal{T}_c$, we observe that the predictions of both models are centered around the true diagonal, indicating that both networks have learned the correct mean value. However, again we can observe that for the network trained on the mixed dataset, its predictions on $\mathcal{T}_c$ are again more tightly clustered in a line around the true diagonal, as was also suggested by the stronger MAE and KT in Table 1. These observations illustrate the powerful synergy between DQs and VQs. The *global, relative information* provided by DQs enables the network to align its predictions roughly along a consistent trajectory—essentially forming a parallel line to the true diagonal. The *absolute value information* from the VQs then fine-tunes this alignment, effectively positioning the line exactly on the true diagonal, ensuring the predicted values match the true values accurately.

### E.3. Learning Experiments for Other Domains

In Tables 4 to 6 we present the results of the learning experiment of Appendix E.2 for all additional domains.

Across all domains, the network trained only on DQs demonstrates the worst generalization performance on the dataset $\mathcal{T}_r$. This is primarily due to two factors: the absence of absolute value information that VQs provide and the distributional shift between $\mathcal{T}_r$ and $\mathcal{T}_p$, with the DQ training data being more aligned with $\mathcal{T}_p$.

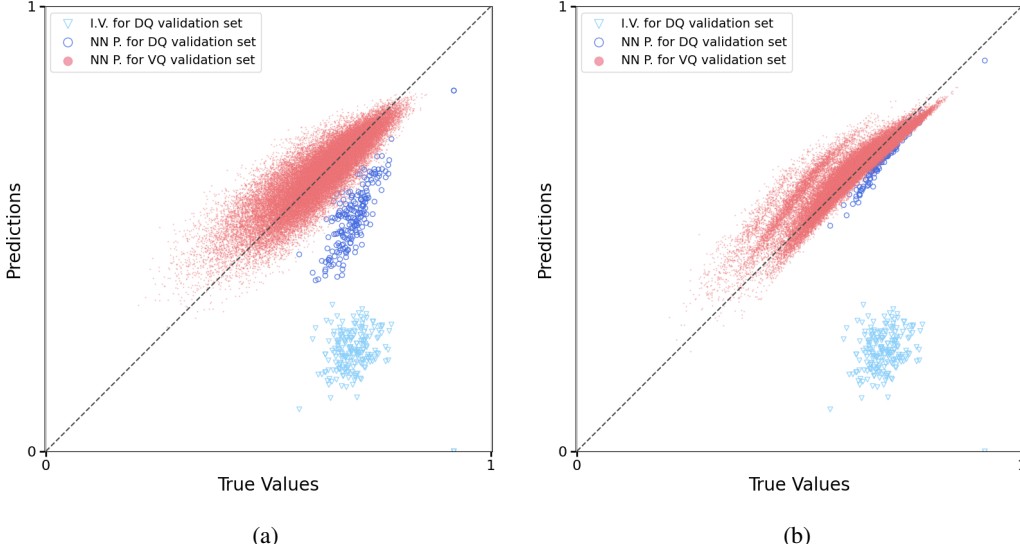

(a)              (b)

Figure 3: Comparison of scaled prediction vs. true values for an mMVNN trained with different query types for the national bidder in the MRVM domain. (a) Training with 60 demand queries. (b) Training with 40 demand queries and 20 value queries.

The performance of the network trained solely on VQs varies by domain. In the GSVM and SRVM domains, the learning task is relatively easy, as indicated by the already strong performance of previous ML-powered ICAs. In these domains, the networks trained only on VQs perform well across both test sets (Tables 4 and 6). However, in the more challenging LSVM domain—similarly to the MRVM domain discussed in Appendix E.2—the network trained exclusively on VQs performs well on the $\mathcal{T}_r$ test set, which contains points from the same distribution as its training data, but performs worse on the utility-maximizing bundles of $\mathcal{T}_p$ compared to the network trained on both query types.

This inferior learning performance on the critical dataset $\mathcal{T}_p$ explains why MLHCA outperforms pure VQ-based ML-powered ICAs, such as Weissteiner et al. (2023); Weissteiner & Seuken (2020), in the LSVM domain.

| OPTIMIZATION | TRAIN POINTS | | $\mathbf{R^2}$ | | KT | | MAE SCALED | | $\mathbf{R_c^2}$ | |
|---|---|---|---|---|---|---|---|---|---|---|
| METRIC | VQS | DQS | $\mathcal{T}_r$ | $\mathcal{T}_p$ | $\mathcal{T}_r$ | $\mathcal{T}_p$ | $\mathcal{T}_r$ | $\mathcal{T}_p$ | $\mathcal{T}_r$ | $\mathcal{T}_p$ |
| $R^2$ ON $\mathcal{V}_r$ | 20 | 40 | 0.96 | 0.95 | 0.90 | 0.94 | 0.07 | 0.12 | 0.96 | 0.98 |
| | 60 | 0 | 0.99 | 0.98 | 0.96 | 0.98 | 0.03 | 0.05 | 0.99 | 0.98 |
| | 0 | 60 | 0.79 | 0.97 | 0.83 | 0.94 | 0.04 | 0.02 | 0.91 | 0.98 |
| $R^2$ ON $\mathcal{V}_p$ | 20 | 40 | 0.96 | 0.99 | 0.91 | 0.96 | 0.07 | 0.04 | 0.97 | 0.99 |
| | 60 | 0 | 0.99 | 0.98 | 0.96 | 0.98 | 0.03 | 0.02 | 0.99 | 0.98 |
| | 0 | 60 | 0.79 | 0.97 | 0.83 | 0.94 | 0.13 | 0.05 | 0.91 | 0.98 |

Table 4: Learning comparison of training only on DQs, only on VQs, or on both for the GSVM domain.

| **OPTIMIZATION** | **TRAIN POINTS** | | **R²** | | **KT** | | **MAE SCALED** | | **R²_c** | |
|---|---|---|---|---|---|---|---|---|---|---|
| **METRIC** | VQs | DQs | $\mathcal{T}_r$ | $\mathcal{T}_p$ | $\mathcal{T}_r$ | $\mathcal{T}_p$ | $\mathcal{T}_r$ | $\mathcal{T}_p$ | $\mathcal{T}_r$ | $\mathcal{T}_p$ |
| $R^2$ ON $\mathcal{V}_r$ | 20 | 40 | 0.38 | 0.88 | 0.65 | 0.80 | 0.46 | 0.33 | 0.44 | 0.91 |
| | 60 | 0 | 0.67 | 0.80 | 0.75 | 0.81 | 0.30 | 0.46 | 0.67 | 0.87 |
| | 0 | 60 | -1.20 | 0.99 | 0.80 | 0.84 | 1.10 | 0.11 | 0.46 | 0.99 |
| $R^2$ ON $\mathcal{V}_r$ | 20 | 40 | 0.38 | 0.88 | 0.65 | 0.80 | 0.46 | 0.33 | 0.44 | 0.91 |
| | 60 | 0 | 0.65 | 0.82 | 0.81 | 0.88 | 0.25 | 0.38 | 0.66 | 0.87 |
| | 0 | 60 | -2.97 | 0.96 | 0.77 | 0.85 | 1.51 | 0.22 | 0.42 | 0.97 |

Table 5: Learning comparison of training only on DQs, only on VQs, or on both for the LSVM domain.

| **OPTIMIZATION** | **TRAIN POINTS** | | **R²** | | **KT** | | **MAE SCALED** | | **R²_c** | |
|---|---|---|---|---|---|---|---|---|---|---|
| **METRIC** | VQs | DQs | $\mathcal{T}_r$ | $\mathcal{T}_p$ | $\mathcal{T}_r$ | $\mathcal{T}_p$ | $\mathcal{T}_r$ | $\mathcal{T}_p$ | $\mathcal{T}_r$ | $\mathcal{T}_p$ |
| $R^2$ ON $\mathcal{V}_r$ | 20 | 40 | 1.00 | 0.89 | 0.97 | 0.90 | 0.02 | 0.03 | 1.00 | 0.93 |
| | 60 | 0 | 1.00 | 0.96 | 0.99 | 0.97 | 0.00 | 0.01 | 1.00 | 0.97 |
| | 0 | 60 | 0.93 | -0.13 | 0.96 | 0.92 | 0.11 | 0.10 | 0.97 | 0.94 |
| $R^2$ ON $\mathcal{V}_p$ | 20 | 40 | 1.00 | 0.94 | 0.98 | 0.92 | 0.01 | 0.02 | 1.00 | 0.94 |
| | 60 | 0 | 1.00 | 0.96 | 0.99 | 0.97 | 0.00 | 0.01 | 1.00 | 0.97 |
| | 0 | 60 | 0.91 | 0.02 | 0.96 | 0.86 | 0.12 | 0.09 | 0.95 | 0.89 |

Table 6: Learning comparison of training only on DQs, only on VQs, or on both for the SRVM domain.

## F. Detailed Auction Mechanism

---

**Algorithm 5:** MLHCA($Q^{\text{CCA}}, Q^{\text{DQ}}, Q^{\text{VQ}}, Q^{\text{round}}$)

---

    **Parameters :** $Q^{\text{CCA}}, Q^{\text{DQ}}, Q^{\text{VQ}}, Q^{\text{round}}$ and $\pi$

1  $R^{\text{VQ}} \leftarrow (\{\})_{i=1}^{N}$

2  $R^{\text{DQ}} \leftarrow (\{\})_{i=1}^{N}$

3  **for** $r = 1, ..., Q^{CCA}$ **do**                                                 ▷ Draw $Q^{\text{CCA}}$ initial prices

4     $p^r \leftarrow CCA(R^{\text{DQ}})$

5     **foreach** $i \in N$ **do**                                ▷ Initial demand query responses

6       $R_i^{\text{DQ}} \leftarrow R_i^{\text{DQ}} \cup \{(x_i^*(p^r), p^r)\}$

7  **for** $r = Q^{CCA} + 1, ..., Q^{CCA} + Q^{DQ}$ **do**                         ▷ ML-powered DQs

8     **foreach** $i \in N$ **do**

9       $\mathcal{M}_i^\theta \leftarrow \text{MixedTraining}(R_i^{\text{DQ}}, R_i^{\text{VQ}})$                 ▷ Algorithm 4

10     $p^r \leftarrow \text{NextPrice}((\mathcal{M}_i^\theta)_{i=1}^{n})$                        ▷ Appendix A.3

11     **foreach** $i \in N$ **do**                         ▷ Demand query responses for $p^r$

12       $R_i^{\text{DQ}} \leftarrow R_i^{\text{DQ}} \cup \{(x_i^*(p^r), p^r)\}$

13     **if** $\sum_{i=1}^{n}(x_i^*(p^k))_j = c_j \ \forall j \in M$ **then**             ▷ Market-clearing

14       $a^*(R^{\text{DQ}}, R^{\text{VQ}}) \leftarrow (x_i^*(p^r))_{i=1}^{n}$       ▷ Set final allocation to clearing allocation

15       Calculate payments $\pi(R^{\text{DQ}}, R^{\text{VQ}}) \leftarrow (\pi_i(R^{\text{DQ}}, R^{\text{VQ}}))_{i=1}^{n}$

16       **return** $a^*(R^{DQ}, R^{VQ})$ *and* $\pi(R^{DQ}, R^{VQ})$

17  **foreach** $i \in N$ **do**                                  ▷ Bridge bid

18     $R_i \leftarrow R_i \cup \{(a_i^*(R), v_i(a_i^*(R)))\}$

19  **for** $r = Q^{CCA} + Q^{DQ} + 2, ..., Q^{CCA} + Q^{DQ} + Q^{VQ}$ **do**       ▷ ML-powered VQs

20     **foreach** $i \in N$ **do**

21       $\mathcal{M}_i^\theta \leftarrow \text{MixedTraining}(R_i^{\text{DQ}}, R_i^{\text{VQ}})$                ▷ Algorithm 4

22     **if** $r \% Q^{round} = 0$ **then**                        ▷ Query Main Economy

23       **foreach** $i \in N$ **do**

24         $x'(R) \leftarrow \arg\max_{x \in \mathcal{F}:x_i \notin R_i^{\text{VQ}}} \sum_{i' \in N} \mathcal{M}_i^\theta(x_{i'})$     ▷ Find predicted optimal allocation

25         $x_i^*(R) \leftarrow x_i'(R)$

26     **else**                                    ▷ Query Marginal Economy

27       **foreach** $i \in N$ **do**

28         $\widetilde{N} \leftarrow$ draw uniformly at random $Q^{\text{round}} - 1$ bidders from $N \setminus \{i\}$

29         $x'(R) \leftarrow \arg\max_{x \in \mathcal{F}:x_{i'} \notin R_{i'}^{\text{VQ}}} \sum_{i' \in \widetilde{N}} \mathcal{M}_i^\theta(x_{i'})$   ▷ Find predicted optimal allocation in marginal
                                               economy

30         $x_i^*(R) \leftarrow x_i'(R)$

31     **foreach** $i \in N$ **do**                         ▷ Value query responses for $x^*(R)$

32       $R_i \leftarrow R_i \cup \{(x_i^*(R), v_i(x_i^*(R)))\}$

33  Calculate final allocation $a^*(R)$ as in Equation (3)

34  Calculate payments $\pi(R)$                              ▷ E.g., VCG (Appendix B)

35  **return** $a^*(R)$ *and* $\pi(R)$

---

In this section, we present a detailed description of MLHCA. The full auction mechanism is presented in Algorithm 5. In Lines 3 to 6, we generate the first $Q^{\text{CCA}}$ DQs using the same price update rule as the CCA. In each of the next $Q^{\text{DQ}}$ ML-powered rounds, we first train, for each bidder, an mMVNN on her demand responses (Line 9). Next, in Line 10, we call NextPrice (Soumalias et al., 2024c) to generate the next DQ based on the agents' trained mMVNNs (see Appendix A.3). If MLHCA has found market-clearing prices, then the corresponding allocation is efficient and is returned, along with payments $\pi(R)$ according to the deployed payment rule (Line 16). If, by the end of the ML-powered DQs the market has not cleared we switch to VQ rounds. In the first VQ round (Line 18) we ask each bidder for her *bridge bid* (see Definition D.8). As proven in Lemma D.9, this single VQ ensures that the MLHCA's efficiency is lower bounded by the efficiency after just the DQ rounds. The difference in the algorithm description compared to the version presented in Section 5 lies in the VQ rounds. Specifically, we make use of *marginal economies*. Once every $Q^{\text{round}}$ VQ rounds, for each bidder, we query her value for the bundle she receives in the predicted optimal allocation (based on all ML models), under the constraint that the bidder in question receives a bundle for which she has not been queried in the past (Lines 22 to 25). This is as described in Section 5. But in the other $Q^{\text{round}} - 1$ rounds, for each bidder, we query her value for the bundle she receives in the predicted optimal allocation based only on the models of the *non-marginalized bidders* (Lines 26 to 30). Each time, for each bidder,

we marginalize $Q^{\text{round}} - 1$ bidders uniformly at random without replacement. The marginal economies have been designed to improve the incentive properties of the auction (for a detailed analysis, see Brero et al. (2021)). Similar to all papers in this line of work, e.g. Brero et al. (2021); Weissteiner et al. (2022a; 2023), we set $Q^{\text{round}} = 4$ in all of our experiments. The final allocation and payments are then determined based on all reports (Lines 24 to 25). Note that ML-CCA can be combined with various possible payment rules $\pi(R)$, such as VCG or VCG-nearest.

# G. Experiment Details

Our code is available on GitHub: `https://github.com/marketdesignresearch/MLHCA`.

## G.1. SATS Domains

In this section, we provide a more detailed overview of the four SATS domains, which we use to experimentally evaluate MLHCA:

- **Global Synergy Value Model (GSVM)** (Goeree & Holt, 2010) has 18 items with capacities $c_j = 1$ for all $j \in \{1, \ldots, 18\}$, 6 *regional* and 1 *national bidder*. In this domain the value of a package increases by a certain percentage with every additional item of interest. Thus, the value of a bundle only depends on the total number of items contained in a bundle which makes it one of the simplest models in SATS. In fact, bidders' valuations exhibit at most two-way(i.e., pairwise) interactions between items.

- **Local Synergy Value Model (LSVM)** (Scheffel et al., 2012b) has 18 items with capacities $c_j = 1$ for all $j \in \{1, \ldots, 18\}$, 5 *regional* and 1 *national bidder*. Complementarities arise from spatial proximity of items.

- **Single-Region Value Model (SRVM)** (Weiss et al., 2017) has 3 items with capacities $c_1 = 6, c_2 = 14, c_3 = 9$ and 7 bidders (categorized as *local*, *high frequency*, *regional*, or *national*) and models UK 4G spectrum auctions.

- **Multi-Region Value Model (MRVM)** (Weiss et al., 2017) has 42 items with capacities $c_j \in \{2, 3\}$ for all $j \in \{1, \ldots, 42\}$ and 10 bidders (*local*, *regional*, or *national*) and models large Canadian 4G spectrum auctions.

In the efficiency experiments in this paper, we instantiated for each SATS domain the 100 synthetic CA instances with the seeds $\{101, \ldots, 200\}$. We used SATS version 0.8.1.

## G.2. Compute Infrastructure

All experiments were conducted on a compute cluster running Debian GNU/Linux 10 with Intel Xeon E5-2650 v4 2.20GHz processors with 24 cores and 128GB RAM and Intel E5 v2 2.80GHz processors with 20 cores and 128GB RAM and Python 3.8.10.

## G.3. Hyperparameter Optimization Details

In this section, we provide details on our exact HPO methodology and the ranges that we used.

We separately optimized the HPs of the mMVNNs for each bidder type of each domain, using a different set of SATS seeds than for all other experiments in the paper. Specifically, for each bidder type, we first trained an mMVNN using as initial data points the demand responses of an agent of that type during 40 consecutive CCA clock rounds, and her value responses for 30 uniformly at random selected bundles, and then measured the generalization performance of the resulting network on a validation set that consisted of 50,000 uniformly at random bundles of items, similar to $\mathcal{V}_r$ in Appendix E.2. The number of seeds used to evaluate each model was equal for all models and set to 10. Finally, for each bidder type, we selected the set of HPs that performed the best on this validation set with respect to the coefficient of determination ($R^2$). The full range of HPs tested for all agent types and all domains is shown in Table 7, while the winning configurations are shown in Table 8.

The winning configurations for both metrics are shown in Table 8.

## G.4. Computational Costs and Bidder-Perceived Latency

All experiments were conducted using the compute infrastructure described in Appendix G.2.

The average wall-clock runtime for a single instance of MLHCA across the GSVM, LSVM, SRVM, and MRVM domains was 1 day, 12 hours, and 42 minutes; 16 hours and 42 minutes; 11 hours and 6 minutes; and 7 days and 36 minutes, respectively. Each instance ran on a single CPU with 20 or 24 physical cores, as detailed in Appendix G.2, without GPU acceleration. GPUs were incompatible with both our training and query generation implementation.

Considering the substantial welfare gains achieved by MLHCA, we regard these compute costs as marginal. In total, we conducted experiments on over 1,000 realistically-sized instances.

---

[26]For the definition of (m)MVNNs with a linear skip connection, please see Weissteiner et al. (2023, Definition F.1)

| Hyperparameter | HPO-Range |
|---|---|
| Non-linear Hidden Layers | [1,2,3] |
| Neurons per Hidden Layer | [8, 10, 20, 30] |
| Learning Rate | (1e-4, 1e-2) |
| Epochs | [30, 50, 70, 100, 300, 500, 1000] |
| L2-Regularization | (1e-8, 1e-2) |
| Linear Skip Connections[26] | [True, False] |
| Cached DQ solution Frequency | [1, 2, 5, 10] |
| Batch Size for VQs | [1, 5, 10] |

Table 7: HPO ranges for all domains.

| DOMAIN | BIDDER TYPE | # HIDDEN LAYERS | # HIDDEN UNITS | LIN. SKIP | LEARNING RATE | L2 REGULARIZATION | EPOCHS | CACHED SOLUTION FREQ. | BATCH SIZE |
|---|---|---|---|---|---|---|---|---|---|
| LSVM | REGIONAL | 1 | 20 | FALSE | 0.001 | 0.000001 | 100 | 20 | 1 |
| | NATIONAL | 1 | 30 | TRUE | 0.0001 | 0.001 | 1000 | 10 | 5 |
| GSVM | REGIONAL | 2 | 30 | TRUE | 0.0001 | 0.001 | 1000 | 10 | 5 |
| | NATIONAL | 2 | 30 | FALSE | 0.0005 | 0.0001 | 200 | 10 | 1 |
| MRVM | LOCAL | 3 | 20 | TRUE | 0.001 | 0.00001 | 200 | 5 | 10 |
| | REGIONAL | 1 | 30 | TRUE | 0.0001 | 0.000001 | 1000 | 20 | 1 |
| | NATIONAL | 2 | 20 | FALSE | 0.001 | 0.000001 | 100 | 5 | 10 |
| SRVM | LOCAL | 1 | 10 | TRUE | 0.01 | 0.0001 | 1000 | 5 | 1 |
| | REGIONAL | 1 | 30 | FALSE | 0.005 | 0.000001 | 500 | 5 | 1 |
| | HIGH FREQUENCY | 1 | 10 | TRUE | 0.005 | 0.00001 | 500 | 10 | 5 |
| | NATIONAL | 1 | 10 | TRUE | 0.005 | 0.00001 | 1000 | 5 | 10 |

Table 8: Winning HPO configurations for $R^2$

Furthermore, in real-world settings, no more than two query rounds are typically conducted within a day. Therefore, the bidder-perceived latency of our mechanism is not a concern.

## G.5. Extended Efficiency Results

In this appendix we compare to further competitors including some that use significantly more queries in Table 9. Each profit-max bid (see Appendix G.6) used by some of our competitors actually consists of 1 constrained DQ plus 1 VQ. However, we count each profit-max query as only 1 VQ. Although this counting scheme is strongly in favor of the competitors, in the 3 most realistic domains LSVM, SRVM and MRVM, our method (MLHCA) significantly outperforms the efficiency of any other competitor except those who use substantially more queries. Even those mechanisms that use more queries did not manage to outperform our efficiency in any of the 4 domains.

| | | | | | | EFFICIENCY LOSS IN % | | | | | | |
|---|---|---|---|---|---|---|---|---|---|---|---|---|
| DOMAIN | MLHCA | BOCA | ML-CCA$_{CLOCK}$ | ML-CCA$_{RAISED}$ | ML-CCA$_{PROFIT}$ | CCA | CCA$_{RAISED}$ | CCA$_{PROFIT}$ | MVNN | NN | FT | RS |
| GSVM | $0.00 \pm 0.00$ | — | $1.77 \pm 0.68$ | $1.07 \pm 0.37$ | 0.00 | $9.60 \pm 1.49$ | 6.41 | 0.00 | $00.00 \pm 0.00$ | $00.00 \pm 0.00$ | $01.77 \pm 0.96$ | $30.34 \pm 1.61$ |
| LSVM | $0.04 \pm 0.07$ | $0.39 \pm 0.31$ | $8.36 \pm 1.70$ | $3.61 \pm 0.77$ | 0.05 | $17.44 \pm 1.60$ | 8.40 | 0.24 | $00.70 \pm 0.40$ | $02.91 \pm 1.44$ | $01.54 \pm 0.65$ | $31.73 \pm 2.15$ |
| SRVM | $0.00 \pm 0.00$ | $0.06 \pm 0.02$ | $0.41 \pm 0.11$ | $0.07 \pm 0.02$ | 0.00 | $0.37 \pm 0.11$ | 0.19 | 0.00 | $00.23 \pm 0.06$ | $01.13 \pm 0.22$ | $00.72 \pm 0.16$ | $28.56 \pm 1.74$ |
| MRVM | $4.81 \pm 0.57$ | $7.77 \pm 0.35$ | $6.94 \pm 0.24$ | $6.68 \pm 0.22$ | 6.32 | $7.53 \pm 0.48$ | 7.38 | 6.82 | $08.16 \pm 0.41$ | $09.05 \pm 0.53$ | $10.37 \pm 0.57$ | $48.79 \pm 1.13$ |
| | | | | | | NUMBER OF QUERIES | | | | | | |
| #DQs | 40 | 0 | 100 | 100 | 100 | 100 | 100 | 100 | 0 | 0 | 0 | 0 |
| #VQs | 60 | 100 | 0 | 1-100 | 101-200 | 0 | 1-100 | 101-200 | 100 | 100 | 100 | 100 |
| #Qs | 100 | 100 | 100 | 101-200 | 201-300 | 100 | 101-200 | 201-300 | 100 | 100 | 100 | 100 |

Table 9: Extending Table 2 by ML-CCA$_{PROFIT}$ (which adds 100 expensive profit-max bids to ML-CCA$_{RAISED}$), CCA$_{RAISED}$, CCA$_{PROFIT}$, MVNN (Weissteiner et al., 2022a), NN (based on classical neural networks (Weissteiner & Seuken, 2020)), FT (based on Fourier Transforms (Weissteiner et al., 2022b)), and RS (random search, as a baseline). Shown are averages and a 95% CI. Winners based on a $t$-test with significance level of 5% are marked in grey.

### G.5.1. COMPARING MLHCA TO ML-CCA$_{PROFIT}$

In each domain, the 2nd highest efficiency is achieved by ML-CCA$_{PROFIT}$, which uses both more DQs and more VQs than our method (see Table 2). In total ML-CCA$_{PROFIT}$ uses 201-300 queries if you count profit-max bids as only 1 query. If you count profit-max bids as 2 queries ML-CCA$_{PROFIT}$ uses 301-400 queries. Although ML-CCA$_{PROFIT}$ asks 2-4 times more queries than MLHCA, ML-CCA$_{PROFIT}$ is not able to outperform MLHCA in any domain, not even slightly. On the contrary, MLHCA substantially outperforms ML-CCA$_{PROFIT}$ in the most realistic domain MRVM.

### G.5.2. UNDERSTANDING THE SCALE OF THESE EFFICIENCY IMPROVEMENTS.

For example, the most realistic simulator MRVM was specifically designed to simulate the 2014 Canadian 4G spectrum auction. The real 2014 Canadian 4G spectrum auction achieved a revenue of USD 5.27 billion (Ausubel & Baranov, 2017). The revenue of an auction is a very conservative lower bound of the social welfare (SCW), which is typically significantly higher than the revenue. With this conservative lower bound 1% point corresponds to more than USD 50 million, probably even significantly more. For this simulation MRVM, MLHCA is outperforming *every* other 100-query method by more than 2% point (averaged over 50 random instances of the auction). Even those results of Table 2 that use substantially more than 100 queries, are outperformed by only 100 queries of MLHCA by over 1.5% points. Every supplemented version of the CCA from Table 2 (with up to 300 queries) is outperformed by only 100 queries of MLHCA by over 2% points. This is particularly interesting, since CCA is among the most popular mechanisms currently used for real-world auctions. Such an improvement 2% points would result in a gain in SCW of USD 100 million of a single instance of the 2014 Canadian spectrum auction.

Translating these results to the latest Canadian Spectrum Auction (Innovation, Science and Economic Development Canada, 2023), MLHCA's welfare gains would equate to over 50 million USD compared to all other mechanisms.

Note that repeatedly multiple spectrum auctions are conducted all over the world. E.g., CCA generated over *USD* 20 *billion* in revenue between 2012 and 2014 alone (Ausubel & Baranov, 2017).

## G.6. Details on Bidding Heuristics

The second phase of the CCA (or ML-CCA) is *the supplementary round*. In this phase, each bidder can submit a finite number of additional bids for bundles of items, which are called *push bids*. Then, the final allocation is determined based on the combined set of all inferred bids of the clock phase, plus all submitted push bids of the supplementary round. This design aims to combine good price discovery in the clock phase with good expressiveness in the supplementary

round. In simulations, the supplementary round is parametrized by the assumed bidder behaviour in this phase, i.e., which bundle-value pairs they choose to report. As in (Brero et al., 2021), we consider the following heuristics when simulating bidder behaviour:

- **Clock Bids:** Corresponds to having no supplementary round. Thus, the final allocation is determined based only on the inferred bids of the clock phase (Equation (3)).

- **Raised Clock Bids**: The bidders also provide their true value for all bundles they bid on during the clock phase.

- **Profit Max:** Bidders provide their true value for all bundles that they bid on in the clock phase, and additionally submit their true value for the $Q^{\text{P-Max}}$ bundles earning them the highest utility at the prices of the final clock phase. I.e., each profit-max bid can be seen as
  - one constrained DQ to determine which bundle is best given a price vector $p$, constrained on not choosing an already chosen bundle plus
  - one VQ to determine the exact value of this bundle.

  However, in Table 2 we count each profit-max query only as 1 VQ.

Note that our mechanism also allows the bidders to voluntarily provide push bids. These additional VQs could probably increase the efficiency of our mechanism even slightly further. However, we evaluate our mechanism in the worst case, where no push bids are provides to MLHCA. And the results in Table 2 show that even without receiving any push bids MLCA achieves the highest average efficiency in each tested domain.

## G.7. Revenue Results

| | **EFFICIENCY LOSS IN %** | | **RELATIVE REVENUE IN %** | |
|---|---|---|---|---|
| **DOMAIN** | **MLHCA** | **BOCA** | **MLHCA** | **BOCA** |
| **GSVM** | $0.00 \pm 0.00$ | — | $70.15 \pm 4.43$ | — |
| **LSVM** | $0.04 \pm 0.07$ | $0.39 \pm 0.31$ | $79.43 \pm 3.05$ | $73.53 \pm 3.72$ |
| **SRVM** | $0.00 \pm 0.00$ | $0.06 \pm 0.02$ | $56.05 \pm 1.69$ | $54.22 \pm 1.46$ |
| **MRVM** | $4.81 \pm 0.57$ | $7.77 \pm 0.35$ | $27.97 \pm 2.16$ | $42.04 \pm 1.89$ |

Table 10: Efficiency loss and relative revenue comparison between MLHCA (40DQs + 60VQs) and BOCA (100VQs). Shown are averages and a 95% CI. Winners marked in gray.

In Table 10, we present the relative revenue results of MLHCA and BOCA, both using VCG payments (see Appendix B.1). We define relative revenue as the percentage of optimal welfare recovered as revenue on a per-instance basis. For a detailed discussion of the corresponding efficiency results, please refer to Section 6.

Unlike efficiency, the best-performing mechanism in terms of revenue varies by domain. In the LSVM and SRVM domains, MLHCA generates higher revenue than BOCA, while in the MRVM domain, the opposite is true.

The explanation for MLHCA's higher revenue in the LSVM and SRVM domains is straightforward: MLHCA achieves higher efficiency than BOCA in these domains and still has at least 42 VQs remaining after matching BOCA's efficiency. This additional exploration afforded by those extra VQs enables MLHCA to identify many high-value allocations, ultimately driving up prices under the VCG payment rule.

The lower revenue of MLHCA compared to BOCA in the MRVM domain can be explained by examining the DQ rounds of MLHCA. In this domain, lower competition among bidders results in relatively low item prices, which reduces the inferred-to-true welfare ratio of the allocations based only on the DQs. This is illustrated in Figure 5. The low inferred value from these queries prevents them from driving up the VCG prices, even though they lead to allocations with high efficiency. As a result, in this domain, only the VQs contribute significantly to the auction's payments. Given that BOCA uses 100 VQs while MLHCA uses only 60, this difference leads to BOCA achieving higher revenue in the MRVM domain.

## G.8. Evaluating the Effectiveness of the Bridge Bid

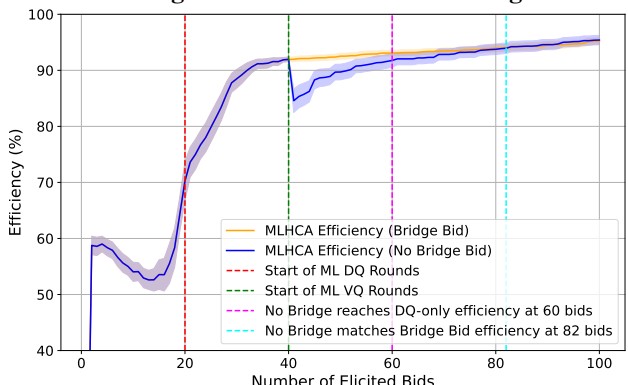

(a) Efficiency of MLHCA with and without the bridge bid (Definition D.8) in the MRVM domain.

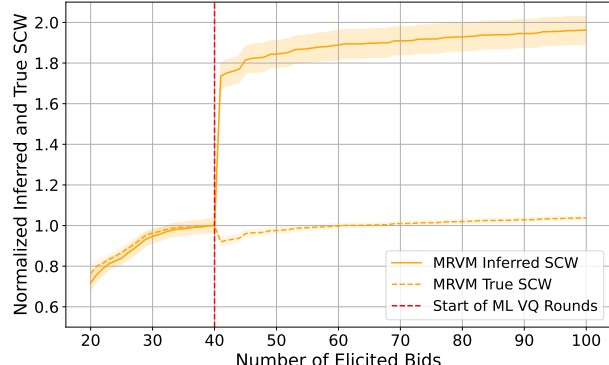

(b) Normalized inferred and true social welfare (SCW) of MLHCA without the bridge bid in the MRVM domain. Both quantities are normalized with respect to their average values at the start of the ML-based VQs.

Figure 4: Comparison of MLHCA's performance in the MRVM domain: (a) Efficiency with and without the bridge bid; (b) Normalized inferred and true SCW. Shown are averages over 50 instances including 95% CIs.

In this section, we experimentally evaluate the effectiveness of the bridge bid from Section 3.

In Figure 4a, we plot MLHCA's efficiency in the MRVM domain as a function of the number of elicited bids, comparing performance with and without the bridge bid. Without the bridge bid, we observe a significant efficiency drop of 7.3% points when MLHCA transitions to its VQ rounds. This is consistent with our theoretical results in Lemma D.7, where we showed that efficiency can decrease when the first VQ is introduced after DQs. In the MRVM domain, the most realistic setting, this effect is particularly pronounced. Notably, the auction requires 20 of our powerful ML-powered VQs just to recover the efficiency lost by the introduction of the first VQ. By contrast, using the bridge bid (Definition D.8) completely mitigates this efficiency drop, as predicted by Lemma D.9. However, as Figure 4a shows, if enough VQs are elicited, MLHCA without the bridge bid can eventually recover its efficiency, and both approaches converge to similar performance levels.

However, given that the auctioneer cannot determine the true efficiency of the auction at runtime, it is prudent to use the bridge bid version, which ensures consistent performance throughout the auction and significantly outperforms the alternative for the majority of rounds. Therefore, we consider this version the default approach for our MLHCA.

To better understand the cause of this efficiency drop, we refer to Figure 4b, where we plot the normalized inferred and true social welfare of MLHCA without the bridge bid in the MRVM domain. Both quantities are normalized to their values at the start of the ML-powered VQ rounds. At this point, we observe a stark contrast: the first VQ increases inferred social welfare by over 70%, while decreasing true social welfare by more than 7%. Before the ML-powered VQs, agents' reports were limited to their responses to DQs, and the auction's inferred social welfare was calculated based on the prices of the allocated bundles, as described in Equation (3). Due to the relatively low competition in MRVM, there was a substantial gap between the agents' true values and the inferred values based on their DQ responses.[27]

When the auction transitioned to VQs, agents responded with their true values for the queried bundles, leading to a sharp increase in inferred social welfare. However, the bidders' true values for the bundles they received during the DQ rounds were much higher than their inferred values, which the WDP failed to capture. As a result, transitioning to ML-powered VQs without the bridge bid caused a sharp increase in *inferred* social welfare alongside a drop in *true* social welfare.

This efficiency drop when transitioning to VQs is less pronounced in other domains. In Figure 5, we plot MLHCA's SCW for all domains as a function of the number of elicited bids, normalized to the start of the ML-powered VQ rounds. In these domains, the higher level of competition leads to inferred values for the queried bundles during the DQ rounds being much closer to the true values. Consequently, the bridge bid is less critical in these settings.

---

[27]Low competition in the auction can be gauged from its revenue, as, in the absence of reserve prices, revenue is primarily driven by competition among bidders. MRVM has the lowest ratio of revenue to welfare across all domains by a factor of nearly 2; see Appendix G.7.

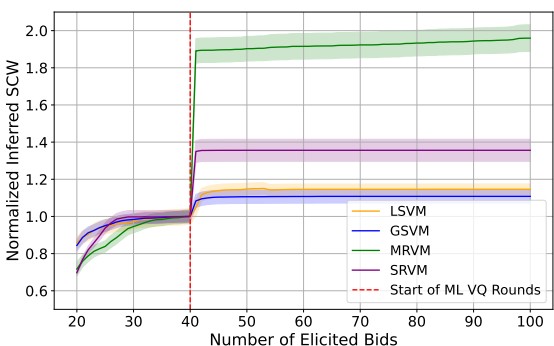

Figure 5: Normalized Inferred Social Welfare of MLHCA (with the bridge bid) in all domains. Shown are averages over 50 instances including 95% CIs.

## G.9. Inverse Query Order

| | EFFICIENCY LOSS IN % | | | | | QUERIES TO REJECT NULL HYPOTHESIS | | |
|---|---|---|---|---|---|---|---|---|
| DOMAIN | MLHCA | INVERSE | BOCA | ML-CCA$_{\text{CLOCK}}$ | ML-CCA$_{\text{RAISED}}$ | BOCA $\geq$ MLHCA | INVERSE $\geq$ MLHCA | ML-CCA$_{\text{RAISED}} \geq$ MLHCA |
| GSVM | 0.00 $\pm$ 0.00 | 0.28 $\pm$ 0.30 | — | 1.77 $\pm$ 0.68 | 1.07 $\pm$ 0.37 | — | 50 | 42 |
| LSVM | 0.04 $\pm$ 0.07 | 5.12 $\pm$ 2.18 | 0.39 $\pm$ 0.31 | 8.36 $\pm$ 1.70 | 3.61 $\pm$ 0.77 | 58 | 43 | 43 |
| SRVM | 0.00 $\pm$ 0.00 | 0.08 $\pm$ 0.16 | 0.06 $\pm$ 0.02 | 0.41 $\pm$ 0.11 | 0.07 $\pm$ 0.02 | 42 | — | 42 |

Table 11: MLHCA (40DQs + 60VQs) vs Inverse (80 VQs + 20 DQs), BOCA (100VQs), ML-CCA (ML-CCA$_{\text{clock}}$) (100DQs) and ML-CCA with raised clock bids (ML-CCA$_{\text{raised}}$) (100DQs and up to 100VQs). Shown are averages and a 95% CI. Winners based on a $t$-test with significance level of 5% are marked in grey.

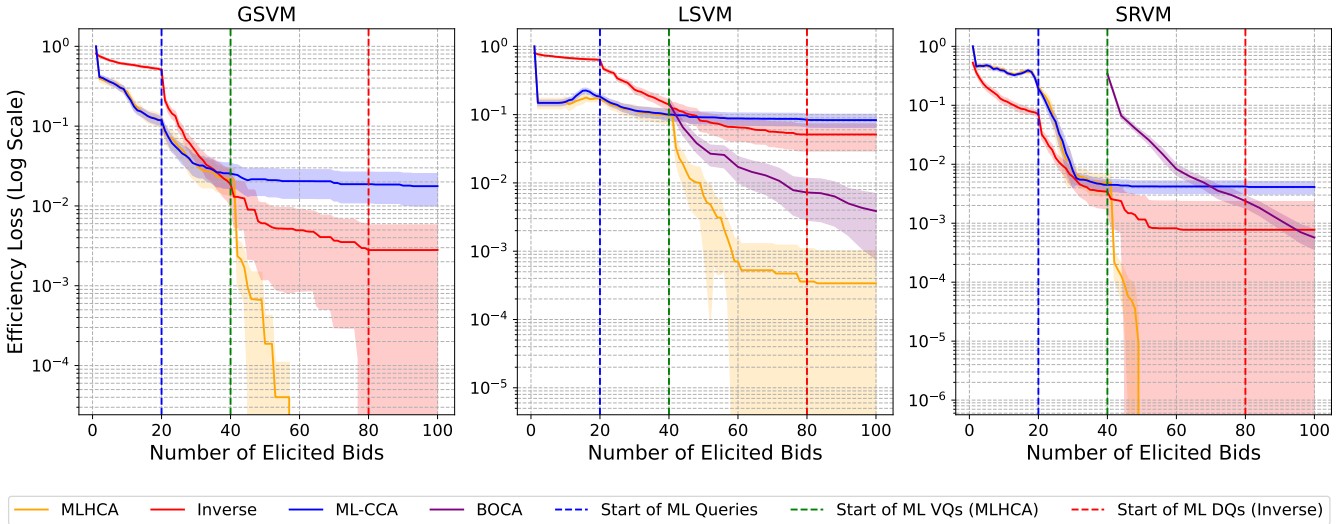

Figure 6: Efficiency loss paths (i.e., regret plots) of MLHCA compared to BOCA, ML-CCA and Inverse, an auction that inverses the order of DQs and VQs compared to MLHCA. Shown are averages over 50 instances with 95% CIs.

In this section, we experimentally evaluate how the order of queries affects auction performance. To this end, we introduce a variant of our mechanism, which we refer to as *Inverse*. This mechanism mirrors the structure of MLHCA described in Section 5, but with the query order reversed: *Inverse* begins with value queries (VQs), followed by demand queries (DQs). Concretely, while MLHCA starts with 20 CCA-based DQs to initialize learning, followed by 20 ML-powered DQs and then 60 ML-powered VQs, *Inverse* instead begins with 20 randomly chosen VQs, proceeds with 60 ML-powered VQs, and concludes with 20 ML-powered DQs. Unlike MLHCA, *Inverse* does not use the *bridge bid* introduced in Section 5. Because its final stage consists of DQs, there is no need for a special bid to preserve its DQ-only efficiency at later phases.

To isolate the effect of query order on auction efficiency, we evaluate all auction formats on the same set of auction instances. For the same reason, we use identical hyperparameters for both MLHCA and the *Inverse* auction, as described in Appendix G.3. For more details on the experimental setup, please refer to Section 6.1.

As in Section 6.2, Table 2 reports the average efficiency loss of each mechanism after 100 queries. For ML-CCA, we additionally report results assuming it is supplemented with the clock bids raised heuristic (see Section 2.2), which may require up to 100 additional VQs per bidder.[28] We also report how many queries MLHCA needs to statistically outperform the final efficiency of the other mechanisms.

We observe that in all domains tested,[29] MLHCA consistently outperforms the *Inverse* auction. In the LSVM domain, the efficiency difference reaches 5 percentage points, which would translate to welfare gains exceeding 200 million USD based on the value of goods typically traded in such auctions. Notably, in both the GSVM and LSVM domains, MLHCA statistically outperforms the *Inverse* auction at the 95% confidence level while using at most half as many queries. At this

---

[28]Under the clock bids raised heuristic, bidders report their value only for each unique bundle they bid on during the auction—up to 100 bundles for 100 DQs.

[29]Due to time constraints between the final reviews and the camera-ready deadline, results for the MRVM domain are not yet available.

point, *Inverse* has already asked each bidder 50 of the more cognitively demanding VQs, whereas MLHCA has used asked 10 VQs. In the SRVM domain, despite achieving higher average efficiency, MLHCA does not statistically outperform the *Inverse* auction. This is due to the simplicity of the domain: the *Inverse* auction reaches perfect (100%) efficiency in 49 out of 50 instances, rendering the two mechanisms statistically indistinguishable.

To gain a more detailed understanding of the impact of the inverse query order, Figure 1 illustrates the efficiency loss path for the GSVM, LSVM, and SRVM domains. Across all three domains, we observe that placing ML-powered DQs after the ML-powered VQs fails to improve the auction's efficiency. This is particularly noteworthy in the LSVM domain, where the efficiency loss after the ML-powered VQ phase was over 5 percentage points, meaning there was still significant room for improvement. Although the bidders' ML models become significantly more accurate once the auction reaches its DQ phase, the DQ algorithm is unable to leverage this improved model accuracy to generate more efficient outcomes.

Taken together, the results in this section provide strong empirical support for our theoretical analysis in Appendices D.3 and D.4: effective hybrid auctions should begin with DQs to gather global information about bidders' valuation functions, and only then transition to VQs to both refine understanding in the most critical regions of the allocation space and to obtain bids for bundles that are compatible for a feasible allocation. Across all tested domains, reversing this order results in substantial efficiency losses: ML-powered DQs fail to improve upon the efficiency achieved by the preceding VQs, while VQs, when used upfront, perform worse than in the standard order due to the reduced learning performance from training solely on VQs. These findings underscore the pivotal role of query ordering in the design of machine learning-powered combinatorial auctions.

While all subplots in Figure 6 clearly support our claim that asking DQs before VQs results in better final efficiency than asking them the other way around, certain details in the subplot for SRVM might raise some questions that we answer in Remarks G.1 and G.2.

*Remark* G.1 (Why are 20 random VQs better now than 20 random VQs were for BOCA for SRVM?). In Figure 6, for SRVM, we see that the efficiency of the *Inverse* auction after its initial random 20 VQs is approximately 4 times better than *BOCA* after its initial 40 random VQs. The reason for this, is a difference in the data pre-processing used for BOCA from (Weissteiner et al., 2023) and our data pre-processing from (Soumalias et al., 2024c). SRVM (Weiss et al., 2017) has 3 items with capacities $c_1 = 6, c_2 = 14, c_3 = 9$. In (Weissteiner et al., 2023), these items were treated as $c_1 + c_2 + c_3 = 6 + 14 + 9 = 29$ unique items. In other words, we and (Soumalias et al., 2024c) provide the mechanism with the prior knowledge that the first 6 items are indistinguishable copies of each other, and the next 14 items are indistinguishable copies of each other, and the remaining 9 items are indistinguishable copies of each other, while (Weissteiner & Seuken, 2020; Weissteiner et al., 2022b;a; 2023) did not use this prior knowledge in any form. This additional prior knowledge used by *Inverse*, but not used by BOCA, explains the advantage of *Inverse* over BOCA in this domain. The reason that those works did not leverage that information is that the *multiset* MVNNs required to capture that prior knowledge were only introduced in Soumalias et al. (2024c), which superseded these works. Nevertheless, the point of the inverse auction is not to compare against the SOTA ML-powered auction, BOCA, but to empirically evaluate the effect of switching the query order compared to MLHCA. In the end, BOCA can still outperform *Inverse* for 2 potential reasons: 1) BOCA keeps asking VQs until the end, while *Inverse* switches to DQs for the last 20 rounds, where DQs lose their effectiveness towards the end of the auction, or 2) BOCA explicitly fosters exploring unexplored regions of the bundle space, which gives it an advantage in the long run.

*Remark* G.2 (Are initial random VQs better for SRVM than initial CCA-DQs?). In Figure 6, we directly see that for GSVM and LSVM, starting the auction with 20 CCA-DQs is better than starting the auction with 20 random VQs. However, for SRVM, the efficiency after the initial random 20 VQs of *Inverse* is better than the efficiency after the initial 20 CCA-DQs of MLHCA or ML-CCA. Nevertheless, for all considered domains, including SRVM, we can see that MLHCA takes the lead after round 42. Our intuitive explanation is that 1) for all considered domains, including SRVM, the initial 20 CCA-DQs foster a better learning of the value functions, 2) the bundles that bidders request in the CCA-DQs, are usually incompatible, which seems to especially hurt the efficiency of SRVM, and 3) ML-VQs are always asked for perfectly compatible bundles, and if the models understand the value functions already sufficiently well, directly the first ML-VQ after the bridge bid can result in an allocation with very high efficiency (as intuitively demonstrated in Example 1 and as supported empirically by all our experiments). In summary, the efficiency observed from DQs can be misleading as a signal of learning progress, since the requested bundles are often incompatible and therefore fail to reflect the model's understanding. In contrast, the efficiency observed immediately after an ML-VQ provides a much more accurate reflection of the current learning state, as it elicits bids on perfectly compatible bundles across all bidders.

