# OpenReview forum: "Prices, Bids, Values: One ML-Powered Combinatorial Auction to Rule Them All"
_ICML.cc/2025/Conference — ICML 2025 oral_

### Official Review · Reviewer_6y6Q · 2025-03-12

**Overall Recommendation:** 4

**Summary:**

This paper focuses on iterative combinatorial auctions (ICAs), aiming to tackle the issue of exponential bundle space growth in combinatorial auctions. The authors introduce a machine learning (ML) algorithm that utilizes information from both demand queries (DQs) and value queries (VQs) and present the ML-powered Hybrid Combinatorial Auction (MLHCA).
MLHCA combines the advantages of DQs and VQs. DQs are more effective in the early auction stage as they are easier for bidders to answer and can provide global information about bidder preferences. VQs are more beneficial in the later rounds as they can precisely capture bidder values. By starting with DQs and then transitioning to VQs, MLHCA can achieve better learning performance.
The paper provides a theoretical framework for combining DQs and VQs, analyzes their advantages and limitations, and presents an algorithm. Experimental results show that MLHCA outperforms previous sota mechanisms in terms of efficiency and speed of convergence. Additionally, MLHCA is compatible with various payment and activity rules and can detect inconsistent misreports.

**Claims And Evidence:**

Yes.

**Essential References Not Discussed:**

No.

**Experimental Designs Or Analyses:**

Yes. No issues have been found so far.

**Methods And Evaluation Criteria:**

Yes.

**Other Comments Or Suggestions:**

No.

**Other Strengths And Weaknesses:**

1. High Efficiency: MLHCA significantly outperforms previous SOTA mechanisms, achieving high efficiency with fewer queries.
2. Reduced Cognitive Load: Combining DQs and VQs lessens bidders' cognitive load.
3. Theoretical and Practical Support: It has a solid theoretical framework and is compatible with practical rules.
4. Smooth Transition between DQs and VQs: I particularly appreciate the transition from DQs to VQs in MLHCA, which effectively combines the two types of queries, leveraging the benefits of each. The bridge bid, a specialized VQ, ensures a seamless transition, preventing potential efficiency drops and enabling the auction to maintain high performance throughout the process. This well-designed transition plays a crucial role here.

**Questions For Authors:**

As a reviewer, I'm concerned about the complexity of the MLHCA algorithm. However, I didn't find relevant analysis in the paper and its appendices. There are suspicions that some steps in the algorithm may have relatively high complexity. I hope the authors can offer a simple analysis of the algorithm's time and space complexity in the rebuttal, considering real-world scenarios to show its impact on practical performance.

**Relation To Broader Scientific Literature:**

The paper's key contributions significantly advance the fields of combinatorial auctions and machine learning. It improves upon traditional iterative combinatorial auctions (ICAs) like CCA, offering a more efficient and less cognitively burdensome alternative in MLHCA. In preference elicitation, the combination of DQs and VQs provides a new approach, enhancing learning performance. Regarding machine learning in auctions, the use of MVNNs and the mixed training algorithm add to the existing knowledge.

**Theoretical Claims:**

Yes. No issues have been found so far.

---

> ### Author Rebuttal · Authors · 2025-03-30
>
> Thank you for the positive feedback! If you have any additional questions, please let us know.
>
> “I am concerned about the complexity of the MLHCA algorithm”:
>
> In terms of theoretical time complexity, our ML-CCA, as every other ICA - including the CCA and all ML-powered auctions discussed - are NP-hard. That is because they need to solve the winner determination problem, which is an NP-hard combinatorial optimization problem.
> However, in practice the computational costs are very manageable, as discussed in Appendix G.4. Even in MRVM, the most computationally intensive and realistically-sized domain, the total time required to run all 100 rounds using 16 GB of RAM and 8 physical cores (clocked at 2.20GHz) was approximately 7 days, so all the necessary computations for 1 round take below 2 hours. This is not an issue, given that in the real-world, at most 2 clock rounds happen per day. For a real-world instance, just like for MRVM, the total computational cost in a paid cluster service for running our mechanism would be below 5 USD. Our efficiency improvements suggest that the welfare gains of doing so, compared to the CCA, would be approximately 100 million USD. For the other domains the computational costs are even lower. For the experimental results presented in this paper, we had to incur over 100 times the computational costs of a real-world instance.
> With that being said, we did put effort into reducing the computational cost of our algorithm. For example, we have empirically observed that our mixed query training algorithm needs approximately 10 times more epochs to converge than the DQ-only training algorithm from [Soumalias et al., 2024c]. From an applied spectrum-auction perspective, increasing the computational costs by a factor 10 does not matter, because this can easily be parallelized. However, for our academic budget when running hundreds of auctions a factor of 10 matters a lot. This is why we had to come up with the technique described in Remark E.1, to reduce computational time approximately by a factor 10.

---

### Official Review · Reviewer_xL2f · 2025-03-14

**Overall Recommendation:** 4

**Summary:**

This paper studies Iterative Combinatorial Auctions (ICA) and proposes a novel ML-based ICA mechanism, MLHCA. Their key empirical finding is that the mixed use of Value Queries (VQ) and Demand Queries (DQ) in their ML-based ICA mechanism significantly improves the efficiency and convergence of ICA compared to prior works. Additionally, they complement this finding with theoretical observations highlighting the disadvantages of using VQ or DQ alone.

**Claims And Evidence:**

The core claim of this paper is that a mixed use of value queries (VQ) and demand queries (DQ) in iterative combinatorial auction (ICA) can significantly improve the efficiency and convergence. This claim is primarily supported by experiments, with some (weak) theoretical observations on the disadvantages of using VQ or DQ alone.

**Essential References Not Discussed:**

All essential references are discussed to the best of my knowledge.

**Experimental Designs Or Analyses:**

Their statistical testing method appears sound and standard, and the use of a standard benchmark enhances the transparency of their results. Given that their improvement (i.e., smaller efficiency loss) is substantial across all four benchmarks, we have no objections to their findings or statistical analysis.

**Methods And Evaluation Criteria:**

They use a standard benchmark, the Spectrum Auction Test Suite (SATS, Weiss et al., 2017), which is a sensible choice and aligns with prior works cited in this paper.

**Other Comments Or Suggestions:**

I have no other comments.

**Other Strengths And Weaknesses:**

**Strengths**
- This paper presents a novel ML-based ICA mechanism that significantly improves the current state-of-the-art.
- The main idea of mixing VQ with DQ has the potential for broader impact and deserves a further study.

**Weakness**
- The theoretical argument is somewhat weak, as it only highlights the disadvantages of using VQ or DQ alone but does not provide a provable positive result for their mixed use, even in a simple toy model.

**Questions For Authors:**

Is there any hope of strengthening ex-post Nash incentive compatibility (i.e., truthful bidding is an ex-post Nash equilibrium) into dominant strategy incentive compatibility (i.e., truthful bidding is a dominant strategy), possibly under certain assumptions?

**Relation To Broader Scientific Literature:**

This paper contributes to the growing literature on ML-based ICAs by providing insights into the mixed use of value queries (VQ) and demand queries (DQ). While previous ML-based ICAs primarily relied on a single query type such as VQ-based approaches in Weissteiner et al., 2023 or DQ-based methods in Soumalias et al., 2024c, this work demonstrates that the hybrid approach significantly improves the efficiency and convergence of ICAs while minimizing bidder cognitive load.

**Theoretical Claims:**

I found no issue with theorems proven in this paper.

---

> ### Author Rebuttal · Authors · 2025-03-30
>
> Thank you for your positive feedback! If you have any more questions, please let us know.
>
>  “Does not provide a provable positive result for their mixed use, even in a simple toy model.”
>
> Lemmata D.9 and D.14, and Example 1 provide positive results, but we agree that our experimental positive results are stronger than our theoretical results. In Appendix D.3 we provide Example 1, which we believe provides very good intuition both on why DQs alone can be ineffective, and on why combining both query types can substantially increase efficiency.
>
> “Is there any hope of strengthening ex-post Nash incentive compatibility into dominant strategy incentive compatibility, possibly under certain assumptions”?
>
> Under extremely strong assumptions, namely that our auction can request an exponential number of value queries from the agents, it is trivial to show that it satisfies DSIC. Additionally, in appendix B.5, we explain how MLHCA can automatically detect if a bidder’s reports are inconsistent with any valuation function (and  automatically exclude them from the auction. This restricts a bidder's misreport space to consistent value functions. For more mild assumptions however, proving DSIC becomes challenging. Please note that DSIC has not been proven neither for the CCA, the most established ICA in the real world, nor for any of the ML-powered ICAs in the literature that we are aware of. However, in appendix B, we discuss in detail the incentive properties of our MLHCA, and we provide strong theoretical hints suggesting that MLHCA is more robust to strategic misreports than both the CCA, and the other ML-powered ICAs discussed in this paper. Please also see our reply to reviewer U2sY.

---

> > ### Comment · Reviewer_xL2f · 2025-04-09
> >
> > Thank you for your thorough response. After consideration, I will stand by my original score.

---

### Official Review · Reviewer_yLNu · 2025-03-15

**Overall Recommendation:** 3

**Summary:**

The paper introduces a new auction method called the Machine Learning-powered Hybrid Combinatorial Auction (MLHCA), designed to improve how items are sold in complex auctions where bidders can place offers on combinations of items. In these auctions, figuring out the best combination of bids is difficult because the number of possible combinations grows very quickly as more items are added. Traditional auctions either use demand queries (DQs) (where bidders state which combination they prefer at a certain price) or value queries (VQs) (where bidders say how much they value a specific combination). MLHCA combines both types of queries, starting with DQs to quickly gather general information and then switching to VQs to fine-tune the final outcome. The paper introduces a "bridge bid" to make the switch between DQs and VQs smooth.

The paper provides a few theoretical insights into the failure of DQ-only or VQ-only mechanisms and provides a new mechanism to mitigate those issues. The paper then conducts extensive experiments on standard datasets to validate the proposed mechanism on the standard datasets and shows that it outperforms existing models.

**Claims And Evidence:**

Most of the proposed claims are well supported. While the experiments clearly show the superior performance of the proposed mechanism which is the highlight of the submission, some of the theoretical insights in the paper are unclear and inadequate. Please see the weaknesses section for more details.

**Essential References Not Discussed:**

All relevant works are discussed in the paper to the best of my knowledge.

**Experimental Designs Or Analyses:**

The experiment design and analysis are correct to the best of my knowledge.

**Methods And Evaluation Criteria:**

The authors test MLHCA on realistic datasets from the spectrum auction test suite (SATS), which simulates different auction environments. They measure success using efficiency loss (how close the final allocation is to the optimal one) and the number of queries required to reach that outcome. MLHCA’s ability to reduce efficiency loss by up to a factor of 10 and cut down the number of queries by up to 58% demonstrates its practical value.

**Other Comments Or Suggestions:**

A: **Typos:**
Line 89, 2nd column: the word ‘preference’ is present twice.

Line 197, 1st column: Proofs are deferred to Appendix D.

Line 290, 1st column: An extra ‘R’ is present in the Algorithm 1.

Line 668, 669: Use \textsc{} for ‘NextQueries’ to be consistent.

Line 692, eq (5): What’s the significance of ‘1’ in \stackrel{}{}?

Line 843: Use \mathcal{X} to represent the class of feasible bundles.

Line 1012, 1013: Typo in word ‘appendix’.

Line 1017: MVNNscan -> MVNNs can.

Line 1162: inAppendix E.2 ->  in Appendix E.2

Line 1440: Typo in word ‘optimization’.

Line 1541: … we showed… instead of …we will show…

Line 1855: Missing full stop.

**Suggestions:**
A key aspect of the paper is combining the DQs and VQs and as the authors argue in Line 419-423 (2nd column), the way in which it is combined is important (which is also one of the key contributions of the work). The paper would have been more compelling if the authors had tried combining the DQs and VQs in other ways (VQs first, then DQs or interleaving them and so on), either theoretically or experimentally. That which would have highlighted the significance of the current method of combining them both.

As a lot of modules and subroutines in Algorithm 1 are from existing literature, it would help the paper if the authors precisely highlighted their novel contributions in designing the mechanism.

**Other Strengths And Weaknesses:**

A: **Strengths:**
The experiments are well designed and extensive which substantiate the superior performance of the proposed mechanism.
The notion of bridge bid is novel conceptually and its effect in practice is also highlighted well.

**Weakness:**
The assumptions on the value function are not clear. A crucial aspect in any combinatorial auctions is the structure of the value function. But the authors do not explicitly mention it in the work (Line 121 suggests it is non-negative). By reading the whole paper including the supplementary material, unless I am missing something, the only assumption on the value function is that it is non-negative and monotone. For the rest of my comments, I will assume this. In any case, the authors need to clarify this in the beginning.

**Demand Queries vs Value Queries.** A common theme which is repeated several times in the paper is that the DQs are cognitively simpler than VQs for the bidder. While it is well known that DQs are more informative than VQs in terms of eliciting preferences (which is also evident from the experiments), from a computational point of view, DQs can be quite hard to evaluate especially if the value function is not ‘simple’. Even for submodular value functions, this is known to be NP-Hard [FV10]. Furthermore, there are several experimental studies which suggest that it is hard for participants to optimally respond to a DQ [SZB12, BSW13]. Even the authors argue in Remark E.1 that evaluating a DQ with the MVNN approximating the value function is time consuming for which they reuse the values. So, how is responding to DQs cognitively simpler? In terms of the supermarket example which is used in the paper to provide intuition: if the value function is not ‘nice’ (linear/all-or-nothing etc), then for a given set of prices, to know the optimal combination of frying pans and coconuts, the bidder needs to evaluate the value of x frying pans and y coconuts in the first place which is equivalent to a VQ.

[FV10] Feige, Uriel, and Jan Vondrák. "The submodular welfare problem with demand queries." Theory of Computing 6.1 (2010): 247-290.

[SZB12] Scheel, Tobias, Georg Ziegler, and Martin Bichler. 2012. “On the Impact of Cognitive Limits in Combinatorial Auctions: An Experimental Study in the Context of Spectrum Auction Design.” Experimental Economics, 15: 667–692.

[BSW13] Bichler, Martin, Pasha Shabalin, and Jürgen Wolf. 2013. “Do Core-selecting Combinatorial Clock Auctions Always Lead to High Efficiency? An Experimental Analysis of Spectrum Auction Designs.” Experimental Economics, 16(4): 511–545.

**Questions For Authors:**

A:
In Lemma B.6, If L(theta)=0, why does it mean that the predicted best response matches the reported one? Am I correct to understand that you are assuming the optimization problem has a unique optimal value? If so, why is that the case? Moreover, the proof handles the DQ phase and VQ phase separately and obtains a contradiction under each setting. So, is it true that DQ-only methods such as ML-CCA can also detect inconsistent misreports?

In Proposition C.5, do MVNNs always learn an (almost) linear approximation of the value function or is it true only for the given example? More broadly, what is the importance of this result? If MVNNs always learn an (almost) linear approximation of the value function, the paper would be strengthened if the authors also consider a simple linear model to approximate the value function as a baseline to compare it against the more sophisticated MVNNs.

What is the significance of the 55% efficiency in Theorem 3.2? The proof only presents an example under which this efficiency can not be achieved. If I chose some other number, say 25%, does a DQ-only auction always guarantee 25% efficiency? I guess with appropriate tweaking of the parameters of the example, one can show that the previous statement is not true. So, why highlight the 55% figure? At a high level, I would suggest presenting this as an Observation/Fact instead of the full-fledged Theorem.

**Minor comments:**
What is the full form of DWP in footnote 15 in Line 1263?
A minor comment but what is the reasoning behind the three words ‘Prices, Bids, Value’ in the paper title?

**Relation To Broader Scientific Literature:**

The proposed two-stage mechanism overcomes several shortcomings of the DQ-only or VQ-only ML-powered mechanisms in the existing literature. The notion of the bridge bid introduced in this work also indicates that (perhaps) instead of the traditional DQs and VQs, future research in this area can focus on designing better queries or newer methods for combining DQs and VQs to elicit more information from the bidders instead of more elaborate NN models.

The key contribution of the paper is the mechanism in Algorithm 1 which leverages several existing methods and modules (MixedTraining (a straightforward extension of the TrainOnDQs module of [SWHS24] which uses both DQs and VQs instead of only DQs) and NextPrice [SWHS24]). A considerable portion of the theoretical contribution leverages the results from the MLCA paper [BLS19] and ML-CCA paper [SWHS24] which is cited in the submission.

[BLS19] Brero, Gianluca, Benjamin Lubin, and Sven Seuken. "Machine learning-powered iterative combinatorial auctions." arXiv preprint arXiv:1911.08042 (2019).

[SWHS24] Soumalias, Ermis Nikiforos, et al. "Machine learning-powered combinatorial clock auction." Proceedings of the AAAI Conference on Artificial Intelligence. Vol. 38. No. 9. 2024.

**Theoretical Claims:**

I checked all the proofs presented in the paper and they seem fine.

---

> ### Author Rebuttal · Authors · 2025-03-30
>
> Thank you very much for your very detailed review! If you have more questions, please let us know.
>
> “The assumptions on the value function are unclear”
>
> We only assume that $v(0) = 0$ and monotonicity, which combined imply non-negativity. Both of those assumptions are well-motivated and fairly standard. In Appendix C.1 we reference theoretical results that MVNNs can represent any monotonic function with $v(0)=0.$ We will clarify this in the final revision.
>
> “Combining DQs and VQs in other ways”
>
> In Appendices D and E we provide a lot of intuition on why starting with DQs and then querying VQs is expected to be much better than the other way around. Initializing an auction with random VQs (there are no better initialization algorithms for ML-powered VQs in the literature) will elicit the values of bundles which are far from optimal for the bidders, providing very little actionable information. The DQs at the end will be ineffective for two reasons: first, they will not provide any further information once the concave envelope of the value function is learned. Second, it is very likely that they will not result in reported bundles which nicely fit together (Example 1 provides good intuition for this). Therefore, we have good reasons to believe that swapping the order would significantly worsen the performance. Finally, one additional argument for this query order is that it results in the same interaction paradigm as the established CCA.  We will emphasize these arguments more in the final revision.
>
> “Is responding to DQs simpler than VQs?”
>
> While your argument on theoretical time complexity is correct, in practice, a bidder does not need to exactly evaluate her utility to answer a DQ. To make the supermarket example more intuitive:
> Suppose you go to the supermarket to buy a bundle of food items.
> Then you can easily disregard most other items, e.g., all the electronic items in the store. You only need to know that an electronic item’s added value for you is lower than its price tag. Reporting your precise value for a random bundle containing some food items, an iPad and a charger is much harder than deciding that you don’t want to buy the electronic items given their prices.
> Based on the experience of real-world consultants in spectrum auctions, there are usually many spectrum licenses that a bidder can easily disregard when answering a DQ, knowing that her values for them are below the posted prices without knowing how much below. In contrast to [SZB12], real-world bidders have much more expertise on the values of bundles that fit into their business model than the values of some random bundle. We agree that VQs for good bundles are easy to answer, but this is not true for random bundles in the real world.
>
> “A DQ with an MVNN approximating the value function is time consuming”
>
> Using 8 cores, it takes on average less than 200 ms. This does not relate to the human complexity for a bidder.
>
> “Does Lemma B.6 require a unique optimal solution?”
> Lemma B.6 does not require a unique optimal solution. Instead, all optimal solutions have the same predicted utility for the user, defined as <predicted bundle value> - <bundle price when the bidder requested it>.
>
> “Is it true that DQ-only methods such as ML-CCA can also detect inconsistent misreports?”
>
> Yes, we will mention this implication of our result in the final revision.
>
> “What is the meaning of proposition C.5? Do MVNNs always learn an almost linear approximation of the true value function?”
>
> MVNNs can fit queries that cannot be explained by a linear model (once those queries appear in the auction), which gives them a big competitive advantage over linear models. Proposition C.5 provides intuition on how MVNNs apply Occam’s razor in the case of not sufficiently informative queries. Proposition C.5 shows that in Example 1, the first VQ MLHCA would ask after the bridge bid would directly push the efficiency from ~55% to 100%.
>
> “What is the significance of the 55% efficiency in Theorem 3.2?”
>
> For the final version of the paper, we will modify the proof to show that for every $\epsilon>0$, there exist infinitely many instances where any DQ-only algorithm cannot achieve more than $50+\epsilon$% efficiency, no matter how many DQs it generates. Since such a family of instances exists, this constitutes a proof that DQs can be highly inefficient as the sole query type in an auction.
> Mathematically it would be an interesting open question if 50% can be reduced to 25%. However, from a practical point of view an efficiency of 50% corresponds to welfare losses of billions of dollars. This theoretically emphasizes that DQs massively lose their power in later rounds even if you ask infinitely many DQs, while VQs can always reach 100% efficiency. So, the claim that “DQs are more informative in an auction” is actually a much more nuanced discussion.
>
>
> Finally, thank you so much for the many typos you pointed out. We will correct them all for the final version of the paper.

---

### Official Review · Reviewer_U2sY · 2025-03-17

**Overall Recommendation:** 4

**Summary:**

The paper introduces MLHCA, a Machine Learning-powered Hybrid Combinatorial Auction that integrates demand queries (DQs) and value queries (VQs) to minimize efficiency loss in iterative combinatorial auctions. The authors provide theoretical insights demonstrating that DQs are most effective in the early auction rounds, while VQs enhance efficiency in later stages by refining allocations. This approach addresses the limitations of previous methods that relied solely on one query type. Empirical results show that MLHCA outperforms existing methods by achieving higher efficiency with fewer queries, significantly reducing bidders’ cognitive load.

**Claims And Evidence:**

Yes. All claims made in the paper are theoretical justified and are supported with adequate empirical validation. The paper convincingly demonstrated that the proposed approach is better than existing methods.

**Essential References Not Discussed:**

References are adequate.

**Experimental Designs Or Analyses:**

The experimental design is solid - with strong baseline comparisons (BOCA, ML-CCA, CCA) and real-world spectrum auction datasets (SATS). The metrics (efficiency and query count) are also appropriately chosen.

**Methods And Evaluation Criteria:**

Yes. They are well defined and well justified.

**Other Comments Or Suggestions:**

- A discussion on incentive compatibility would be nice.  Can the agents misreport their preference in the elicitation stage? How robust is the VQ/ DQ and proposed approach to such strategic misreports!
- Have you tried other approaches for enforcing monotonicity in neural networks (such as [this](https://arxiv.org/abs/2307.07512))

**Other Strengths And Weaknesses:**

In a way, this paper simply combines two existing approaches, but it does so in a principled way with theoretical justications and empirical validations!

**Questions For Authors:**

N/A

**Relation To Broader Scientific Literature:**

This paper extends prior work in ICAs by combining demand queries and value queries (VQs). The problem studied is significant and the proposed achieves better performance than existing SOTA approaches.

**Theoretical Claims:**

The proofs seem logically sound. I however didn't go through the proofs in the Appendix in detail

---

> ### Author Rebuttal · Authors · 2025-03-30
>
> Thank you for your positive and thorough feedback!  If you have any more questions, please let us know.
>
> “A discussion on incentive compatibility would be nice.”
>
> We discuss incentive compatibility in detail in Appendix B, where we try to provide intuition on this topic and prove some theoretical results under some assumptions. In summary, neither the most established mechanism, the CCA, nor any of the ML-powered approaches discussed in the literature are perfectly strategy-proof. However, there are strong theoretical hints suggesting that MLHCA is more robust to strategic misreports than the CCA, currently used in the real world, or most other ML-powered ICAs, such as Weissteiner et al. 2022a, 2023 and Soumalias 2024c. First, MLHCA’s DQ phase is compatible with the activity rules used in the CCA to improve its incentive properties (Appendix B.3). Second, MLHCA can automatically  detect when a bidder provides inconsistent reports (Lemma B.6). Third, MLHCA leverages marginal economies in the same way as the VQ-based ML-powered auctions to further align incentives. Fourth, as pointed out in Remark B.7, our experiments show that Assumption 2 in Appendix B.5 has become much more realistic for MLHCA than for any previously proposed mechanism.
>
> "Other approaches for enforcing monotonicity in neural networks?"
>
> No, we did not compare MVNNs against other architectures. MVNNs have a strong theoretical foundation and have proven their success in learning value functions across multiple market design papers [Weissteiner et al., 2022a,Weissteiner et al., 2023,  Soumalias et al. 2024b,c]. Additionally, since MVNNs are the main architecture behind these papers that we compare against, by using the same architecture, we can more easily isolate the effect of our query generation algorithms and the learning advantages of combining both query types. From an auction perspective, finding alternatives to MVNNs probably does not have the highest priority, since MVNNs work very well in practice. Finally, there is also no evidence suggesting that the suggested architecture would perform better, as the authors of that paper did not compare against MVNNs.
>
> “In a way, this paper simply combines two existing approaches, but it does so in a principled way with theoretical justifications and empirical validations!”
>
> Thank you for this comment—we believe it is a fair and accurate characterization of our work. Indeed, this was precisely the goal of our paper: to combine the two most prominent query types in iterative combinatorial auctions (ICAs) in a principled and effective way. We used our theoretical results to achieve an unprecedented improvement in efficiency while at the same time trying to keep the interaction paradigm as close as possible to the most established interaction paradigm used in practice.
>
> Our aim was not to propose a radically new interaction paradigm, but to improve auction performance in ways that are realistically implementable in high-stakes settings like spectrum auctions. To do this, we developed a new theoretical framework for understanding the advantages of combining demand and value queries, both from a combinatorial optimization and a machine learning perspective. For all theoretical results, we were able to experimentally demonstrate their real-world significance.
>
> We then leveraged these insights to create MLHCA, an auction that follows the same interaction paradigm as the established CCA, yet achieves substantial perf
> ormance gains. MLHCA is the most efficient auction in practice, reducing efficiency loss by up to a factor of 10 and surpassing the previous state of the art while using up to 58% fewer queries.
>
> While MLHCA’s similarity to the CCA may make it appear less novel at first glance, we believe this is a strength. By identifying the core challenge in value elicitation and addressing it through minimal yet effective changes, we were able to dramatically improve efficiency while remaining fully compatible with established, battle-tested, auction paradigms.
>
> Soumalias, E., Zamanlooy, B., Weissteiner, J., and Seuken,S.
> Machine learning-powered course allocation. EC 2024b
>
> Soumalias, E. N., Weissteiner, J., Heiss, J., and Seuken, S.
> Machine learning-powered combinatorial clock auction. AAAI 2024c
>
> Weissteiner, J., Heiss, J., Siems, J., and Seuken, S.
> Monotone-value neural networks: Exploiting preference monotonicity in combinatorial assignment. IJCAI 2022a
>
> Weissteiner, J., Heiss, J., Siems, J., and Seuken, S.
> Bayesian optimization-based combinatorial assignment.  AAAI 2023

---

### Decision · Program_Chairs · 2025-05-01

**Decision:**

Accept (oral)

**Comment:**

This paper makes a strong contribution to the literature on ML-based auction design. The paper introduces a novel method to the mixing demand and value queries in combinatorial auctions, which induces in practice a very natural transition between the two. The reviewers appreciated the theoretical and empirical contributions of the work, and all were supportive of acceptance. That said, the reviewers had a number of valuable suggestions on the paper, including on the overall perspective, that the authors should incorporate in any revision.